# The histone demethylase Kdm6b regulates subtype diversification of mouse spinal motor neurons during development

Wenxian Wang[1,3], Hyeyoung Cho[2,3], Jae W. Lee [1] & Soo-Kyung Lee[1✉]

How a single neuronal population diversifies into subtypes with distinct synaptic targets is a fundamental topic in neuroscience whose underlying mechanisms are unclear. Here, we show that the histone H3-lysine 27 demethylase Kdm6b regulates the diversification of motor neurons to distinct subtypes innervating different muscle targets during spinal cord development. In mouse embryonic motor neurons, Kdm6b promotes the medial motor column (MMC) and hypaxial motor column (HMC) fates while inhibiting the lateral motor column (LMC) and preganglionic motor column (PGC) identities. Our single-cell RNA-sequencing analyses reveal the heterogeneity of PGC, LMC, and MMC motor neurons. Further, our single-cell RNA-sequencing data, combined with mouse model studies, demonstrates that Kdm6b acquires cell fate specificity together with the transcription factor complex Isl1-Lhx3. Our study provides mechanistic insight into the gene regulatory network regulating neuronal cell-type diversification and defines a regulatory role of Kdm6b in the generation of motor neuron subtypes in the mouse spinal cord.

[1] Department of Biological Sciences, College of Arts and Sciences, University at Buffalo, The State University of New York (SUNY), Buffalo, NY 14260, USA.
[2] Computational Biology Program, School of Medicine, Oregon Health & Science University, Portland, OR 97239, USA. [3]These authors contributed equally:
Wenxian Wang, Hyeyoung Cho. ✉email: slee229@buffalo.edu

The central nervous system comprises amazingly divergent neuronal cell types, most of which are generated during the early stages of life. Different neuronal types have distinct identities in cell body locations, axonal trajectory patterns, and synaptic partners. Achieving this level of cellular heterogeneity involves the diversification of neuronal types following the initial neurogenesis and requires intricate transcriptional regulations. Although the diversification of neuronal population is crucial in building a functional neural circuit, the gene regulatory mechanisms by which a neuronal population further differentiate into subtypes remain poorly understood, particularly compared to the initial neuronal differentiation process.

Spinal motor neurons (MNs) are one of the best-studied systems for the postmitotic diversification of neuronal fates. MNs serve as the motor circuit's final and critical step through pre- and post-synaptic connections with upstream afferent neurons and downstream peripheral muscular targets, respectively, thus channeling enormously versatile behaviors[1,2]. MNs are produced from neural progenitors in a narrow time window during embryogenesis (E9.5–E13.5 in mice). As MN progenitors exit the cell cycle and differentiate into neurons, they also acquire a pan-MN identity, such as expression of cholinergic neurotransmitter genes and transcription factors (TFs) Isl1, Lhx3, and Mnx1 (also known as Hb9)[3–10]. Notably, the newborn MNs with pan-MN characteristics further differentiate into various MN subtypes that innervate distinct muscular targets. Cell bodies of the same MN subtypes are clustered in an orderly manner, forming motor columns in confined spinal cord areas, and project their axons to similar peripheral targets[11]. Thus, MN subtype diversity is critical for connecting a large number of peripheral targets with different segments of the spinal cord. Across the whole spinal cord, the medial motor column (MMC) resides in the ventromedial region and innervates the dorsal axial muscles. At the lower cervical and lumbar levels, the lateral motor column (LMC) MNs are formed and migrate to the ventrolateral region of the spinal cord, and their axons connect to limb muscles. At thoracic levels, the hypaxial motor column (HMC) and the preganglionic motor column (PGC) are clustered in ventrolateral and mediolateral areas of the spinal cord, respectively. HMC MNs innervate intercostal and abdominal muscles, whereas PGC MNs target sympathetic ganglia. MNs in the same column share overall cell body positions and axonal trajectory, and they can be divided into subgroups to connect to specific muscle groups within the common peripheral target. For instance, LMC differentiates into the two subtypes, medial LMC (LMCm) and lateral LMC (LMCl), which targets the ventral and dorsal limb muscles, respectively. However, it remains unclear if other motor columns also subdivide into distinct subgroups.

Multiple TFs were initially recognized to mark different motor columns in a combinatorial manner[5,6]. Subsequent studies have revealed that these TFs are essential players of the intricate transcriptional regulatory mechanism that shapes MN subtype identity[11]. As MN progenitors are about to exit the cell cycle, they upregulate the expression of Isl1, Lhx3, Lhx4, and Mnx1[7–10]. Lhx3 and Lhx4 exhibit a similar expression pattern and act redundantly[7]. Isl1 and Lhx3 associate with a cofactor Ldb (LIM domain binding, aka NLI) to form the Isl1-Lhx3 complex (aka MN-hexamer) consisting of two Isl1, two Lhx3, and two Ldb molecules[4,12–14]. The Isl1-Lhx3 complex binds and activates pan-MN genes, including a battery of cholinergic neurotransmitter genes, establishing generic MN identity in newborn neurons[12,14–20]. The autoregulatory feedback of Isl1-Lhx3, combined with the induction of Mnx1 and Lmo4 by Isl1-Lhx3, is crucial to consolidate the pan-MN identity while suppressing the unwanted V2 interneuron fate promoted by Lhx3[13,21,22]. Interestingly, as newborn MNs further differentiate to gain MN subtype identity, all MNs, except MMC, downregulate Lhx3[7]. Thus, among motor columns, the co-expression of Isl1 and Lhx3 is restricted to MMC MNs, in which the Isl1-Lhx3 complex plays a role in consolidating MMC identity[4,23]. In contrast, Foxp1 is upregulated in LMC and PGC and drives the formation of LMC and PGC, in part via collaboration with Hox genes[24,25]. Then, LMC is further divided into Isl1-expressing LMCm and Lhx1-expressing LMCl. Notably, Lhx3 and Foxp1 exhibit the exclusive expression patterns among MN subtypes and promote the different motor column identities[24,25]. In Foxp1-deficient mice, Lhx3 expression expands and, concurrently, MMC and HMC increase at the expense of LMC and PGC[25], suggesting MMC and HMC may share the developmental pathway suppressed by Foxp1 and promoted by Lhx3. Supporting this notion, HMC and MMC show a similarity in their molecular profiles, such as co-expression of Isl1 and Mnx1, dendritic morphology, and premotor input patterns[26]. To date, however, the developmental relationship between MMC and HMC as well as the HMC-defining TFs remain ambiguous.

Given that transcriptional coregulators are required for TFs to control gene expression[27,28], they are likely to play essential roles in neuronal diversification. However, the role of transcriptional coregulators in MN diversification remains unclear. Histone-modifying enzymes constitute an important group of transcriptional coregulators. In particular, H3K27 demethylases (H3K27DMs) remove the transcriptionally repressive histone mark histone H3-lysine27 tri-methylation (H3K27me3), and thus they are involved in transcriptional activation[29]. Thus far, three H3K27DMs have been identified; Kdm6a (Lysine Demethylase 6a, aka UTX), Kdm6b (Lysine Demethylase 6b, aka Jmjd3), and Kdm6c (Lysine Demethylase 6c, aka UTY). Kdm6a and Kdm6b are the main H3K27DMs involved in cell differentiation and organ development, whereas Kdm6c displays only mild enzymatic activity and complements Kdm6a[29–31]. Kdm6a and Kdm6b can be recruited to the specific target genes by associating with their partner TFs and activate the target gene transcription by triggering changes in histone marks and thus chromatin landscape[29,32]. Both Kdm6a and Kdm6b are implicated in neural stem cell generation and neurogenesis[33–37], but their role in neuronal diversification remains unexplored.

In this study, we investigated the role of H3K27DMs in MN development. Interestingly, Kdm6b, but not Kdm6a, was required for MN subtype diversification. The inactivation of Kdm6b in MNs resulted in a marked loss of MMC and HMC and a concomitant gain of LMC and PGC. Our single cell RNA-sequencing (scRNAseq) analyses of embryonic MNs revealed new developmental logics and marker genes for MN subtypes and a remarkable heterogeneity of PGC and MMC MNs. Moreover, the systematic comparison between Kdm6b-deficient and control MN transcriptomes identified Kdm6b-dependent gene profiles in MN subtypes. Further analyses, combining bioinformatics, mouse genetics, and molecular methods, uncovered that Kdm6b acts as a transcriptional coactivator for Isl1-Lhx3 in promoting MMC fate and suppressing LMC and PGC identities. Together, our studies identified a critical component of the transcriptional regulatory networks driving MN subtype diversification and demonstrated a central role of Kdm6b in this context.

## Results

**Kdm6b plays a crucial role in MN differentiation and columnar diversification.** *Kdm6b* mRNA transcripts were upregulated as newborn MNs emerged from the progenitors in the spinal cord at E10.5, and *Kdm6b* levels remained higher in neurons than progenitors at E12.5 (Fig. 1a). Strong induction of *Kdm6b* expression during MN differentiation prompted us to ask if

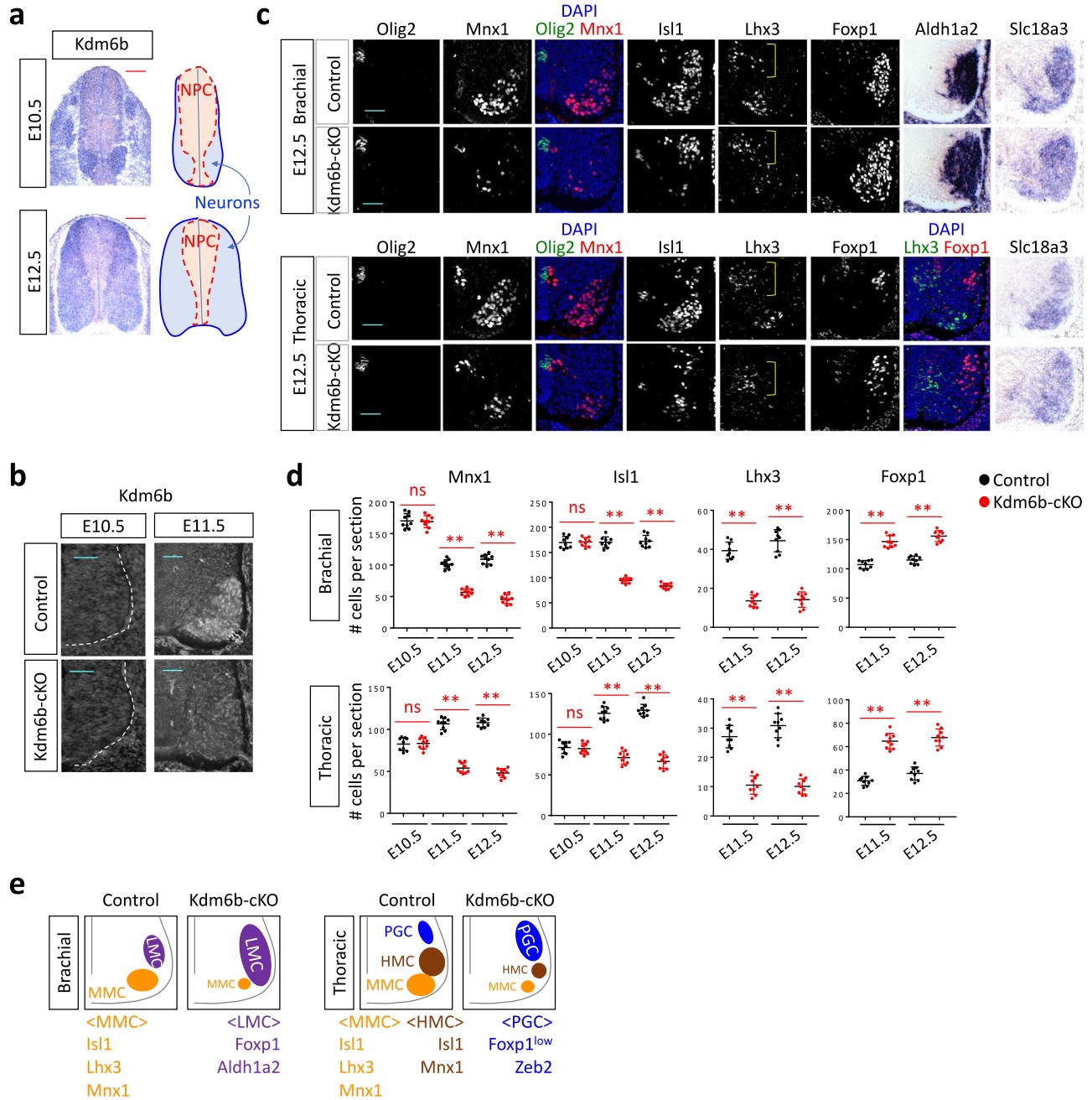

**Fig. 1 Kdm6b is required for the balanced production of MN columnar subtypes. a** The expression pattern of Kdm6b in the developing spinal cord at E10.5 and E12.5, as detected by in situ hybridization analyses. Scale bars, 100 μm. The schematics depict the location of neural progenitor cells (NPC) in the ventricular zone and neurons in the mantle zone. **b** Immunohistochemical analysis with Kdm6b antibody shows that Kdm6b protein was detected in E11.5 control and eliminated in E11.5 *Kdm6b*-cKO mice. Scale bars, 50 μm. $n = 3$ mice per genotype. **c** Immunohistochemical analyses with Olig2, Mnx1, Isl1, Lhx3, and Foxp1 antibodies and in situ hybridization analyses for Aldh1a2 and Slc18a3 in the brachial or thoracic levels of E12.5 spinal cords. In E12.5 *Kdm6b*-cKO mice, the MNs expressing Mnx1, Isl1, or Lhx3 decreased, whereas the MNs expressing Foxp1 or Aldh1a1 increased. The brackets indicate Lhx3$^+$ V2 interneurons. Scale bars, 50 μm. $n = 3$–6 mice per genotype. **d** Quantification of the number of MNs expressing Mnx1, Isl1, Lhx3, or Foxp1 per 12 μm thick section of the spinal cord at E10.5, E11.5, and E12.5. Mnx1$^+$, Isl1$^+$, and Lhx3$^+$ MNs markedly decreased in Kdm6b-deficient mice at E11.5 and E12.5, but not at E10.5. In contrast, Foxp1$^+$ MNs significantly increased in *Kdm6b*-cKO spinal cords at E11.5 and E12.5. The error bars represent the standard deviation of the mean. **$p < 0.001$; ns non-significant in the two-tailed Student's *t*-test. $n = 3$–6 mice per genotype and 9 slices per genotype. Source data are provided as a Source Data file. The exact *p*-values are as follows. Brachial, Mnx1, E10.5, 0.83; E11.5, $7.14 \times 10^{-11}$; E12.5, $3.67 \times 10^{-12}$, Isl1, E10.5, 0.81; E11.5, $2.27 \times 10^{-12}$; E12.5, $6.48 \times 10^{-13}$, Lhx3: E11.5, $1.38 \times 10^{-10}$; E12.5, $5.78 \times 10^{-10}$; E11.5, $4.39 \times 10^{-8}$; E12.5, $9.94 \times 10^{-9}$. Thoracic, Mnx1, E10.5, 0.82; E11.5, $1.07 \times 10^{-11}$; E12.5→$7.33 \times 10^{-14}$, Isl1, E10.5, 0.70; E11.5, $1.93 \times 10^{-10}$; E12.5, $1.09 \times 10^{-11}$, Lhx3, E11.5, $5.52 \times 10^{-10}$; E12.5, $8.05 \times 10^{-10}$, Foxp1, E11.5, $5.52 \times 10^{-10}$; E12.5, $2.67 \times 10^{-8}$. **e** Schematics for altered MN columnar diversification in *Kdm6b*-cKO spinal cords in comparison to control spinal cords.

Kdm6b plays any role in MN development. Thus, we generated MN-specific Kdm6b conditional knockout mice (*Kdm6b^f/f;Olig2-Cre*, referred to as *Kdm6b*-cKO), in which Kdm6b[38] was deleted by *Olig2-Cre*[39]. *Kdm6b*-cKO mice died perinatally, suggesting a role of Kdm6b in *Olig2-Cre*-active cell lineages.

Kdm6b protein was detected in MNs at E11.5, but not at E10.5, and it was largely eliminated in MNs of E11.5 *Kdm6b*-cKO mice (Fig. 1b). In *Kdm6b*-cKO mice, newborn MNs marked by Isl1 and Mnx1 did not change at E10.5, but they significantly reduced by E11.5 when MNs began to acquire columnar identities (Fig. 1c, d, Supplementary Fig. 1a, c). Olig2$^+$ MN progenitors did not change (Fig. 1c, Supplementary Fig. 1c).

The combinatorial expression of Isl1, Lhx3, Mnx1, and Foxp1 marks distinct motor columns (Fig. 1e). *Kdm6b*-cKO mice showed remarkable changes in the composition of MN subtypes at E11.5 and E12.5, but not at E10.5, while maintaining a cholinergic marker *Slc18a3* (aka *Vacht* for vesicular acetylcholine transporter)-expressing MN areas similar to the control mice (Fig. 1c, d, Supplementary Fig. 1a–c). In *Kdm6b*-cKO mice, Lhx3$^+$ MNs markedly decreased, whereas Foxp1$^+$ MNs significantly increased in both brachial and thoracic levels at E11.5 and onward, indicating reduced MMC and increased LMC and PGC MNs. Consistently, the expression domain of *Aldh1a2*, an LMC marker gene, was expanded both ventrally and medially in the brachial spinal cord of *Kdm6b*-cKO mice. Notably, Isl1$^+$ and Mnx1$^+$ MNs also reduced in *Kdm6b*-cKO mice, suggesting a decreased HMC population in *Kdm6b*-null MNs.

While Olig2-lineage cells produce V3 interneurons (V3IN) and a subset of V2 interneurons (V2IN)[39,40], V3IN and V2IN did not show a significant change in *Kdm6b*-cKO mice (Supplementary Fig. 1c, d).

Together, our data establish that LMC and PGC MNs increase at the expense of MMC and, likely, HMC MNs in the Kdm6b-deficient MN lineage (Fig. 1e). These results suggest that Kdm6b plays a crucial role in promoting MMC and HMC fates while repressing LMC and PGC identities.

**Motor axonal trajectory changes in the absence of Kdm6b.** To examine how MN subtype alterations affect motor axonal projections in *Kdm6b*-cKO mice, we generated *Kdm6b*-cKO (*Kdm6b^f/f;Olig2-Cre*) and control (*Kdm6b^f/+;Olig2-Cre*) mice that carry the Cre-dependent reporter allele ROSA$^{mT/mG}$ (hereafter referred to as *mTmG*)[41]. As *Olig2-Cre* active cells and their progeny express the cell membrane-localized GFP (mGFP) in these mice, only motor axons were labeled with mGFP in the periphery (Supplementary Fig. 2a, b). We first investigated GFP$^+$ motor axonal projection patterns in transverse sections of embryos. The MMC axon bundles that make a turn to the dorsal axial musculature were markedly reduced, and less furcated in *Kdm6b*-cKO mice compared to control mice (Fig. 2a). In contrast, the LMC nerves targeting limb muscles and the PGC axonal bundles innervating the sensory ganglia became thicker in *Kdm6b*-cKO mice (Fig. 2a). We next acquired a broader view of motor axon trajectory in the periphery using the half-mount embryo preparation (Supplementary Fig. 2b). The analyses of LMC axons innervating the forelimb revealed that most LMCl-derived nerves, such as the radial and thoracic dorsal nerves[42], became thicker and elongated, whereas LMCm-derived nerves, such as the median and medial anterior thoracic axon bundles, became thinner and showed drastically reduced branches in *Kdm6b*-cKO (Fig. 2b). Of note, among LMCl-derived nerves, only the axillary nerve was thinner in *Kdm6b*-cKO mice (Fig. 2b). *Kdm6b*-cKO mice also exhibited aberrant HMC-derived intercostal nerve patterns (Fig. 2c, Supplementary Fig. 2c). The termini of the intercostal nerves of *Kdm6b*-cKO were elongated, merging

to form ectopic axon bundles close to the edge of the intercostal nerves. The intercostal nerves of *Kdm6b*-cKO also displayed aberrant axonal bridges across multiple nerve bundles at lower thoracic levels, suggesting axonal projection errors. In addition, the ambilateral branches at the periphery of the intercostal nerves of *Kdm6b*-cKO mice were longer and less complex than those of control mice.

Together, *Kdm6b*-cKO mice exhibited various motor axonal projection deficits that were highly correlated to the changes in MN subtype composition. In *Kdm6b*-null mice, MMC and HMC axons showed thinner bundles, axon projection errors, and reduced branches, whereas the overall LMC nerves, particularly LMCl-derived axon bundles, became thicker. These results underline that Kdm6b is required for establishing the appropriate motor axonal trajectory pattern.

**Kdm6a is not needed for MN differentiation and diversification.** We considered the possibility that Kdm6a, the only other major H3K27DM, plays a redundant function with Kdm6b in MN development. Interestingly, *Kdm6a* mRNA expression was higher in progenitors than in neurons (Supplementary Fig. 3a), which is largely complementary to *Kdm6b* expression (Fig. 1a). *Kdm6a*-cKO (*Kdm6a^f/f;Olig2-Cre*) mice survived into adulthood and did not show significant changes in the generation or diversification of MNs (Supplementary Fig. 3b, c). These results indicate that Kdm6a is dispensable for MN development.

**Single-cell transcriptome and unsupervised clustering analyses of *Olig2-Cre* lineage cells.** To characterize Kdm6b-deficient MNs at single-cell levels using a genome-wide tool, we dissected low cervical to upper lumbar segments (C4-L2) of the spinal cord from E12.5 control mouse with the *mTmG* reporter, isolated GFP$^+$ cells using FACS, and performed scRNAseq assay using 10X Chromium Single Cell platform[43] (Supplementary Fig. 4a). Next, we subjected the single-cell transcriptomes of GFP$^+$ control cells to unsupervised clustering using Seurat[44], which identified 13 clusters (Fig. 3a). As GFP$^+$ cells represent cells derived from *Olig2-Cre* active cells, we named these clusters as OC for *Olig2-Cre* lineage clusters. The 13 clusters (OC0-12) belong to four groups of cell types (Fig. 3b–j, Supplementary Fig. 4b–d). First, OC0, 7, and 9 represent progenitors that show a high-level expression of progenitor markers Sox2 and Hes5 and a low-level of pan-neuronal genes, Nefl (Neurofilament light) and Nefm (Neurofilament medium). The expression pattern of markers of spinal progenitor domains[3] revealed that these progenitor clusters contain p2, pMN, p3, and floor plate cells. Second, OC1, 3, 6, and 12 belong to the V3IN group, given that they express V3IN markers, Sim1 and Nkx2-2, and a glutamatergic neuronal marker gene Slc17a6 (aka Vglut2 for vesicular glutamate transporter). Third, OC8 and 10 belong to the V2IN group. OC8 consists of glutamatergic V2a interneurons (V2aIN) expressing Slc17a6 and Vsx2, whereas OC10 is composed of GABAergic V2b interneurons (V2bIN) marked by Gad1 (for glutamate decarboxylase), Gata2/3, and Tal1 (aka Scl). Notably, among clusters in V3IN and V2IN groups, a small number of cells in OC8 and 12 expressed a low level of Slc18a3. Lastly, a high level of Slc18a3 indicated that OC2, 4, 5, and 11 are MNs. The combinatorial expression of Isl1/2, Lhx3/4, Mnx1, and Foxp1, along with Aldh1a2 and Nos1, assigned the identity of OC2, 4, and 5 as LMC, PGC, and MMC, respectively. OC11 had mixed cells with LMC, PGC, and MMC identities.

Together, the single-cell transcriptome analysis of *Olig2-Cre* lineage cells from the control embryo defined the cell types derived from *Olig2-Cre* active cells, supporting the notion that

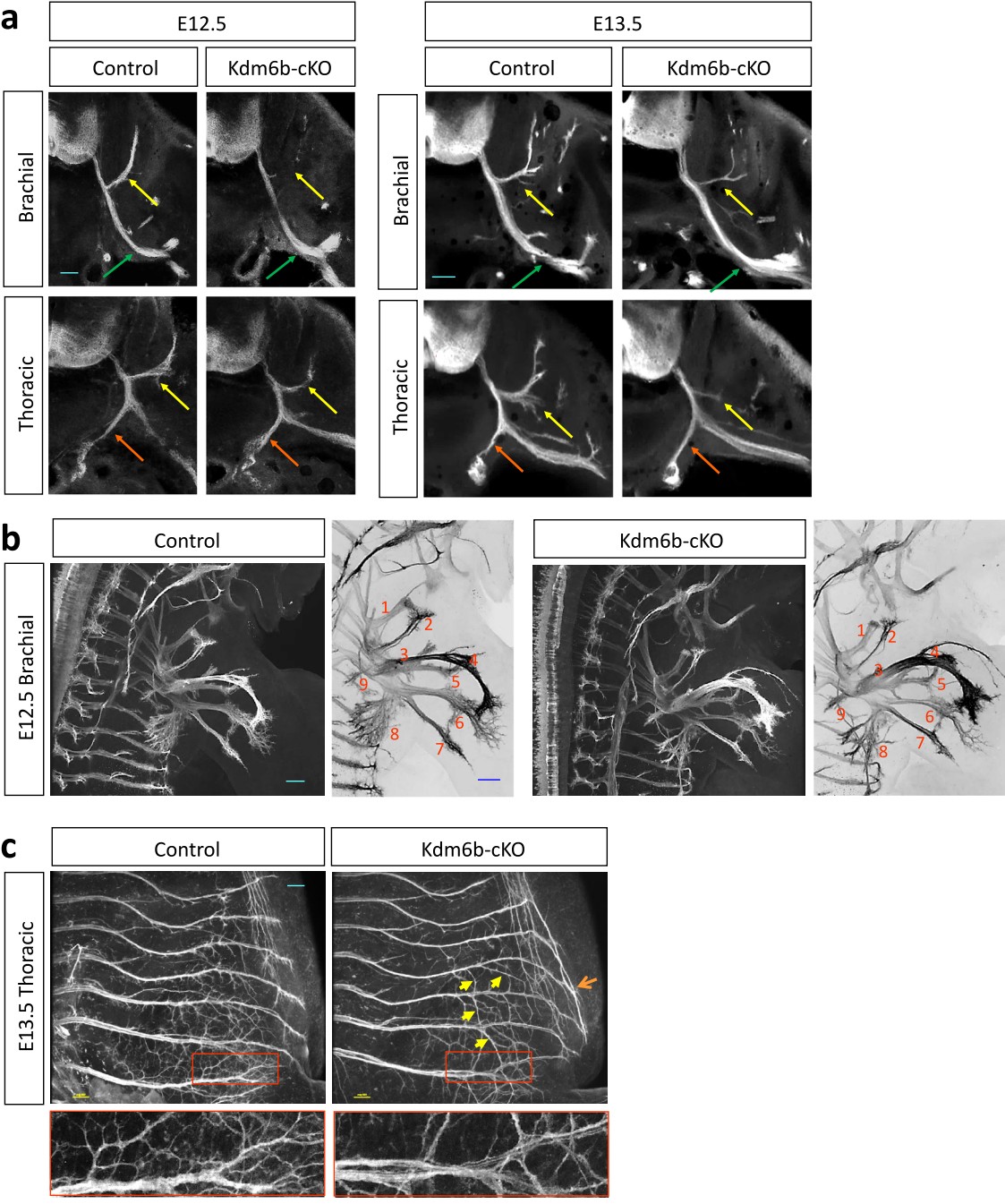

**Fig. 2 Kdm6b is required for establishing an accurate motor axonal trajectory pattern. a** Analysis of GFP+ motor axonal projection pattern in the periphery in transverse sections of E12.5 and E13.5 mice. In *Kdm6b*-cKO mice, MMC axonal bundles (yellow arrows) were reduced in diameter and less furcated, whereas LMC axons (green arrows) and PGC axons (red arrows) were thickened in brachial and thoracic levels, respectively. Scale bars, 100 μm. **b** The lateral view of E12.5 embryos to analyze GFP+ motor axonal trajectory into the forelimb muscles. Most LMCl-derived axonal bundles, such as the radial and thoracic dorsal nerves, were thicker and more elongated, whereas LMCm-derived nerves, such as the median and medial anterior thoracic axon bundles, were thinner and less branched in *Kdm6b*-cKO than control mice. Forelimb nerves from LMC neurons are numbered as follows: supra scapular, 1; axillary, 2; lateral pectoral, 3; radial, 4; musculocutaneous, 5; median, 6; ulnar, 7; medial anterior thoracic, 8; thoracic dorsal, 9. Scale bar, 100 μm. **c** The lateral view of E13.5 embryos to analyze HMC-derived GFP+ motor axonal trajectory at thoracic spinal cord levels. In *Kdm6b*-cKO mice, HMC axonal termini became elongated and merged to form ectopic axon bundles close to the edge of the intercostal nerves (orange arrows). The intercostal nerves of *Kdm6b*-cKO also formed aberrant axonal bridges across multiple nerve bundles at lower thoracic levels (yellow arrows). Additionally, the branches at the periphery of the intercostal nerves of *Kdm6b*-cKO mice showed strikingly reduced complexity (red rectangles, as enlarged on the bottom panels). **a–c** 3–5 embryos for each genotype and stage were analyzed and the representative images were shown.

Olig2-lineage cells produce MNs, V3IN, and a subset of V2IN[39,40]. However, it did not reach the resolution to identify all known MN subtypes as it neither identified HMC as an independent cluster nor separated LMCm and LMCl cell types.

**Unsupervised clustering analysis of MN-lineage cells**. We reasoned that contrasting single cell transcriptomes only among the cells committed to the MN fate would define MN subtypes better and thus reveal unique transcriptome profiles of MN subtypes. As

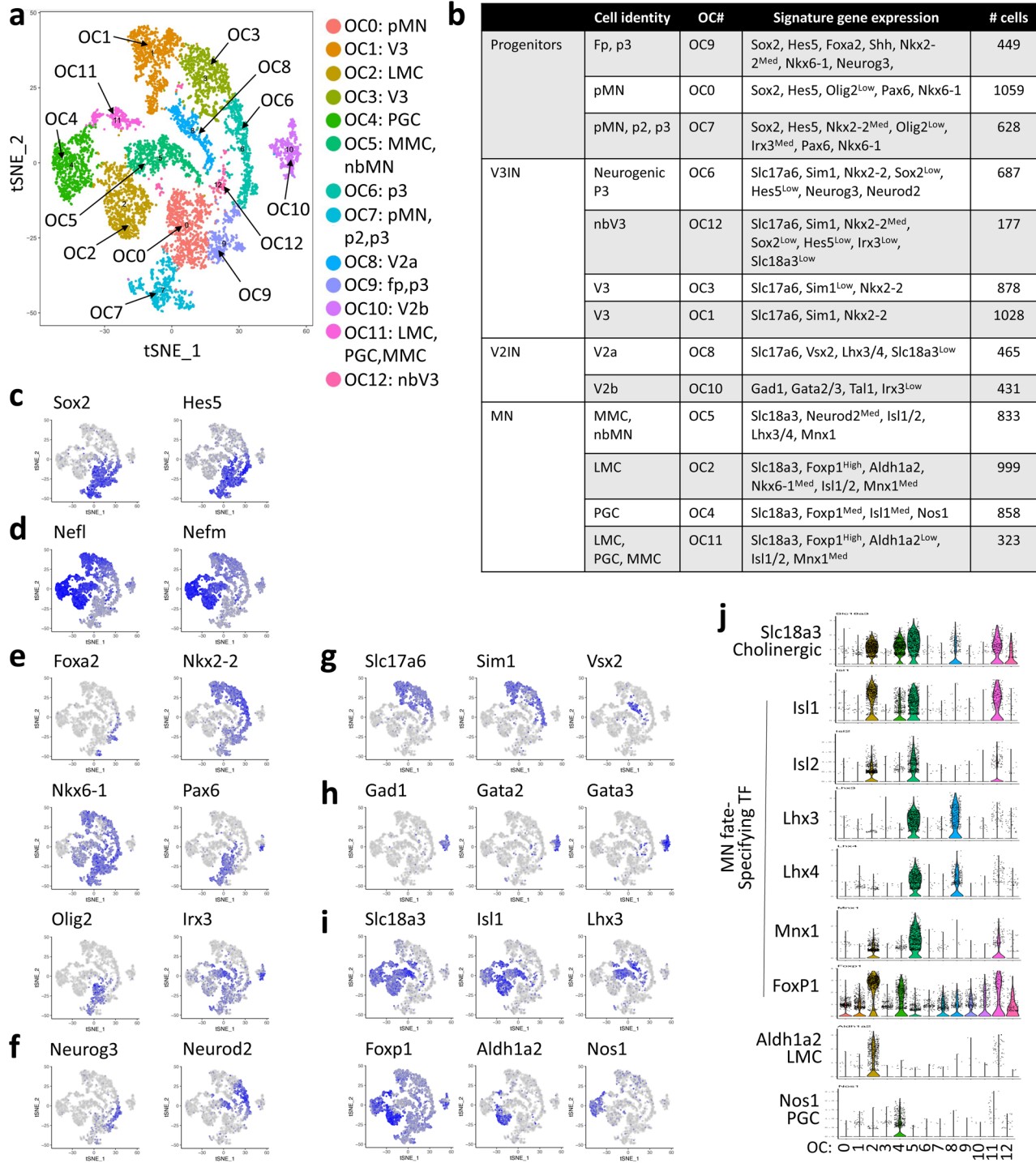

**Fig. 3 Single-cell transcriptome analyses of *Olig2-Cre* lineage cells in E12.5 spinal cord. a, b** Unsupervised clustering analysis of the single-cell transcriptomes of GFP$^+$ cells from E12.5 control spinal cord (*Kdm6b$^{f/+}$;Olig2-Cre; mTmG*). 13 Olig2-Cre lineage clusters, OC0 to OC12, were visualized using t-distributed stochastic neighbor embedding (tSNE). Cell types were assigned according to the expression of specific marker genes. Each dot represents a single cell and was color-coded based on the cluster identity. tSNE plots show the expression of the selected marker genes for progenitors **c**, neurons **d**, progenitor domains **e**, neurogenesis **f**, V3 and V2a interneurons **g**, V2b interneurons **h**, and MNs **i**. The expression levels of the genes were indicated by blue color intensity. **j** Violin plots for selected marker genes in OC0-OC12.

MNs are being produced and actively acquiring MN subtype identities in E12.5 spinal cord, Slc18a3$^+$ cells are expected to consist of newborn MNs (nbMN), MN subtypes, and a small number of progenitors that upregulated Slc18a3 but still retained progenitor gene transcripts. To test this idea, we selected Slc18a3$^+$ cells, a total of 3020 cells, performed unsupervised

clustering analysis[44], and identified 11 control clusters (CC), marked as CC0-10 (Fig. 4a, b, Supplementary Fig. 5a). Kdm6b was expressed similarly across all clusters (Supplementary Fig. 5b, c). CC0 and 1 were not mature neurons given a lower level of Nefl and Nefm (Fig. 4c). CC0 showed a low-level Slc18a3 and expressed Sox2, Hes5, and Cyclin D2 (Ccnd2) (Fig. 4c, e,

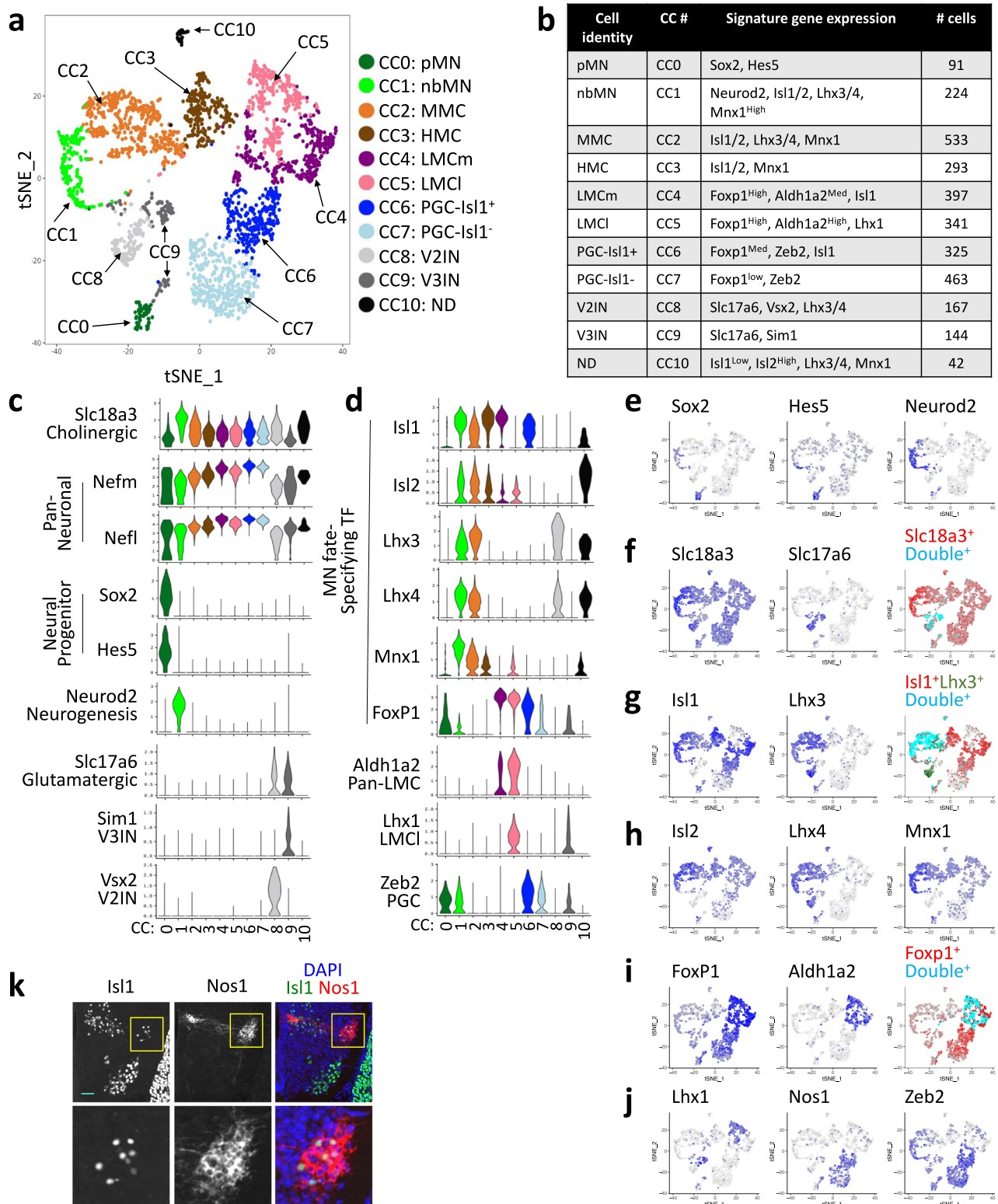

| Cell identity | CC # | Signature gene expression | # cells |
|---|---|---|---|
| pMN | CC0 | Sox2, Hes5 | 91 |
| nbMN | CC1 | Neurod2, Isl1/2, Lhx3/4, Mnx1High | 224 |
| MMC | CC2 | Isl1/2, Lhx3/4, Mnx1 | 533 |
| HMC | CC3 | Isl1/2, Mnx1 | 293 |
| LMCm | CC4 | Foxp1High, Aldh1a2Med, Isl1 | 397 |
| LMCl | CC5 | Foxp1High, Aldh1a2High, Lhx1 | 341 |
| PGC-Isl1+ | CC6 | Foxp1Med, Zeb2, Isl1 | 325 |
| PGC-Isl1- | CC7 | Foxp1low, Zeb2 | 463 |
| V2IN | CC8 | Slc17a6, Vsx2, Lhx3/4 | 167 |
| V3IN | CC9 | Slc17a6, Sim1 | 144 |
| ND | CC10 | Isl1Low, Isl2High, Lhx3/4, Mnx1 | 42 |

Supplementary Fig. 5b), suggesting that a small number of pMN cells express Slc18a3 while maintaining progenitor gene mRNAs. CC1 expressed a high level of proneural basic helix-loop-helix (bHLH) gene Neurod2 (Fig. 4c–e), suggesting CC1 is a nbMN cluster. A high level of neurofilament gene expression suggests that CC2-7 and CC10 are terminally differentiated MN subtypes (Fig. 4c). The combinatorial expression of Isl1/2, Lhx3/4, Mnx1 and Foxp1 enabled the annotation of their MN subtype identity

(Fig. 4d, f–i). CC2 expressed MMC markers Isl1/2, Lhx3/4, and Mnx1, and thus it was named MMC. CC3 showed a bona fide characteristic of HMC, which is the expression of Isl1/2 and Mnx1, but neither Lhx3/4 nor Foxp1. Both CC4 and CC5 exhibited the typical trait of LMC, such as Aldh1a2 expression and a highest level of Foxp1, but they differ in that CC4 is Isl1+ whereas CC5 is Lhx1+. Thus, CC4 and CC5 represent LMCm and LMCl, respectively. The two clusters CC6 and CC7 showed

**Fig. 4 Single-cell transcriptome analyses of embryonic MNs. a, b** Unsupervised clustering analysis of the single-cell transcriptomes of Slc18a3+ cells from E12.5 control spinal cord (*Kdm6b^f/+;Olig2-Cre;mTmG*). 11 control clusters, CC0 to CC10, were visualized using tSNE. Cell types were assigned according to the expression of specific marker genes. Each dot represents a single cell and was color-coded based on the cluster identity. **c, d** Violin plots for selected marker genes in CC0-CC10. **e–j** tSNE plots to show the expression of the indicated genes in each plot. The color intensity indicates the expression levels of the genes. The tSNE plots in **f, g,** and **i** reveal the co-expression pattern of the two genes. **f** CC8 (V2IN) and CC9 (V3IN) co-express Slc18a3 and Slc17a6 (cyan, Double+), while the remaining clusters express only Slc18a3 (red, Slc18a3+). **g** CC1 (nbMN) and CC2 (MMC) co-express Isl1 and Lhx3 (cyan, Double+), whereas CC3 (HMC), CC4 (LMCm), and CC6 (PGC-Isl1+) express Isl1 only (red, Isl1+). CC8 (V2IN) express Lhx3 only (green, Lhx3+). **i** CC4 (LMCm) and CC5 (LMCl) co-express Foxp1 and Aldh1a2 (cyan, Double+), whereas CC6 (PGC-Isl1+) and CC7 (PGC-Isl1−) express only Foxp1 (red, Foxp1+). **k** Immunohistochemical analyses with Isl1 and Nos1 antibodies in E13.5 wild-type spinal cords at thoracic levels. Nos1+ PGC cells were divided into Isl1+ and Isl1− subpopulations. Scale bar, 50 μm. 3–6 sections in 3 embryos were analyzed and the representative images were shown.

defining features for PGC, which include the expression of Foxp1 and Zeb2. They both also expressed another PGC marker Nos1, but the Nos1 level was higher in CC6 than CC7 (Fig. 4j, Supplementary Fig. 5b). Interestingly, Isl1 was expressed only in CC6, not in CC7, suggesting that PGC can be divided into Isl1+ (CC6) and Isl1− (CC7) subtypes, which we named PGC-Isl1+ and PGC-Isl1−. Consistent with the scRNAseq data, the immunostaining analyses revealed that Nos1+ PGC in the intermedio-lateral region of the spinal cord consisted of Isl1+ and Isl1− cells (Fig. 4k). The CC10 cluster of only 42 cells showed the mixed characteristics of both MMC and HMC, but did not match with established motor columnar properties, and thus we designated C10 as ND for not defined.

CC8 and 9 were positive for both Slc17a6 and Slc18a3 (Fig. 4c, f), indicating they possess hybrid neurotransmitter characteristics. CC8 also expressed a V2aIN marker Vsx2, whereas CC9 expressed a V3IN marker Sim1 (Fig. 4c). In this study, we named CC8 and CC9 as V2IN and V3IN clusters for the sake of clarity, although they are not fully differentiated interneurons, and their identity may be consolidated between ventral interneurons and MNs at later stages.

Together, our unsupervised clustering analyses identified distinct cell types in the spinal MN lineage based on single-cell transcriptomes. Importantly, our single-cell transcriptome data revealed the unexpected cellular heterogeneity of developing PGC MNs.

**Identification of markers and relationships among MN clusters.** To define comprehensive marker sets for MN cell types, we selected the genes that are specifically enriched in each cluster (FDR < 0.01, log2fold change > 0.4) (Fig. 5a, Supplementary Data 1). The pMN cluster had the most significant number of uniquely enriched genes (228 genes), followed by nbMN and V2IN clusters (138 genes each), indicating that the transcriptomes of these clusters are substantially different from those in other clusters. The marker analyses identified a list of interesting genes that are significantly enriched in each cluster. The pMN cluster expressed the neural progenitor gene Ttyh1, Notch pathway genes Hes5, Hey1, Notch1/2, Wnt genes Wnt 4/7b, and ID genes Id1/3 (Fig. 5b, Supplementary Fig. 6a). The pMN cluster also expressed glial precursor genes, such as Fabp7, Fgfbp3, Sox9, and Slc1a3 (aka Glast)[45] (Fig. 5b, Supplementary Fig. 6a), indicating that some pMN cells gain glial precursor characteristics by E12.5 when oligodendrocyte precursors begin to emerge from the ventricular zone[46]. The nbMN cluster expressed several TFs involved in neurogenesis, Neurod1/2/6, Nhlh1, Ebf2, and St18, which were downregulated in terminally differentiated MN clusters (Fig. 5b, Supplementary Fig. 6b).

The top three enriched genes in the MMC cluster were the known MMC markers, Lhx3, Mecom, and Pou3f1 (aka Scip), followed by Casz1 and Lhx4 (Fig. 5b, Supplementary Fig. 6c). The MMC marker list also included Fgf9, Rasgef1c, and Rxrg. To date, there is no good marker for HMC MNs. Thus, they have been

primarily identified by their cell body location in the ventro-lateral region of the thoracic spinal cord and the absence of Lhx3 expression among Isl1+Mnx1+ MNs. Our analyses revealed several genes that are specifically enriched in HMC (Fig. 5b, Supplementary Fig. 6d). The most specific HMC gene was Klf5, which was confirmed by the immunostaining analyses in the developing spinal cord (Fig. 5c). HMC cluster also showed a relatively specific enrichment of Plch1, Lix1, Isoc1, and Etv1.

LMCm and LMCl clusters had each cluster-specific markers while sharing a set of features, such as Foxp1, Aldh1a2, and Etv4 (Figs. 4d, 5b, Supplementary Fig. 6e, Supplementary Data 1). Sst was enriched in LMCm, whereas Lhx1os (for Lhx1 opposite strand) and Sema5a were enriched in LMCl. To better define the features distinguishing LMCm and LMCl, we directly compared the transcriptomes of LMCm and LMCl (Supplementary Data 2). The most differentially expressed genes between LMCm and LMCl were Isl1 and Sst for LMCm and Lhx1 and Lhx1os for LMCl. Our analyses also revealed several differentially expressed genes between the two LMC subtypes, including Nkx6-1 and Nkx6-2 enriched in LMCm and LMCl, respectively (Fig. 5b, Supplementary Data 2).

PGC-Isl1+ and PGC-Isl1− clusters shared over 30% of markers, including Kcnip4 and Kitl (Fig. 5b, Supplementary Fig. 6f, Supplementary Data 1), suggesting their close developmental relationship. They each had cluster-specific marker genes, such as Bmper for PGC-Isl1+ and Rgcc for PGC-Isl1−. The direct comparison of the two clusters uncovered that the prominent distinguishing features for PGC-Isl1+ and PGC-Isl1− clusters are Uts2b, Arhgap36, and Isl1 enrichment in PGC-Isl1+ and Vsnl1 and Vwc2 enrichment in PGC-Isl1– cluster (Supplementary Data 2). Notably, Lmo1 and Lrp1b were enriched in both Isl1-expressing LMC (LMCm) and PGC (PGC-Isl1+) clusters (Fig. 5b, Supplementary Fig. 6g).

V2IN and V3IN clusters had overlapping markers, such as Slc17a6, but they also had unique markers for each cluster (Fig. 4c, f, Supplementary Fig. 6h). The top three markers for the V2IN cluster were Shox2, Sox14, and Vsx2, the well-established V2aIN-specific genes. The expression of V2bIN genes, such as Gata2 and Gad1, was very low. Thus, V2IN cluster cells possess V2aIN-like identity, not V2bIN characteristics, consistent with the findings that MNs and V2aIN share developmental programs[4,13,21,47]. V3IN cluster was enriched in the expression of the established V3IN genes, Sim1, Nkx2-2, and Uncx.

To define how each cell cluster is related in an unbiased manner, we performed the hierarchical clustering analysis using the top markers (Fig. 5d). This analysis revealed the two large groups of clusters, the MN subtype group and the immature cell group. Within the MN subtype group, a pair of clusters were highly related to each other; LMCm and LMCl, PGC-Isl1+ and PGC-Isl1−, and interestingly MMC and HMC. Notably, the MMC, HMC, and LMC clusters consisting of somatic MNs are more related to each other than the PGC clusters representing visceral MNs. Within the immature cell group, the most related

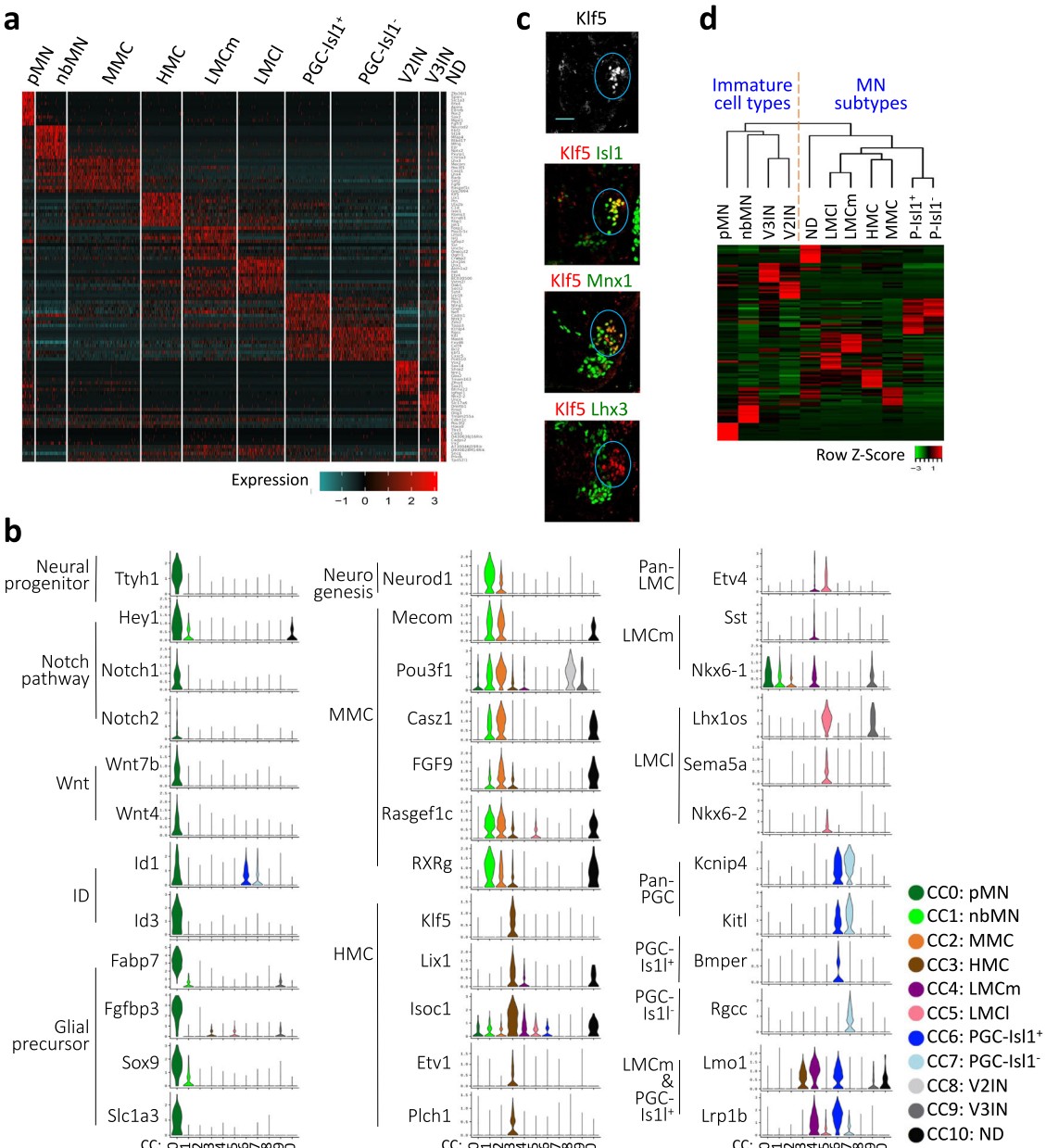

**Fig. 5 Relationships among MN lineage cells. a** Gene expression heatmap of individual cells (columns) for marker genes of each of CC0-CC10. Cells were grouped according to the Seurat classification in Fig. 4a, and their group identity was indicated on the top. Color code shows the normalized expression levels of the marker genes (rows). **b** Violin plots for the marker genes in CC0-CC10. **c** Immunohistochemical analyses in E12.5 wild-type thoracic spinal cords. Klf5 was specifically expressed in Isl1+Mnx1+Lhx3− HMC MNs, as marked by blue circles. Scale bars, 50 μm. 3–6 sections in 3 embryos were analyzed and the representative images were shown. **d** Hierarchical clustering analysis of CC0-CC10 cell types using their marker genes. The dendrogram shows the relationships of CC0-CC10 cell types. Immature cell types and mature MN subtypes were grouped separately. Among MN subtypes, somatic MNs of MMC, HMC, LMCm, and LMCl were more closely related to each other than visceral MNs of PGC-Isl1+ and PGC-Isl1− cell types.

were V2IN and V3IN clusters, which were connected with nbMN and then more distantly with pMN cluster, supporting the notion that the cells in these clusters were still undergoing neuronal differentiation and yet to acquire the terminally differentiated cellular phenotypes.

Together, our datasets revealed comprehensive transcriptome profiles for differentiating MN cell types and motor columns, thus providing invaluable resources for MN developmental studies.

**Unsupervised clustering analysis of Kdm6b-deficient MN-lineage cells.** In parallel with a control embryo, we isolated GFP+ cells from a littermate *Kdm6b*-cKO embryo and performed

scRNAseq analysis (Supplementary Fig. 7a). Employing the same single-cell transcriptome analysis strategy, we selected 3,278 Slc18a3+ cells in the *Kdm6b*-cKO sample and performed unsupervised clustering[44]. This identified 10 knockout clusters (KC), marked as KC0-9 (Fig. 6a–l, Supplementary Fig. 7b). The expression pattern of key markers readily identified pMN (KC9), nbMN (KC3), and V2/V3IN (KC5) clusters. As control cells, the two distinct PGC clusters, PGC-Isl1+ and PGC-Isl1−, were identified in *Kdm6b*-cKO cells. Notably, PGC-Isl1− cells vastly outnumbered PGC-Isl1+ cells in *Kdm6b*-cKO (PGC-Isl1−/PGC-Isl1+ ratio, 2.2), compared to control cells (PGC-Isl1−/PGC-Isl1+ ratio, 1.4), suggesting an overproduction of PGC-Isl1− cells in

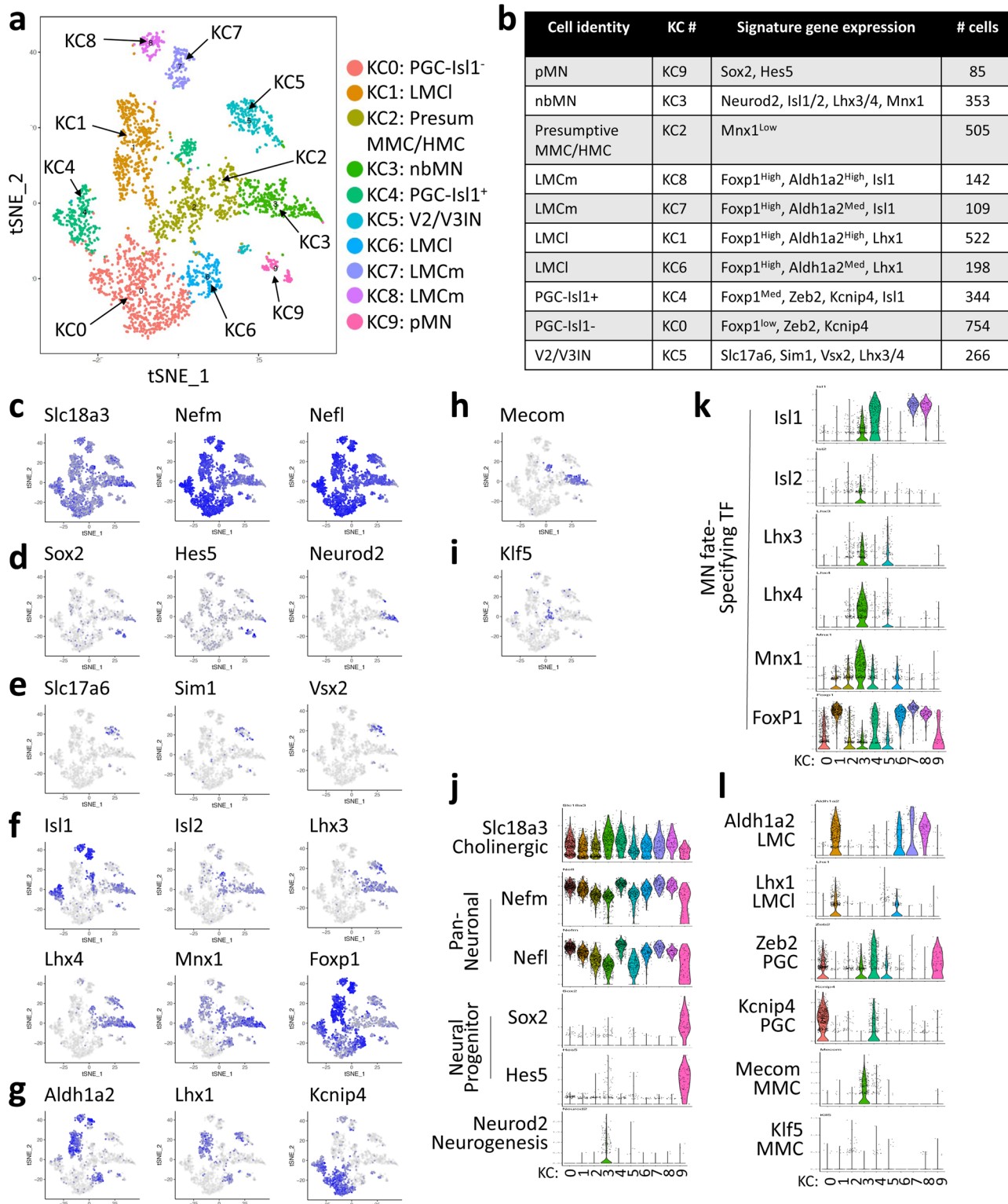

**Fig. 6 Single-cell transcriptome analyses of Kdm6b-deficient MN lineage cells. a, b** Unsupervised clustering analysis of the single-cell transcriptomes of Slc18a3+ from E12.5 *Kdm6b*-cKO spinal cord (*Kdm6b^f/f; Olig2-Cre;mTmG*). 10 knockout clusters (KC), KC0 to KC9, were visualized using tSNE. Cell types were assigned according to the expression of specific marker genes. Each dot represents a single cell and was color-coded based on the cluster identity. **c–i** tSNE plots to show the expression of the indicated genes in each plot. The expression levels of the genes were indicated by blue color intensity. **j–l** Violin plots for selected marker genes in KC0–KC9.

*Kdm6b*-null spinal cord. Interestingly, two clusters belong to either LMCm (KC7, 8) or LMCl (KC1, 6). The LMCl/LMCm ratio in *Kdm6b*-null MNs (2.9) was much greater than the LMCl/LMCm ratio in control MNs (0.85), suggesting that LMCl MNs

disproportionally increased in *Kdm6b*-cKO spinal cord. While the expression of Slc18a3, Nefl and Nefm suggests KC2 are differentiated MNs, the MN subtype identity of KC2 remained elusive as the expression of MN fate-specifying TFs, Isl1/2, Lhx3/4,

Mnx1, and Foxp1, was low in KC2 (Fig. 6f, j, k). The levels of Mecom and Klf5, markers for MMC and HMC, respectively, were also low in KC2 (Fig. 6h, i, l). Given that none of the KC clusters exhibited MMC or HMC properties, it is possible that MNs failing to acquire robust MMC or HMC identities became KC2 cells. Together, the unsupervised clustering analysis of *Kdm6b*-null MN lineage cells indicates that MMC and HMC identities were not properly established in the absence of Kdm6b and that LMCl and PGC-Isl1$^-$ MNs were overproduced relative to LMCm and PGC-Isl1$^+$ MNs in *Kdm6b*-cKO spinal cord.

Next, we projected both control and *Kdm6b*-cKO cells into the same UMAP feature space (Supplementary Fig. 8a, b). This analysis shows that control and *Kdm6b*-cKO cells in pMN, nbMN, V2IN, and V3IN clusters are relatively well intermingled, whereas MN subtype cells tend to show more segregated patterns between control and *Kdm6b*-cKO cells, suggesting that Kdm6 plays a role for MN subtype diversification. To examine how Kdm6b deficiency alters MN subtype fate choices, we analyzed the ratio of *Kdm6b*-cKO over control cells in matching MN subtype clusters after normalizing cell numbers between the two conditions (Supplementary Fig. 8c, d). If *Kdm6b*-deficient MNs favor particular MN subtypes over the other MN subtypes, this will result in skewed ratios of *Kdm6b*-cKO over control cells. Of note, we compared the combined cell number for MMC and HMC clusters (CC2,3) in the control sample to the cell number for the presumptive MMC/HMC cluster (KC2) in the *Kdm6b*-cKO sample. The *Kdm6b*-cKO/control cell ratio was highest in LMCl and PGC-Isl1$^-$, indicating that surplus LMCl and PGC-Isl1$^-$ MNs were formed in Kdm6b-deficient MN lineage. When all LMC clusters (LMCl and LMCm) and PGC clusters (PGC-Isl1$^+$ and PGC-Isl1$^-$) were combined as 'super-cluster'', *Kdm6b*-cKO/control cell ratio still remained larger than 1 for LMC and PGC, indicating the excess number of LMC and PGC were generated in *Kdm6b*-cKO mice. In contrast, the *Kdm6b*-cKO/control cell ratio for MMC/HMC and LMCm was smaller than 1, suggesting that the specification of MMC, HMC, and LMCm was defective in *Kdm6b*-cKO spinal cord.

**Unsupervised clustering of the merged single-cell transcriptomes of Kdm6b-deficient and control MNs identified MMC-Zfhx4$^+$ and LMCd MN subtypes.** As MNs are heterogeneous cell populations consisting of multiple MN subtypes, defining Kdm6b-dependent gene expression changes in each MN subtype, rather than in all MNs, is important to understand the target genes of Kdm6b in developing MNs. However, a lack of typical MMC and HMC clusters in the *Kdm6b*-cKO sample (Fig. 6) made it difficult to compare the gene expression profiles between *Kdm6b*-cKO and control in each MN subtype. If a subset of Kdm6b-deficient presumptive MMC and HMC MNs maintain overall transcriptome profiles resembling their control counterparts despite a low level of MN subtype-defining markers, they may cluster together with their counterparts in the control sample when analyzed as a merged dataset. Given that *Kdm6b*-cKO and control datasets show little batch-to-batch difference (Supplementary Fig. 7c, d), we compiled single-cell transcriptomes of *Kdm6b*-cKO and control MN-lineage cells (total 6298 cells) using Seurat canonical correlation analysis (CCA) and performed unsupervised clustering analysis[48]. This analysis identified 13 merged clusters (MC), named MC0-12 (Fig. 7a, b). The increased number of single cells in merged samples likely enabled the identification of two additional clusters that were not consolidated as separate clusters in the analysis of the control sample alone (Fig. 4a).

To annotate cell types of MC0-12, we determined a list of genes that were significantly enriched in each MC (FDR < 0.01, log2fold change > 0.4) (Supplementary Data 3) and compared this gene list to the markers that we identified in the control datasets (Supplementary Data 1). This identified the highest matching cell types for most MCs (Fig. 7c–e, Supplementary Fig. 9a–c). The four clusters, MC0, MC1, MC10, and MC11, expressed a low level of the neuronal gene Nefm, indicating they were less-differentiated cell types. They were annotated as pMN (MC0), nbMN (MC1), V2IN (MC10), and V3IN (MC11). The eight clusters expressed a high-level Nefm, indicating they are MN subtypes with fully differentiated neuronal characteristics. The matching score analysis readily identified six clusters; MMC (MC2), HMC (MC4), LMCm (MC5), LMCl (MC6), PGC-Isl1$^+$ (MC8) and PGC-Isl1$^-$ (MC9). The three clusters were not immediately defined by the matching score analysis. Among them, MC7 was most closely related to LMCm based on the matching score and high-level expression of Foxp1 and Isl1 (Fig. 7c, e). However, unlike LMCm, MC7 did not express pan-LMC marker Aldh1a2 and instead expressed Cpne4, Fign, and Pou3f1 (Fig. 7e, f, Supplementary Fig. 9d). As this transcriptome profile of MC7 fits nicely with the property of digit muscle-innervating LMC MNs[49], we named MC7 as LMCd for digit muscle-innervating LMC. Our single-cell transcriptome data identified LMCd-enriched marker genes, such as Ppp1r1c, Nexn, Gfra3, and Mgat4c (Fig. 7f, Supplementary Fig. 9d, Supplementary Data 3), which will be useful to label developing LMCd MNs.

MC3 expressed Mnx1 and Lhx3/4, like MMC, but showed only a low-level expression of Isl1 and Mecom, unlike typical MMC MNs (Fig. 7c, e, Supplementary Fig. 9c). Interestingly, MC3 uniquely expressed high-level of Zfhx4, Zfhx3, Sp8, and Uts2, whereas MMC (MC2) did not (Fig. 7f, Supplementary Fig. 9e), suggesting that MC3 is a distinct subtype of MMC. Consistently with the scRNAseq data, a subset of Lhx3$^+$Mnx1$^+$ MMC MNs in the ventro-medial region of the spinal cord expressed Zfhx4 and Zfhx3 (Fig. 7g). These results indicate that a subpopulation of MMC MNs have a distinct transcriptome profile, and they can be identified by Zfhx4 expression among Lhx3$^+$ MMC MNs. Thus, we named MC3 as MMC-Zfhx4$^+$.

As MC12 containing 194 cells showed mixed marker genes, we labeled C12 as ND (for not defined).

Together, the unsupervised clustering of the merged datasets uncovered the transcriptomes and markers for two additional MN subtypes, MMC-Zfhx4$^+$ and LMCd.

**scRNAseq dataset revealed that Kdm6b plays a crucial role in promoting MMC and HMC fates.** *Kdm6b*-cKO/control cell ratio among MN subtypes from the merged dataset was lowest in MMC and HMC, while being highest in LMCl and PGC-Isl1$^-$ (Fig. 8a), consistent with the results from separate sample analyses (Supplementary Fig. 8d). LMC super-clusters (LMCl, LMCm, and LMCd) and PGC super-clusters (PGC-Isl1$^+$ and PGC-Isl1$^-$) also exhibited >1 in *Kdm6b*-cKO/control cell ratio. Interestingly, within LMC or PGC super-cluster, Isl1-low/negative cell types, such as LMCl, and PGC-Isl1$^-$, increased, whereas Isl1-high/positive cell types, such as LMCm, LMCd, and PGC-Isl1$^+$, decreased in Kdm6b-deficient MN lineage.

The quantification of MN subtypes in spinal cord sections of *Kdm6b*-cKO and control mice validated that LMC and PGC MNs, particularly LMCl and PGC-Isl1$^-$, increased at the expense of MMC and HMC in *Kdm6b*-null MN lineage (Fig. 8b). Consistently, Mecom$^+$ MMC and *Klf5*$^+$ HMC markedly decreased, and *Nos1*$^+$ *Kcnip4*$^+$ PGC area was expanded in *Kdm6b*-cKO sections (Fig. 8c, Supplementary Fig. 10a, b). The total number of MNs was comparable between *Kdm6b*-cKO and control mice, and *Kdm6b*-cKO mice did not show a significant increase of apoptotic cells (Fig. 8b, Supplementary Fig. 10c),

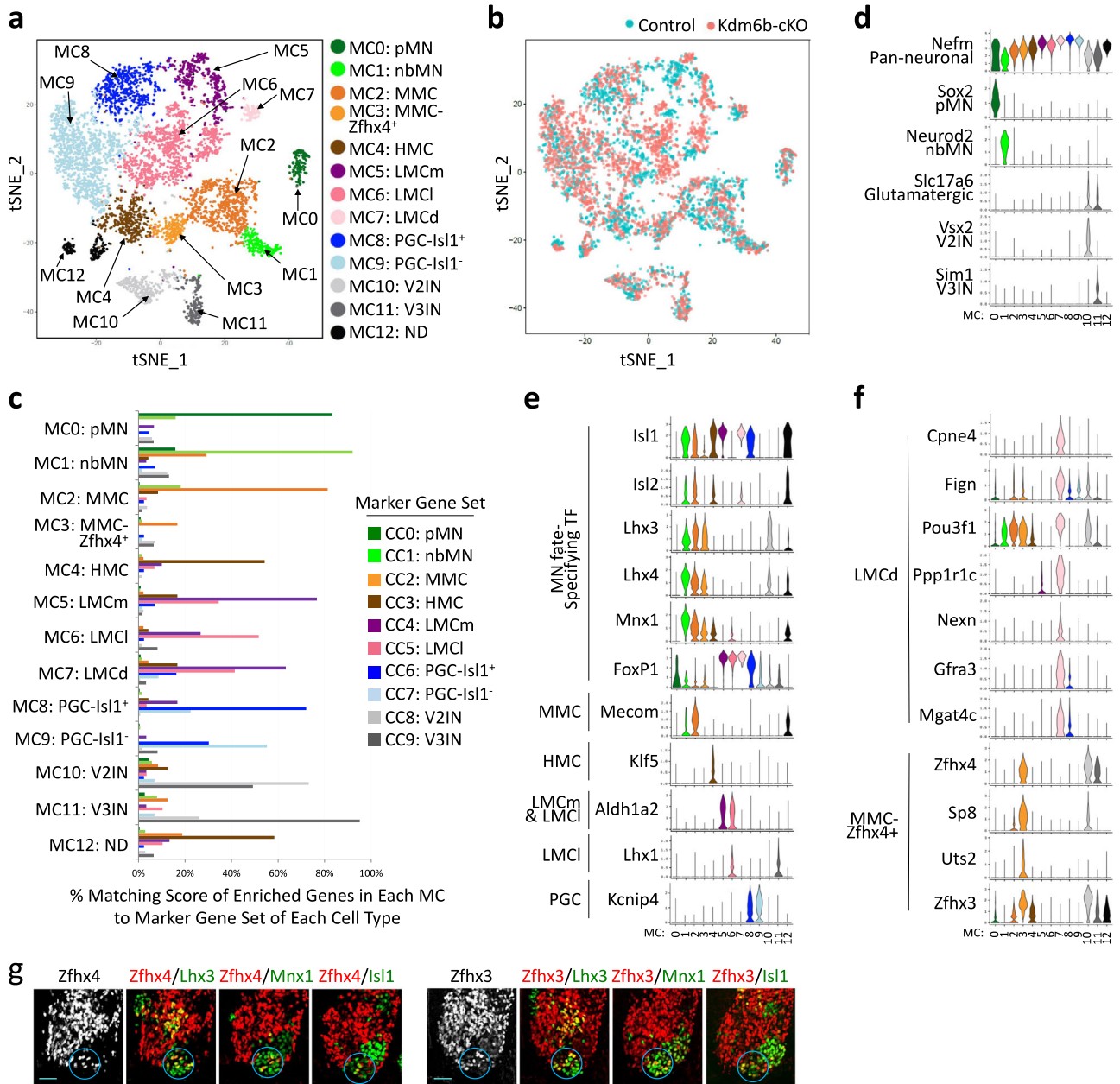

**Fig. 7 Analyses of the merged single-cell transcriptomes of *Kdm6b*-cKO and control MN-lineage cells. a** Unsupervised clustering analysis of the merged single-cell transcriptomes of Slc18a3+ cells from E12.5 control and *Kdm6b*-cKO spinal cords. 13 merged clusters, MC0 to MC12, were visualized using tSNE. Cell types were assigned according to the matching score in Fig. 7c and the expression of specific marker genes. Each dot, representing a single cell, was color-coded based on the cluster identity. **b** In this tSNE plot, each dot was color-coded based on the genotype. Control and *Kdm6b*-cKO cells were marked in cyan and orange, respectively. **c** Matching score analysis of each cluster-enriched gene revealed the cell-type identity of MC0-MC12. Pairwise comparison of the genes significantly enriched in each MC (FDR < 0.01, log2fold change > 0.4) to the marker gene sets in the control datasets CC0-CC9 in Fig. 5a. Matching scores show the percentage of each MC's enriched genes among the marker gene sets in CC0-CC9. This analysis identified the highest matching cell types for most MCs. **d–f** Violin plots for the selected marker genes in MC0-MC12. **g** Immunohistochemical analyses with Zfhx4, Zfhx3, Lhx3, Mnx1, and Isl1 antibodies in E12.5 wild-type thoracic spinal cords. MMC-Zfhx4+ MNs are located in the ventro-medial region of the spinal cord, like typical MMC MNs, and express high levels of Zfhx4 and Zfhx3 (circles). Scale bars, 50 μm. 3–6 sections in 3 embryos were analyzed and the representative images were shown.

suggesting that cell fate changes, not cell death, mainly contributed to the altered number of MN subtypes in *Kdm6b*-cKO mice.

Together, these results established that Kdm6b-deficient MNs favored taking LMC and PGC identity over MMC and HMC, highlighting a critical role of Kdm6b in promoting MMC and HMC fates while suppressing LMC and PGC fates during MN development. Further, our data suggest that Kdm6b plays an important role in LMC and PGC subtype diversification by augmenting Isl1+ LMC and PGC cell-type identity.

**Kdm6b is required for establishing MN subtype-specific gene expression patterns**. To gain insights into Kdm6b-mediated transcriptional regulation, we identified differentially expressed genes (DEG) between *Kdm6b*-cKO and control cells in each

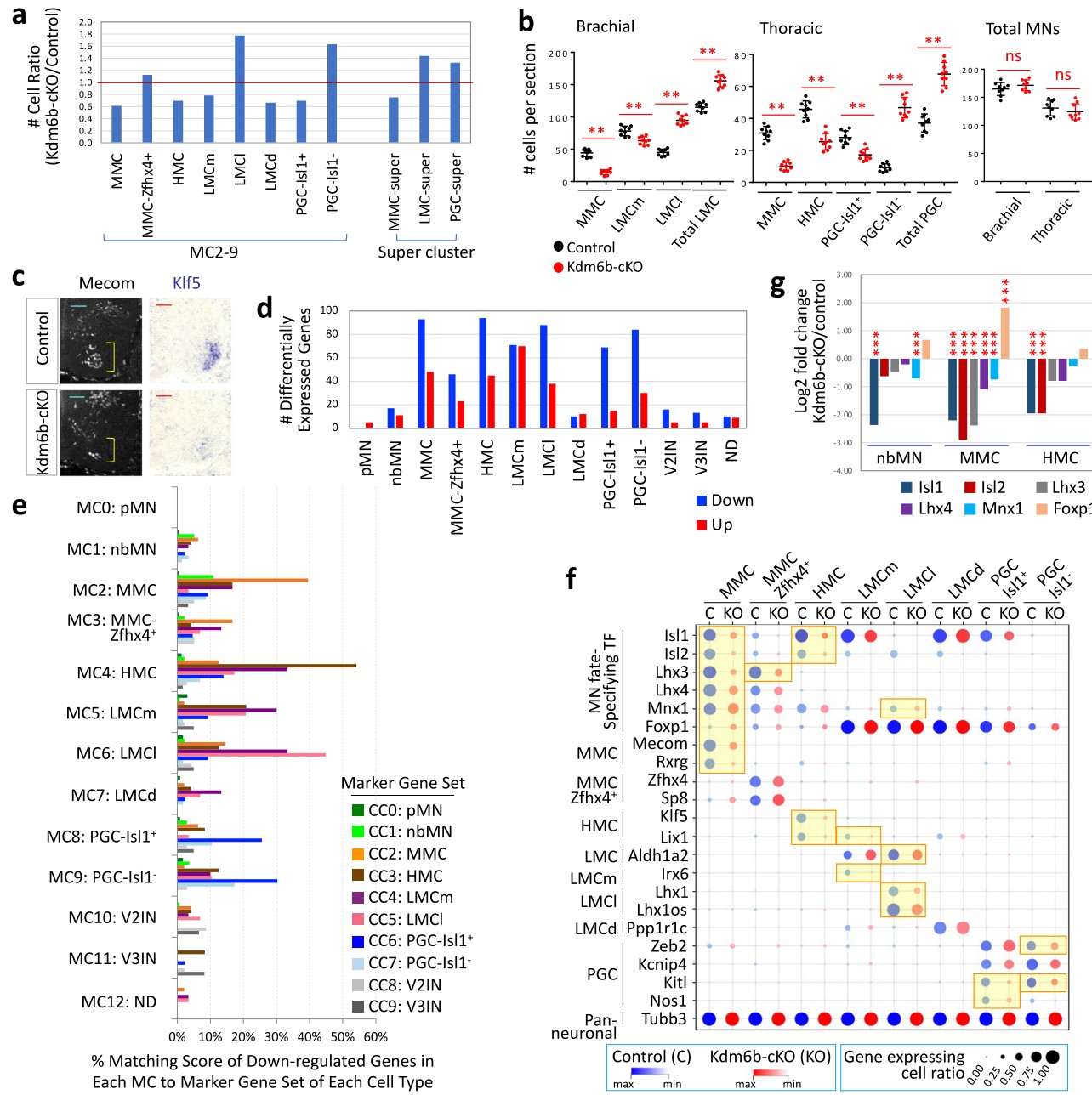

cluster (FDR < 0.01, absolute log2 fold change > 0.8, Supplementary Data 4). More genes were downregulated than upregulated in *Kdm6b*-cKO, relative to control cells, in most clusters (Fig. 8d), suggesting that Kdm6b is involved more in transcriptional activation than repression in MN lineage cells.

To investigate the characteristics of Kdm6b-dependent genes, we compared the up- and down-regulated genes in each cluster with the marker sets of MN-lineage cell types (Fig. 5a, Supplementary Data 1). The pairwise comparison revealed an intriguing correlation between the down-regulated genes in each cluster and the corresponding MN subtype-specific marker set (Fig. 8e). For instance, *Kdm6b*-cKO HMC down-regulated 54% of HMC marker genes, including Klf5 and Lix1, but did not down-regulate as much for other subtype marker genes. Likewise, Kdm6b-deficient MMC, LMCm, and LMCl down-regulated 40%, 30%, and 45% of the corresponding marker gene sets, respectively. The two Kdm6b-null PGC clusters also showed the down-regulation of 17–30% of their marker genes. Of note,

MMC-Zfhx4+ and LMCd clusters did not have the corresponding marker sets to compare, as they did not emerge as independent clusters from control data analysis. The expression of neuronal gene Tubb3 was comparable between *Kdm6b*-cKO and control cells across all MN subtypes (Fig. 8f). The down-regulated genes in less-differentiated Nefm-low clusters did not show as remarkable correlation with their corresponding marker genes as MN subtype clusters (Fig. 8e). Notably, the overlap between up-regulated genes for each cluster and its corresponding marker gene set was not more than 10% for any cluster (Supplementary Fig. 10d). Among all pairwise comparisons between up-regulated genes and all marker sets, the up-regulated genes in Kdm6b-deficient MMC and the LMCl marker set showed the highest overlap score of 17% (Supplementary Fig. 10d), suggesting that several LMCl markers, including Foxp1 (Fig. 8f), was aberrantly induced in Kdm6b-deficient MMC MNs.

Together, the significant downregulation of the marker genes for each *Kdm6b*-cKO MN subtype indicates that MN subtype

**Fig. 8 Kdm6b promotes MMC and HMC fates and suppresses LMC and PGC identities in part by regulating the expression levels of MN fate-specifying TFs. a** The ratio of *Kdm6b*-cKO cells over control cells in MC for each MN subtype or "super-cluster" marked as MMC-super (MMC and MMC-Zfhx4$^+$), LMC-super (LMCl, LMCm, and LMCd), or PGC-super (PGC-Isl1$^+$ and PGC-Isl1$^-$). **b** Quantification of MN numbers per 12 µm thick section of E12.5 spinal cords. MN subtypes were defined based on the following criteria; for the brachial spinal cord, MMC, Lhx3$^+$ cells in the ventral spinal cord; LMCm, Foxp1$^+$Isl1$^+$ cells in the ventro-lateral area; LMCl, Foxp$^+$Isl1$^-$ cells in the ventro-lateral area; total LMC, all Foxp1$^+$ cells in the ventro-lateral area; total MNs, MMC and LMC; for the thoracic spinal cord, MMC, Isl1$^+$Lhx3$^+$ cells in the ventral spinal cord; HMC, Isl1$^+$Lhx3$^-$ cells in the ventral spinal cord; PGC-Isl1$^+$; Foxp1$^+$Isl1$^+$ cells in the intermediate spinal cord; PGC-Isl1$^-$, Foxp1$^+$Isl1$^-$ cells in the intermediate spinal cord; total PGC, all Foxp1$^+$ cells in the intermediate spinal cord; total MNs, MMC, HMC, and PGC. Foxp1$^+$ LMC and PGC MNs at brachial and thoracic levels, respectively, were divided by the presence (LMCm, PGC-Isl1$^+$) and absence (LMCl, PGC-Isl1$^-$) of Isl1. Data are presented as mean values $+/-$ standard deviation. **$p < 0.01$; ns non-significant in the two-tailed Student's *t*-test. $n = 3$–6 mice per genotype and 9 slices per genotype. The exact *p*-values are as follows. Brachial, MMC, $5.78 \times 10^{-10}$; LMCm, $2.4 \times 10^{-4}$; LMCl, $1.24 \times 10^{-11}$; total LMC, $9.94 \times 10^{-9}$. Thoracic, MMC, $8.05 \times 10^{-10}$; HMC, $3.69 \times 10^{-7}$; PGC-Isl1$^+$, $2.65 \times 10^{-5}$; PGC-Isl1$^-$, $1.75 \times 10^{-11}$; total PGC, $2.67 \times 10^{-8}$. **c** Immunohistochemical analyses for Mecom and in situ hybridization analyses for Klf5 in E12.5 spinal cords. Mecom+ MMC (brackets) and Klf5+ HMC MNs decreased in *Kdm6b*-cKO mice. Scale bars, 50 µm. 3–6 sections in 3 embryos were analyzed and the representative images were shown. **d** The number of differentially expressed genes (DEG) between *Kdm6b*-cKO and control cells in MC0-MC12 clusters (FDR < 0.01, absolute log2 fold change > 0.8). Up- and down-regulated genes were marked by red and blue, respectively. **e** Pairwise comparison of the genes significantly down-regulated in each MC to the marker gene sets in CC0-CC9. Matching scores show the percentage of each MC's down-regulated genes among the marker gene sets in CC0-CC9. A substantial portion of the specific marker genes for each MN subtype was down-regulated in *Kdm6b*-cKO cells. **f** Expression levels of marker genes in control (C, blue dots) and *Kdm6b*-cKO (KO, red dots) across MN subtype clusters. The size of dots indicates the ratio of the gene expressing cells among all control or *Kdm6b*-cKO cells in each cluster. The color intensity denotes the relative gene expression levels in control (blue) or *Kdm6b*-cKO (red). DEGs (FDR < 0.01, absolute log2 fold change > 0.8) are highlighted in yellow. **g** Relative gene expression levels of MN fate-specifying TFs between *Kdm6b*-cKO and control cells in nbMN, MMC, and HMC clusters. In *Kdm6b*-cKO, Isl1/2, Lhx3/4, and Mnx1 expression decreased, while Foxp1 expression was increased. ***$p.adj < 0.001$. The *p*-value was calculated using the *WilcoxDETest* function in Seurat package, in which a two-sided alternative hypothesis testing is performed. Multiple testing correction was carried out using the *p.adjust* function with –*fdr* option in R package. The exact *p*-values are as follows. nbMN, Isl1, 1.58E-23; Mnx1, 5.29E-05. MMC, Isl1, 7.53E-47; Isl2, 6.59E-60; Lhx3, 1.48E-68; Lhx4, 3.32E-24; Mnx1, 4.23E-09; Foxp1, 1.53E-08. HMC, Isl1, 5.60E-28; Isl2, 2.82E-13.

identities were eroded in the absence of Kdm6b. These results strongly support that Kdm6b is important to consolidate MN subtype-specific gene expression patterns.

**The expression pattern of MN fate-specifying TFs alters in *Kdm6b*-cKO.** *Kdm6b*-cKO cells showed a significant reduction of Isl1/2, Lhx3/4, and Mnx1 and an aberrant induction of Foxp1 in MMC, and similar trends were observed in nbMN and HMC (Fig. 8f, g). The pseudo-bulk analysis, which took all Slc18a3$^+$ cells into account[50], also revealed that Isl1/2, Lhx3/4, and Mnx1 genes were markedly down-regulated, whereas Foxp1 level was significantly elevated in *Kdm6b*-null MN lineage (Supplementary Fig. 11a). Similarly, the percentage of Isl1/2$^+$, Lhx3/4$^+$, and Mnx1$^+$ cells among all Slc18a3+ cells reduced, but that of Foxp1$^+$ cells increased in Kdm6b-deficient MN lineage (Supplementary Fig. 11b, c). Together, Kdm6b inactivation in developing MNs led to a decrease in Isl1/2, Lhx3/4, and Mnx1 levels and a concomitant increase in the Foxp1 level.

**Lhx3 suppresses Foxp1 expression in developing MNs.** Given that Foxp1 inhibits Lhx3 expression during MN development[25], the aberrant Foxp1 induction in Kdm6b-deficient MNs might have led to the suppression of Lhx3, which would result in the reduction of Isl1/2 and Mnx1. However, if Lhx3 inhibits the expression of Foxp1 in MNs, Lhx3 reduction in Kdm6b-null MNs could be the initial event that triggered Foxp1 elevation. To distinguish between these two possibilities and to ask if Lhx3 suppresses Foxp1 expression in developing MNs, we analyzed *Mnx1p::Lhx3* transgenic mice, in which Lhx3 expression is induced in all MNs by the *Mnx1* promoter[23]. The misexpression of Lhx3 in all MNs drastically reduced Foxp1$^+$ cells in both brachial and thoracic levels of the spinal cord (Fig. 9a), in line with the previous chick electroporation data[24,25]. Moreover, Lhx3 misexpression suppressed *Aldh1a2$^+$* LMC and Nos1$^+$ PGC MNs in the brachial and thoracic levels, respectively (Fig. 9a). These results demonstrate that Lhx3 is capable of suppressing Foxp1 expression and the antagonistic relationship between Lhx3 and

Foxp1 is important not only for setting up their complementary expression domains among MN subtypes but also for establishing MN subtype identities.

**A later inactivation of Kdm6b led to a loss of Lhx3$^+$ MNs without a gain of Foxp1$^+$ MNs.** We next asked if it is Lhx3 downregulation or Foxp1 upregulation that serves as an initial triggering event for the reciprocal expression changes of Foxp1 or Lhx3 in *Kdm6b*-null MNs. We reasoned that, if we inactivate Kdm6b after MN subtype fate is more committed, that will reveal a direct outcome of the loss of Kdm6b with fewer secondary effects on Lhx3 or Foxp1 expression changes. Thus, we deleted *Kdm6b* using *Isl1-Cre*, which expresses the Cre recombinase at a later time point in the MN lineage than *Olig2-Cre*[51]. Correspondingly, Kdm6b protein was still detected at E11.5 and significantly downregulated by E12.5 in *Kdm6b*-cKO;*Isl1-Cre* (*Kdm6bf/f;Isl1-Cre*) mice (Fig. 9b). *Kdm6b*-cKO; *Isl1-Cre* mice exhibited a significant reduction of Lhx3$^+$ MNs, but did not change Foxp1$^+$ MNs in either brachial or thoracic levels (Fig. 9c, d), indicating that a later removal of Kdm6b led to the reduction of Lhx3$^+$ MMC without a significant increase in Foxp1$^+$ LMC or PGC. Consistently, neither *Aldh1a2$^+$* LMC nor *Kcnip4$^+$* PGC area expanded in *Kdm6b*-cKO;*Isl1-Cre* mice (Fig. 9e, f). Mnx1$^+$ MNs, but neither Isl1$^+$ MNs nor *Klf5$^+$* HMC area, reduced in *Kdm6b*-cKO;*Isl1-Cre* embryos (Fig. 9e–g). These results support the notion that the primary action of Kdm6b in developing MNs is to keep Lhx3 expression and activity rather than suppressing Foxp1 expression.

**Kdm6b acts as a coactivator of Isl1-Lhx3.** The Isl1-Lhx3 complex plays a crucial role in inducing and maintaining Lhx3 expression in developing MNs. Isl1-Lhx3 directly binds Isl1/2 and Lhx3/4 genes themselves and the Mnx1 gene and activates their expression in MNs[12,18,22]. Thus, a compromised transcriptional activity of Isl1-Lhx3 could lead to the downregulation of Isl1/2, Lhx3/4, and Mnx1, as seen in Kdm6b-deficient MNs. The Isl1-Lhx3-directed transcriptional activation is also critical for MN

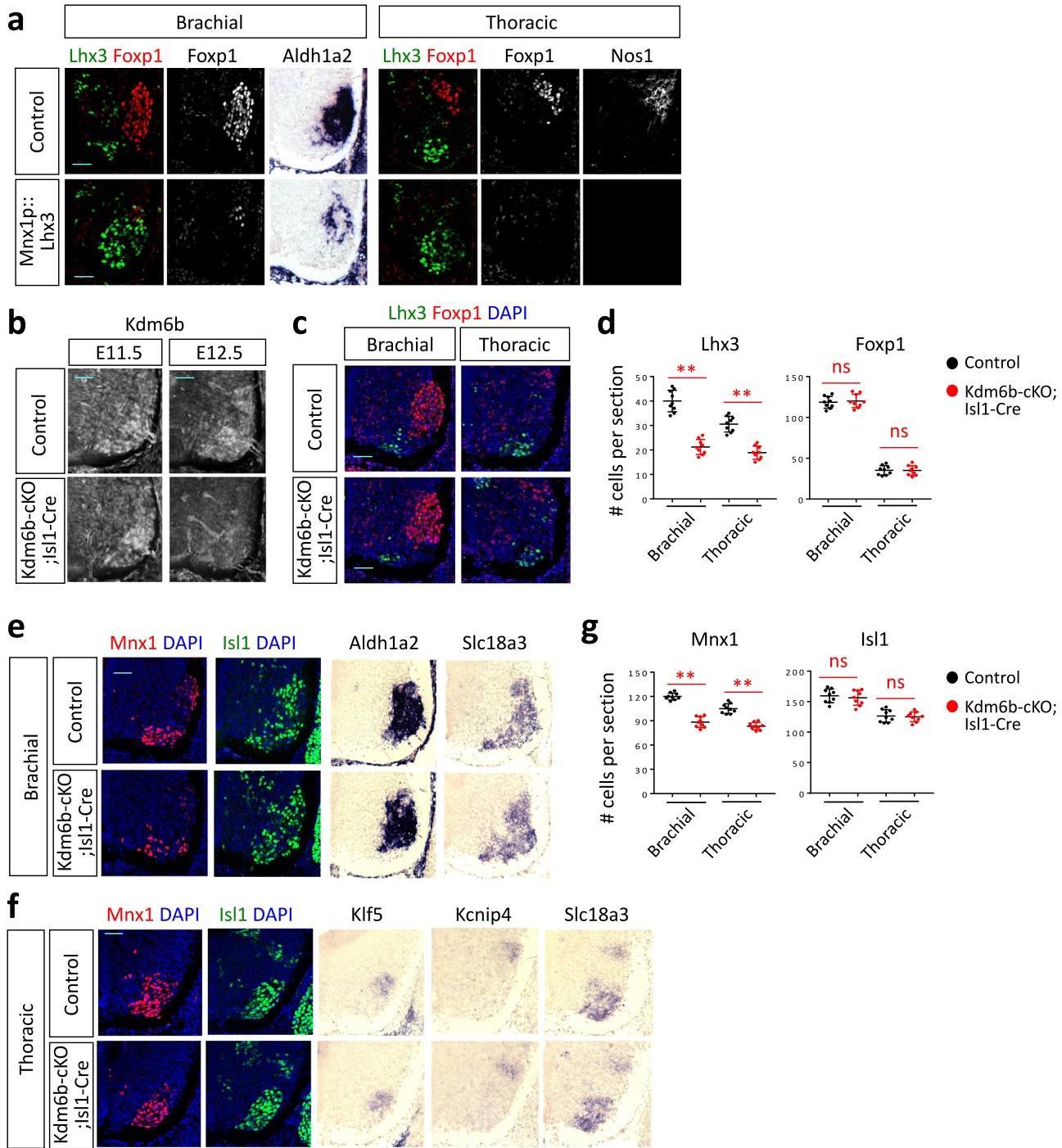

**Fig. 9 A later inactivation of Kdm6b resulted in reduced Lhx3+ MNs, but not increased Foxp1+ MNs. a** Immunohistochemical analyses for Lhx3, FoxP1, and Nos1 and in situ hybridization analyses for Aldh1a2 in E12.5 spinal cords. Foxp1+ MNs, Aldh1a2+ LMC, and Nos1+ PGC were markedly reduced in *Mnx1p::Lhx3* transgenic mice, in which Lhx3 expression was expanded to all MNs. Scale bars, 50 μm. **b** Immunohistochemical analysis with Kdm6b antibody. Kdm6b protein was detected at E11.5 and substantially removed by E12.5 in *Kdm6b*-cKO;*Isl1-Cre* mice. Scale bars, 50 μm. **c-g** Immunohistochemical analyses for Lhx3, FoxP1, Mnx1, and Isl1, in situ hybridization analyses for Aldh1a2, Slc18a3, Klf5, and Kcnip4 in E13.5 spinal cords, and the quantification of specific marker-labeled MNs per 12 μm thick section. In *Kdm6b*-cKO;*Isl1-Cre* mice, Lhx3+ and Mnx1+ MNs were significantly reduced, but neither Foxp1+ nor Isl1+ MNs changed significantly. Scale bars, 50 μm. The error bars represent the standard deviation of the mean. **p < 0.01; ns non-significant in the two-tailed Student's *t*-test. n = 3–6 mice per genotype and 9 slices per genotype. Source data are provided as a Source Data file. The exact *p*-values are as follows. Lhx3, brachial, $1.44 \times 10^{-8}$; thoracic, $8.05 \times 10^{-10}$. Foxp1, brachial, 0.68; thoracic, 0.94. Mnx1, brachial, $1.79 \times 10^{-9}$; thoracic, $2.36 \times 10^{-7}$. Isl1, brachial, 0.55; thoracic, 0.76.

fate specification and MMC generation[4,13,14,18–20]. Combined with Kdm6b's action as a coactivator for TFs in various biological processes, our analyses of *Kdm6b*-null MNs strongly suggest that Kdm6b acts as a coactivator of the Isl1-Lhx3 complex, which directly promotes MMC fate and indirectly suppresses Foxp1 expression and LMC and PGC identities (Fig. 10e). To further test this idea, we examined if Kdm6b interacts with Isl1 or Lhx3 using co-immunoprecipitation assays (Fig. 10a). The N-terminal

region of Kdm6b, but not its C-terminal region, is associated with Lhx3. Kdm6b co-immunoprecipitated both Isl1 and Lhx3 when Isl1 is co-expressed with Lhx3, but it failed to associate with Isl1 alone in the absence of Lhx3 (Fig. 10a, b). These results indicate that Kdm6b associates with the Isl1-Lhx3 complex via its interaction with Lhx3.

Next, to ask if Kdm6b influences the transcriptional activity of Isl1-Lhx3, we monitored Isl1-Lhx3-directed transcription using the luciferase reporter linked to the Isl1-Lhx3 binding MN-hexamer-response elements (HxRE)[13] (Fig. 10c). Isl1-Lhx3 markedly activated the HxRE reporter. Kdm6b further augmented the Isl1-Lhx3-directed transcriptional activation in a dose-dependent manner, while it had little effect on HxRE in the absence of Isl1-Lhx3, suggesting that Kdm6b can activate HxRE-containing MN genes by associating with Isl1-Lhx3. In contrast, Kdm6a failed to increase the transcriptional activity of Isl1-Lhx3.

To test if Kdm6b is recruited to Isl1-Lhx3 target genomic loci, we performed chromatin immunoprecipitation (ChIP) assays in E12.5 spinal cords. Kdm6b bound to the representative Isl1-Lhx3 target loci in Mnx1, Isl1, and Lhx4 genes, indicating that Kdm6b is recruited to Isl1-Lhx3-bound genomic regions in vivo (Fig. 10d).

Together, these results indicate that Kdm6b is recruited to and activates Isl1-Lhx3 target MN genes by associating with the Isl1-Lhx3 complex in developing MNs (Fig. 10e). Thus, by acting as a critical coactivator of Isl1-Lhx3, Kdm6b promotes MMC fate and suppresses LMC and PGC identities.

## Discussion

One of the most fundamental issues in neuroscience is to understand how remarkably divergent neuronal cell types are generated in a spatially and temporally regulated manner during development. To achieve such a great level of neuronal cell diversity and connectivity required for a functional nervous system, the gene transcription should be dynamically but precisely regulated. The combinatorial actions of neuronal cell-type-specific TFs serve as a critical mechanism to generate neuronal cell diversity. However, the role of transcriptional coregulators, which are indirectly recruited to DNA via interaction with TFs and enable the transcriptional regulation by those TFs, in neuronal diversification has remained unclear.

We addressed this cardinal issue by investigating the role of H3K27DMs in the development of spinal MNs. The cell fate specification and diversification processes of spinal MNs have been relatively well studied, providing an ideal platform to test the role of transcriptional coregulators. By employing the ensemble of mouse genetics and exploratory single-cell transcriptomes analyses, we found that Kdm6b plays essential functions in creating neuronal diversity in the developing MNs and validated the key findings with immunohistochemical analyses. Specifically, Kdm6b plays an instructive role in determining the fate of diversifying columnar MNs by promoting MMC and HMC fates while inhibiting LMC and PGC identities. Further, our study demonstrates that Kdm6b acquires specificity in choosing particular cell fates by collaborating with the transactivating TF complex Isl1-Lhx3. Importantly, our single-cell transcriptome profiling, followed by unsupervised clustering and hierarchical analyses, unraveled the previously unrecognized heterogeneity of motor columns and their relationship and MN subtype-specific molecular features. These studies also present crucial insights into the transcriptional regulatory mechanisms by which distinct MN subtypes emerge from newborn MNs. Together, our study provides invaluable information in deciphering the gene regulatory network orchestrating neuronal cell type diversification as well as understanding the nervous system organization controlling locomotion.

In the developing CNS, cells undergo dynamic changes of gene expression as the progenitor cells exit the cell cycle, differentiate to neurons, acquire specific neuronal type identity and further differentiate into neuronal subtypes. The precise control of gene expression throughout this process deploys transcriptionally activating and repressing complexes[52]. H3K27me3 is a well-established histone mark for a repressive but dynamically poised chromatin state during development[53]. Thus, H3K27DMs can activate the transcriptionally poised developmental genes by removing the H3K27me3 mark and modulating chromatin accessibility. Indeed, Kdm6b and Kdm6a are involved in diverse organ development, but rather than acting redundantly, they often play distinct and even opposing roles[29]. For instance, Kdm6a occupies Hox gene loci and regulates Hox genes and Hox-mediated posterior development, whereas Kdm6b has little effect on Hox gene expression[54]. Supporting this notion, our studies highlight distinct biological activities of Kdm6b and Kdm6a in neurodevelopment. While Kdm6b was essential for the balanced production of MN subtypes, Kdm6a was largely dispensable for MN development. At least two factors are likely to contribute to these disparate functions of the two H3K27DMs. First, the Kdm6b level goes up, and the Kdm6a level goes down during neuronal differentiation in the developing spinal cord, and thus Kdm6b is more enriched in differentiating MNs than Kdm6a. Second, Kdm6b cooperates with the MN fate-specifying Isl1-Lhx3 complex, but Kdm6a does not.

Intriguingly, rather than promoting the overall production of all MN subtypes, Kdm6b clearly favors the generation of MMC and HMC and suppresses the formation of LMC and PGC. Our findings indicate that Kdm6b plays an instructive role in determining MN subtype identity, not a general role to support all transcriptional activation. This specificity of Kdm6b action in MN subtype generation can be attributed to our finding that Kdm6b acts as a critical coactivator of Isl1-Lhx3 (Fig. 10e). This model predicts that when Kdm6b is inactivated, the transcriptional activity of Isl1-Lhx3 is compromised, resulting in a decrease of MMC MNs and an increase of LMC and PGC MNs. Both the immunohistochemistry analyses with established markers and the unbiased single-cell transcriptome analyses supported this prediction. The unbalanced production of MN subtypes was also reflected in the accordingly altered motor axon projection patterns in the periphery, such as thinner MMC nerves and thicker PGC and LMC axon bundles in Kdm6b-cKO embryos.

How does a reduced transcriptional activity of Isl1-Lhx3 in Kdm6b-deficient MNs profoundly alter MN differentiation? Previous studies have elucidated the intricate transcriptional regulatory network that orchestrates MN fate specification and subtype diversification. In this network, the TFs Isl1/2, Lhx3/4, Mnx1, and Foxp1 are the main players that show complex auto-regulatory, feed-forward, and cross-repressive regulatory interactions. Isl1 and Lhx3 are induced as MNs emerge from the progenitors and form the transactivating Isl1-Lhx3 complex (aka MN-hexamer). Isl1-Lhx3 directly binds and activates a broad range of MN genes, including a battery of cholinergic neurotransmitter genes, as well as Isl1/2, Lhx3/4, and Mnx1[16,18–20,22]. Lhx3 and Lhx4 play redundant roles in MNs[7]. Isl2 is upregulated later than Isl1 in differentiating MNs, and Isl2 can replace Isl1 in the Isl1-Lhx3 complex[55,56]. Mnx1, induced by Isl1-Lhx3, reinforces MN fate by supporting Isl1 and Lhx3 expression and suppressing V2IN fate, in part by directly binding to the Vsx2 gene[4,9,12,13,55]. As newborn MNs further differentiate to MN subtypes, Lhx3 is down-regulated in all MNs except MMC, while Isl1 expression persists in broader MN subtypes. Thus, the Isl1-Lhx3 complex is maintained in MMC and activates genes involved in terminal differentiation and axon pathfinding[14,18–20].

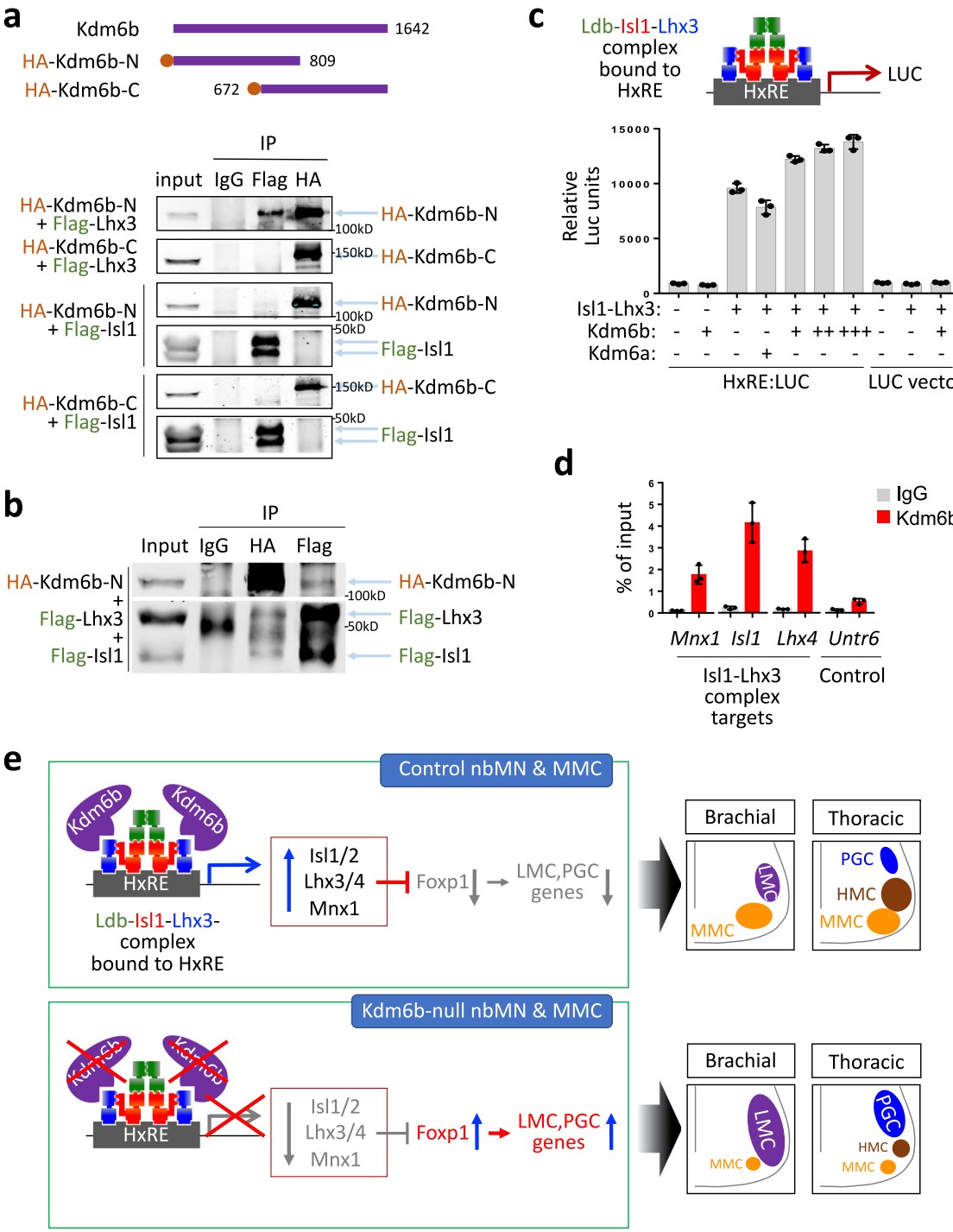

**Fig. 10 Kdm6b acts as a coactivator of Isl1-Lhx3.** Kdm6b is associated with Lhx3 (**a**) or Isl1-Lhx3 (**b**) in co-immunoprecipitation assays in HEK293 cells. HEK293 cells were transfected with the constructs indicated on the left of the immunoblotting results. Antibodies used for immunoprecipitation were noted on the top. HA or Flag antibodies were used for immunoblotting assays, as shown on the right. The assays were performed three times in each condition and the representative results were shown. **c** Luciferase reporter assays in HEK293 cells transfected with *HxRE:LUC* or control *LUC* vector with (+) or without (−) Isl1-Lhx3, Kdm6b or Kdm6a, as indicated on the graph. Kdm6b enhanced Isl1-Lhx3-mediated transcriptional activity in a dose-dependent manner. $n = 3$ biologically independent samples. Error bars represent the standard deviation of the mean. **d** ChIP assays in E12.5 spinal cords with IgG or Kdm6b antibody. Kdm6b was recruited to the Isl1-Lhx3 binding loci in Mnx1, Isl1, and Lhx4 genes. The Untr6 was used as a negative control region. $n = 3$ biologically independent samples. Error bars represent the standard deviation of the mean. **e** The model depicts Kdm6b action in the development of spinal MNs. Kdm6b is recruited to the Isl1-Lhx3-binding hexamer response element (*HxRE*) by associating with Isl1-Lhx3 complex and activates the Isl1-Lhx3 target genes, including Isl1/2, Lhx3/4, and Mnx1 genes. This transcriptional activation promotes nbMN and MMC fates, in part by further enhancing Isl1-Lhx3 activity. Moreover, as Lhx3 inhibits Foxp1 expression, Kdm6b-mediated transcriptional activation indirectly suppresses LMC and PGC fates. Thus, Kdm6b controls the balanced production of MN subtypes.

In HMC, most of the Isl1-Lhx3 target genes still maintain their expression through dynamic enhancers bound by Isl1-Onecut1 during the conversion of MMC to HMC[57]. This suggests one of the mechanisms by which a loss of Isl1-Lhx3 activity in Kdm6b-deficient MNs impacts the generation of HMC MNs and their gene expression patterns. In LMC and PGC, Foxp1 plays vital roles in establishing their MN subtype identities, along with Hox genes and Pbx1/3[24,25,58,59]. The early inactivation of Kdm6b in the MN lineage by *Olig2-Cre* resulted in reduced Lhx3 and reciprocally increased Foxp1, whereas the later elimination of Kdm6b by Isl1-Cre decreased Lhx3 but did not significantly change Foxp1 levels. Our findings, combined with previous studies[24,25], strongly support the notion that the cross-repressive relationship between Lhx3 and Foxp1 plays a vital role in MN subtype diversification. This antagonism between Lhx3 and Foxp1, along with Isl1-Lhx3-driven MN gene activation, provides crucial insights into how the diminished Isl1-Lhx3 activity in *Kdm6b*-null MNs led to the overproduction of LMC and PGC at the expense of MMC and HMC. Our data also suggest that MNs lose the plasticity in switching among different MN subtypes as they acquire a terminally differentiated state. In addition, our study shows that the continuous action of Kdm6b is needed for maintaining MMC identity.

The scRNAseq provides a powerful tool to understand cell-type diversity. Taking this advantage further, we employed scRNAseq analysis to analyze the unbalanced production of MN subtypes and to comprehensively determine dysregulated genes in each MN subtype in MN-specific *Kdm6b*-cKO mice. Our analysis pipeline was highly effective in dissecting phenotypes in heterogeneous cell types undergoing dynamic gene expression changes and cell fate shifts over the course of organogenesis and thus would be applicable to a wide range of studies.

Besides characterizing phenotypes of Kdm6b-deficient MNs, our study identified additional MN subtypes and comprehensive marker sets for MN subtypes and provided crucial insights into MN diversification. The recent scRNAseq study revealed multiple subtypes in interneurons and MNs in the developing spinal cord[60]. However, this study did not uncover additional MN subtypes, likely because when all spinal cord cells were combined for clustering analysis, relatively smaller differences among MN subtypes were not readily recognized to form distinct clusters. In this study, by focusing our analysis on Slc18a3$^+$ MN lineage cells, we increased clustering resolutions and were able to identify previously unrecognized subpopulations of developing MNs based on transcriptome profiles.

Many interesting themes emerged from our single-cell transcriptome study. First, our study revealed that the developing PGC MNs could be divided into at least two groups, PGC-Isl1$^+$ and PGC-Isl1$^-$, based on gene expression profiles. The heterogeneity of PGC MNs in the mature spinal cord has been proposed[61,62], but their developmental origin remained unclear. Future studies should be directed at understanding the relationship between the embryonic PGC subtypes and the adult PGC subtypes in molecular features, axonal targets, and electrophysiological membrane properties. Second, the similarity between the LMC subdivision and the PGC subdivision based on Isl1 expression is notable. As LMC MNs mature, Isl1-negative LMCl MNs emerge from Isl1-positive LMCm MNs. LMCl expresses Lhx1 instead of Isl1, and the mutual cross-repression between Isl1 and Lhx1 is important to establish distinct identities between LMCm and LMCl, including their axonal projection into ventral and dorsal limb musculature, respectively[63,64]. Future studies may identify the TFs that positively define the identity of PGC-Isl1$^-$ MNs, like Lhx1 in LMCl. Third, our study identified that MMC-Zfhx4$^+$ cells have a molecular profile distinct from the typical MMC MNs that were previously considered as a relatively

homogenous cell population. Both groups of MMC MNs occupy the medio-ventral region of the spinal cord and co-express Lhx3 and Mnx1, but MMC-Zfhx4$^+$ cells express only a low-level Isl1 and turn on a specific gene set, such as Zfhx3/4. In the future, it will be interesting to investigate if MMC-Zfhx4$^+$ and classic MMC MNs possess distinct functionality. Fourth, in *Kdm6b*-cKO mice, high-level Isl1-expressing MN subtypes, such as MMC, HMC, LMCm, LMCd, and PGC-Isl1$^+$ MNs, decrease, whereas low-level Isl1-expressing or Isl1-negative MN subtypes, such as LMCl, PGC-Isl1$^-$, and MMC-Zfhx4+ MNs, increase. These results suggest that Kdm6b plays a central role in maintaining Isl1 levels in high-level Isl1-expressing MN subtypes, thus contributing to the balanced production of MN subtypes. Given that Isl1-Lhx3 binds and activates the transcription of the Isl1 gene[22] and that Kdm6b acts as a critical coactivator of Isl1-Lhx3, the chromatin accessibility established by the cooperative actions of Isl1-Lhx3 and Kdm6b may be important in maintaining Isl1 levels even after Lhx3 is down-regulated. In addition, Kdm6b may collaborate with other TFs to sustain Isl1 expression. Onecut TFs are good candidates in this regard, as Onecut TFs directly stimulate Isl1 expression and maintain Isl1 levels at the time of MN diversification[65], facilitating the recruitment of Isl1-Lhx3 to late-stage MN enhancers[66], and also form a complex with Isl1 in MNs when Lhx3 is downregulated[57]. Fifth, newborn MNs and MMC MNs share expressions of many marker genes, such as Isl1/2, Lhx3/4, Mnx1, and Mecom, but show a difference in that only nbMN cells highly express TFs involved in neurogenesis, such as Neurog2, Neurod1/2/4/6, Nhlh2, Ebf2 and St18[12,67–71]. Thus, the timely down-regulation of these neurogenic TFs is a prerequisite for acquiring the fully-differentiated neuronal features, while their induction is required for initiating neuronal differentiation. Sixth, the two control clusters, CC8 (V2IN) and CC9 (V3IN), exhibited hybrid neurotransmitter characteristics co-expressing the cholinergic gene Slc18a3 and the glutamatergic gene Slc17a. Given that CC8 and CC9 express a relatively low level of neurofilament genes and are close to nbMN cluster in hierarchical analysis, they may represent the cells consolidating their neurotransmitter identity soon after being born from the dorsal and ventral boundaries of the pMN domain. Our findings are in line with the previous studies that Olig2 is transiently expressed in pMN-neighboring progenitor domains[39,40] and the cross-repressive interactions between MN-specific TF Mnx1 and V2aIN-specific TF Vsx2 in differentiating neurons are needed to effectively separate MN and V2aIN fates[13,21]. CC8 and CC9 cells may also contribute to the heterogeneity of interneurons in the mature spinal cord. In this regard, it is noteworthy that a subpopulation of V2aIN in the adult spinal cord co-express glutamatergic and cholinergic neurotransmitter genes[72,73]. Last but not least, our study provided valuable resources to study MN diversification, such as comprehensive markers for each MN subtype. These markers include genes with diverse molecular activities, such as cell adhesion and transmembrane receptor genes that may play roles in columnar separation and axon guidance in each MN subtype. Our study also identified much-needed markers for HMC MNs, such as Klf5, which is more robust and specific to HMC than the previously used marker Etv1.

Taken together, our study shows that Kdm6b serves as a critical transcriptional coregulator that orchestrates the balanced production of MN subtypes by building the exquisite gene regulatory network, along with a cohort of TFs.

## Methods

**Mouse lines**. *Kdm6b*$^{flox/flox}$ (*Kdm6b*$^{f/f}$), *Kdm6a*$^{f/f}$, *Olig2-Cre*, *Isl1-Cre*, *mTmG* and *Mnx1p::Lhx3* mouse lines were described previously[23,32,38,39,41,74]. The mice were housed at 12 h dark/12 h light cycle, 25 °C, and 50% humidity conditions. All housing and analyses were approved by the University at Buffalo Institutional

Animal Care and Use Committee (IACUC) (protocol number, 201900078). For timed mating, embryos were considered to be at 0.5 day of gestation (E0.5) on the day when a vaginal plug was detected.

**DNA constructs**. Mouse Kdm6b nucleotides 801–1059, Kdm6a nucleotides 1381–1950, Klf5 nucleotides 1–763, and Kcnip4 nucleotides 1–295 were cloned into pBluescript vector for making in situ hybridization (ISH) probes. Kdm6b's N (1–809 amino acids)- and C (672–1642 amino acids)-terminal regions were cloned into pcDNA-3HA mammalian expression vector. The other DNA constructs used in this study included pBS-mouse Slc18a3, pBS-mouse Aldh1a2, pCIG-Kdm6b, pcDNA3-Flag-Lhx3, pcDNA3-Flag-Isl1, pCS2-Myc-Isl1, pCS2-HA-Isl1-Lhx3, and TK-HxRE:Luc, as reported previously[4,13,14,16–18,36].

**Immunohistochemistry and in situ hybridization**. The mouse tissues were collected at various stages, fixed in 4% paraformaldehyde for 2 h, immersed in 30% sucrose for additional 2 h, and embedded in OCT blocks for cryosectioning and subsequent analyses. Assays were performed as following[75]. For immunohistochemistry, 12 μm cryosectioned slices were incubated with the primary antibodies overnight at 4 °C using 2% BSA blocking buffer (0.1% TritonX-100 in PBS buffer). Then, fluorophore-conjugated species-specific secondary antibodies were used as recommended. The secondary antibodies from Jackson ImmunoResearch Laboratories include donkey anti-goat Alexa Fluor 488 (Cat.705- 545–147, 1:500), donkey anti-guinea pig Alexa Fluor 488 (Cat.706–545–148, 1:500), donkey anti-rabbit Alexa Fluor 488 (Cat.711–545–152, 1:500), donkey anti-goat Alexa Fluor 594 (Cat.705- 585–147, 1:500), donkey anti-guinea pig Alexa Fluor 594 (Cat.706–585–148, 1:500), and donkey anti-rabbit Alexa Fluor 594 (Cat.711–585–152, 1:500). Zeiss Imager Z2 fluorescence microscope with Apotome was used for imaging. Primary antibodies were listed as follow: goat anti-Olig2 (R&D Systems AF2418, 1:500), rabbit anti-Olig2 (Millipore AB9610, 1:1000), guinea pig anti-Mnx1 (Homemade, 1:500), rabbit anti-Isl1 (Abcam ab109517, 1:250), guinea pig anti-Isl1 (Homemade, 1:2000), rabbit anti-Lhx3 (Abcam ab14555, 1:1000), guinea pig anti-Lhx3 (Homemade, 1:250), guinea pig anti-Vsx2 (Homemade, 1:1000)[47], guinea pig anti-Sox14 (Homemade, 1:500)[21], rabbit anti-Nkx2-2 (Abcam ab191077, 1:50), rabbit anti-Foxp1 (Abcam ab16645, 1:2000), rabbit anti-Nos1 (Immunostar 24287, 1:1000), rabbit anti-Kdm6b (Novus Biologicals NBP1-06640, 1:100), guinea pig anti-Klf5[76], rabbit anti-Mecom (Cell Signaling Technology #2593, 1:500), sheep anti-Zfhx3 (Novus Biologicals AF7384, 1:200), and rabbit anti-Zfhx4 (Sigma HPA023837, 1:200).

For in situ hybridization analysis, embryos were harvested and cryosectioned at 18 μm. The transverse sections were post-fixed in 4% PFA, permeabilized with proteinase K, treated with acetylation, and then hybridized overnight with digoxigenin-labeled riboprobes. After washing and blocking, sections were incubated with anti-digoxigenin alkaline phosphatase-conjugated antibody (Roche). The color reaction was performed with NBT/BCIP substrates (Roche). The plasmids to make digoxigenin-labeled probes complementary to mouse Kdm6b, Kdm6a, Slc18a3, Aldh1a2, Klf5, and Kcnip4 were mentioned in the section of DNA constructs. For in situ hybridization combined with immunohistochemistry, after color development from in situ hybridization, the slides were washed, incubated with Nos1 antibody overnight at 4 °C, and then incubated with secondary antibody conjugated with HRP (peroxidase) enzyme for 2 h. DAB (3,3′-diaminobenzidine) substrate kit (Vector) was used to develop brown color according to the manufacturer's protocol (Vector).

**Half-mount axonal analysis**. Mouse embryos at E12.5 or E13.5 were harvested, fixed in 4% PFA in PBS overnight at 4 °C. After washing in PBS, embryos were transferred through a series of ascending MeOH/PBS concentrations and equilibrated in absolute MeOH for 1 h on ice. To quench autofluorescence, embryos were incubated in Dent's bleach solution (MeOH: DMSO: 30% H$_2$O$_2$, 4:1:1) for 2 h at room temperature. After equilibrating to PBS through a series of descending MeOH/PBS concentrations, embryos were dissected from the midline into two sides. Half embryos were incubated in blocking buffer (3% BSA in PBST with 0.2% TritonX-100) for 2 hours, treated with GFP antibody (Abcam ab290, 1:500) for 2 days, and then with secondary antibody overnight at 4 °C. After washing with PBST with 0.2% TritonX-100 followed by washing with PBS, tissues were equilibrated to absolute MeOH through graded MeOH/PBS concentrations and then cleared with BABB solution (benzyl alcohol: benzyl benzoate,1:2). The transparent half-embryos were used for imaging with Nikon spinning disk confocal microscope (Yokogawa CSU-W1).

**Single cell preparation and fluorescence-activated cell sorting**. Spinal cords were dissected from E12.5 embryos (control sample from Kdm6b$^{f/+}$;Olig2-Cre;mTmG and Kdm6b-cKO sample from Kdm6b$^{f/f}$; Olig2-Cre; mTmG) and diced into small pieces (~1 mm³). After washing and spinning down, the pellets were dissociated with Papain Dissociation System (Worthington # LK003150) following the manufacturer's instruction. Briefly, the tissues were enzyme digested for 1 h at 37 °C and then gently triturated. The single cell solution was washed and then suspended in the medium (Neurobasal medium: DMEM/F12, 1:1) with 10% fetal bovine serum (FBS) at 4 °C. Propidium Iodide (PI) was added into the solution and subjected to fluorescence-activated cell sorting (FACS, BD Influx). The sorted GFP

$^+$PI$^-$ cells were collected in the non-sticking tube (ThermoFisher #AM12450) with 0.5 ml medium (neurobasal medium:DMEM/F12, 1:1) with 10% FBS.

As our mating scheme (Kdm6b$^{fl/+}$;Olig2-Cre;mTmG/mTmG male x Kdm6b$^{fl/fl}$ female) produce one Kdm6b-cKO (Kdm6b$^{f/f}$;Olig2-Cre;mTmG/+) in each litter most of the time, we used one Kdm6b-cKO and one littermate control embryo as a set of the scRNAseq libraries, which were processed under the same condition. We performed the scRNAseq analyses in two separately collected biological replicates (two sets of one control and one Kdm6b-cKO; a total of four 10x samples processed). The first set included less limb-level spinal cord (i.e., less LMC) than the second set, and we optimized our dissection range of the spinal cord for the second set to improve the representation of all MN subtypes. The results from the two biological replicates were highly similar to each other in cell ratio changes. Thus, we reported the second set of scRNAseq results as this set had a better representation of all MN subtypes in both limb and thoracic levels. Importantly, the scRNAseq results were validated with independent methods, such as immunohistochemical and motor axon trajectory analyses.

**Single cell RNAseq, data processing, and alignment**. Single cell suspensions from FACS were loaded onto a Gemcode single cell machine (10x Genomics) to generate single-cell gel beads in emulsion (GEMs). Single cell RNAseq (scRNAseq) libraries were constructed using Chromium Single Cell 3′ V2 reagent kit. The scRNAseq libraries were then sequenced on the Illumina NextSeq 500. Raw sequencing cell barcodes were filtered to distinguish valid cell barcodes from empty cell barcodes using an algorithm in Cellranger count v2.1.1[43], which is an analysis pipeline for Chromium single cell 3′ RNA-seq result with an expected recovered cell parameter; expect-cells set to 9000 and 7800 for control and Kdm6b-cKO, respectively. The filtered reads were then aligned to mm10 (refdata-cellranger-mm10-1.2.0) using STAR v2.4.0 with a default setting[77]. Only the confidently mapped (MAPQ = 255), non-PCR duplicates with valid barcodes and UMIs were employed to generate gene-barcode matrix. Among 403,858,864 and 421,556,269 reads, 87.60% and 87.40% reads were mapped for control and Kdm6b-cKO, respectively.

**Filtering, Slc18a3$^+$ cell picking, normalization, and dimensionality reduction**. Gene-barcode count matrix generated was read using the Read10X function in Seurat v3.1.4[48]. Initially, we kept all genes expressed in >3 cells and all cells with at least 200 detected genes. To exclude cells in poor quality and possible doublets, possible outliers of cells were further removed by visual inspection of the distribution of a total number of mapped genes. Cells with a high level of mitochondrial genes (>7%) were also removed, which were treated as mitochondrial cytoplasmic RNAs, which were released and sequenced due to cell lysis. Cells with a high level of hemoglobin gene expression (Hba-a1 > 10) were also removed, as they are likely to represent rare blood cell contaminants. Among the GFP-positive cells that passed quality control (Supplementary Figs. 4a, 7a), we isolated Slc18a3$^+$ cells (Slc18a3 > 0) as MN lineage cell population, leading to 3020 and 3278 cells in control and Kdm6b-cKO, respectively. We also checked for the presence of biases according to the number of Unique Molecular Identifier (UMI), a number of genes, and percentage of mitochondrial genes in low-dimensional feature space (Supplementary Figs. 5a, 7b), as well as for the presence of batch effects between two conditions before Seurat canonical correlation analysis (CCA) integration (Supplementary Fig. 7c, d). The resulting gene-barcode count matrix was then processed using a global-scaling normalization method LogNormalize with gene expressions in each cell being normalized by the total expression, then multiplied by a scale factor 10,000 to account for the sequencing depth, and then transformed in log scale. After normalization, we calculated scaled expression (z-scores for each gene) for downstream dimensionality reduction. To reduce feature dimensionality, we first identified highly variable genes, for which the average expression and dispersion (variance to mean ratio) of each gene across all cells were calculated using the FindVariableGenes function with a dispersion option of LogVMR[48]. To mitigate unwanted sources of variation from confounding factors, a linear model was used to regress out the effects of the number of detected molecules per cell as well as the percent of mitochondrial gene expression. Then, the resulting residuals were zero-centered using ScaleData function to bring the gene expressions in the same range. Principle component analysis (PCA) was performed on the scaled matrix with highly variable genes selected as above. To determine the significant number of principal components (PC) required for downstream analysis, Jack-Straw function in Seurat package was used to check the significance of PC from an empirical null distribution. To embed our cells in low dimensional feature space, we employed t-distributed stochastic neighbor embedding (tSNE) technique[78] using 20 PCs[79], upon which feature plots were drawn to color single cells according to their gene expression using FeaturePlot function in Seurat.

DoubletFinder[80] identified 4% of 3020 control Slc18a3+ cells (total 122 cells mostly from pMN (CC0), nbMN (CC1), and MMC (CC2)) as potential doublets. Considering that the pMN, nbMN, and MMC cells are transitioning from progenitor state to fully differentiated MN state, our results are in line with the view that it is difficult to interpret doublet predictions in data containing cellular trajectories because, by definition, cells in the middle of a trajectory are always intermediate between other cells and are liable to be incorrectly detected as doublets[81].

**Modularity-based clustering and cell-type marker gene determination in the control sample**. To partition cells into clusters, cells embedded in tSNE space were grouped using a graph-based clustering approach implemented in Seurat, in which cell-cell distances are computed based on their identified PCs to detect communities in network with Louvain algorithm using *FindClusters* function. The whole GFP$^+$ control sample of 8,815 cells was partitioned into 13 clusters, prefixed with 'OC' for *Olig2-Cre* lineage clusters (OC0-12), with a resolution parameter of 0.6. Slc18a3$^+$ 3020 and 3278 cells in control and *Kdm6b*-cKO were grouped into 11 and 9 clusters using a resolution parameter of 0.6 and 0.5, respectively, to match the number of clusters similar to each other to an extent. They were labeled with a prefix of 'CC' (control cluster, CC0-10) and 'KC' (knockout cluster, KC0-9) for control and *Kdm6b*-cKO, respectively. To define cell-type-specific markers in each cluster, we used *FindAllMarkers* function in Seurat, in which the fold-change ratio was calculated by comparison of expression level between one cluster versus all the remaining clusters. Feature plots were used to color single cell in blue according to their gene expression using *FeaturePlot* function in Seurat. Expression levels of marker genes in each cluster were visualized by violin plots using *VlnPlot* function in Seurat. Genes with Log2FC > 0.4, an adjusted *p*-value < 0.01 were called as cell-type-specific markers. The marker genes in the heatmap were visualized using ggplot2[82].

**Single-cell transcriptome comparison between control and *Kdm6b*-cKO samples**. To compare single-cell transcriptomes between two different conditions of control and *Kdm6b*-cKO in an unbiased way, we first integrated two data sets by aligning them onto the common low dimensional feature space using Seurat, which learns a shared gene correlation structure that is conserved between data sets using CCA approach. For the gene-barcode count matrix from each condition, individual-specific effect of the fraction of mitochondrial gene expression was regressed out using *ScaleData* before CCA to mitigate the source of variation from each data set. For each condition, we obtained 1,000 highly variable genes by calculating the dispersion for all genes and took their union set of genes being used as input to run *RunCCA* with number of CC of 30. Then, we used *Alignsubspace* function with 20 PCs to run nonlinear warping algorithms to normalize for differences in feature scale, which enables us to perform integrated downstream analyses including unbiased clustering analysis and differential gene expression testing. We then ran *FindClusters* function with a resolution parameter of 0.6 to partition cells into clusters. All 6298 cells were grouped into 13 different clusters being prefixed with 'MC' (merged cluster). Cell types of these clusters were determined by comparing the enriched genes in MC to the identified gene markers in the control sample.

**UMAP projection analysis to check for the presence of batch effect between control and *Kdm6b*-cKO samples**. To check back the possibility of Seurat CCA integration forcing to introduce biases due to the presence of batch effects, we tried to embed both the Slc18a+ cell populations of control and *Kdm6b*-cKO unto the same UMAP (Uniform Manifold Approximation and Projection) feature space. For this UMAP analysis, we used python packages pandas v1.1.2, Scanpy v1.6.0, and anndata v0.7.6.

Raw count matrix of Slc18a3$^+$ population of cells for each condition of control and *Kdm6b*-cKO was read in a dataframe using python pandas package individually, 3020 cells for control and 3278 cells for *Kdm6b*-cKO. Each cell was annotated with information of both 'batch" (or condition) and 'celltype", which was previously defined as from the CCA integration of both control and *Kdm6b*-cKO population of cells. These two matrices were simply concatenated into one matrix containing 6298 cells using anndata.concatenate function. With this concatenated matrix, PCA coordinates were computed using scanpy.tl.pca function with default parameters and then a neighborhood graph of observations was computed using scanpy.pp.neighbors function with default parameters. We then embedded the neighborhood graph using scanpy.tl.umap function with each cell being visualized using scanpy.pl.umap function, which is color-coded based on 'batch" and 'celltype" as defined from the original CCA integration.

**Pseudo-bulk differential gene expression in an integrative framework**. Pseudo-bulk profile analysis[50] allows us to perform differential gene expression (DGE) between two conditions while mitigating technical noises by summing read counts based on the experimental condition. First, we aggregated cell-level counts into sample-level according to individual sample conditions being treated as an in silico mimicry of real bulk RNAseq sample. Next, we ran *WilcoxDETest* in Seurat package for DGE testing for each condition, and their *p*-values were adjusted with *p.adjust* function with -fdr option in the R package for multiple hypothesis testing correction. In addition to DGE testing, we counted the number of cells expressing a certain gene (nUMI > 1) to get a percentage of cells expressing that gene by dividing the total number of cells in each condition. Pseudo-bulk analysis was employed using the whole cells from control and *Kdm6b*-cKO samples as well as the cells from each cluster merged with two conditions[50]. In the dot plot analysis for MN fate-specifying TF genes and MN subtype marker genes, gene expression level and the percentage of cells expressing certain gene were visualized by color gradient and circle size respectively using ggplot2 v3.3.5[82].

**Co-immunoprecipitation and western blotting assays**. HEK293 cells (ATCC) were transfected using Lipofectamine 2000 (Invitrogen) according to the manual instruction and, after 48 h, they were harvested in lysis buffer (20 mM Tris-HCl, pH8.0, 120 mM NaCl, 1 mM EDTA, 0.5% NP-40, and protease inhibitors). The lysates were subsequently incubated with 2 μg of antibodies overnight and 20 μl of Protein A Agarose for 2 hours at 4 °C. The immunoprecipitated proteins were evaluated by western blotting. Blotted membranes were scanned using Odyssey infrared imaging system (Li-COR). The antibodies used for immunoprecipitation and immunoblotting were listed as below: mouse anti-Flag (Sigma), mouse anti-HA (Covance MMS-101R), rabbit anti-HA (Abcam ab9110), mouse anti-Myc (Millipore 05-419). Uncropped and unprocessed scans of the blots are provided in the Source Data file.

**Transfection and luciferase assay**. HEK293 cells were cultured in DMEM media supplemented with 10% FBS and seeded in 48 well plates overnight prior to transfection with Lipofectamine 2000 (Invitrogen). TK-Luc or *HxRE*-TK-Luc luciferase reporter plasmids were transfected with pCS2-Isl1-Lhx3 with or without pCIG-Kdm6b or pCIG-Kdm6a. An actin promoter-β-galactosidase plasmid was cotransfected for normalization of transfection efficiency. Cells were lysed 48 h post-transfection, and cell extracts were assayed for luciferase activity. The values were normalized with β-galactosidase activity. Histograms show mean normalized luciferase units and error bars represent standard deviation from representative experiments. All transfections were independently repeated at least three times.

**Chromatin immunoprecipitation (ChIP) assay**. ChIP assays were performed as follows[75]. Briefly, E12.5 mouse spinal cords were dissected, and tissues were suspended in cold PBS by pipetting up and down for several times. Cells were then fixed by 1% formaldehyde for 10 min, quenched with 125 mM glycine for 5 min, and then lysed in cell lysis buffer for 5 min at 4 °C. Nuclei were harvested from lysates, and suspended in nuclei washing buffer for 5 min at 4 °C. Nuclei were pelleted and then subjected to sonication for DNA shearing. Sonicated chromatins were immunocleared with 3 μg IgG and 20 μl prerinsed protein A agarose beads for 2 h at 4 °C, followed by collection of the supernatant. Immunoprecipitation of chromatin was then performed by incubating with Kdm6b antibody and protein A agarose beads overnight at 4 °C. After pull-down of chromatin/antibody complex, the beads were sequentially washed with low salt, high salt, LiCl solutions and TE buffer. Protein/chromatin complexes were eluted and de-crosslinked by incubation at 65 °C overnight. Eluate was incubated with Proteinase K at 42 °C for 2 h. Next, DNA was purified with phenol/chloroform, and DNA pellet was precipitated by ethanol and dissolved in water. The Kdm6b antibody was raised using the antigen (human Kdm6b 798–1095 amino acids) and was utilized in a previous report[83]. Quantitative real-time PCR was performed using FastStart Essential DNA Green Master (Roche) and LightCycler 96 system (Roche). The input was used for normalization. All ChIP experiments were independently repeated at least three times. Data are represented as the mean of triplicate values obtained from one representative experiment and error bars represent standard deviation. The ChIP-qPCR primers were listed as follow. Mnx1 forward: 5′-CGTCTGTCACTGCAGGAGG and reverse 5′-GCAACACTTCCAGGCTCAG; Isl1 forward: 5′-GTGCGTGCATTTGCTTATTTC and reverse 5′-GTGTAGTAATT AGATCTTAAGTGAGGC; Lhx4 forwards: 5′-CAGATTTGAACGTAGCGCATG and reverse 5′-AATAATGTGAGTGCATGATGGG; Untr6 forward: 5′-TCAG GCATGAACCACCATAC and reverse 5′-AACATCCACACGTCCAGTGA.

**Quantification and statistical analyses**. Cell count was obtained per section or area with indicated gene markers. At least 3 embryos for each condition and at least 3 sections per embryo were used for quantification. Error bars represent the standard deviation of the mean. The statistical significance was determined using unpaired two-tailed Student's *t*-test.

**Reporting summary**. Further information on research design is available in the Nature Research Reporting Summary linked to this article.

## Data availability

The datasets generated during and/or analyzed during this study have been deposited in the Gene Expression Omnibus (GEO) under accession number "GSE156609". The plasmids and antibodies are available upon request. A reporting summary for this article is available as a Supplementary Information file. Source data are provided with this paper.

## Code availability

Exemplary scripts to process and analyze data are available at https://github.com/epigenomekdm6b/kdm6b.

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

## Acknowledgements
We are grateful to Naoki Iwamori for *Kdm6b* flox mice, Ben Novitch and Thomas Jessell for *Olig2-Cre* mice, Sam Pfaff for *Mnx1p::Lhx3* transgenic mice, Jeffrey Whitsett for Klf5 antibody, Martínez-Balbás for *Kdm6b* plasmid, Younjung Park and Hyereyong Park for the excellent mouse management and antibody generation, and the Lee laboratory members for helpful discussions. This research was supported by grants from NIH (R01NS111760 and R01NS100471 to S.-K.L., and R56DK118912 and R01NS118748 to J.W.L.).

## Author contributions
W.W. performed the experimental studies and data analysis. H.C. designed and carried out the computational data analysis. J.W.L and S.-K.L supervised the work.

## Competing interests
The authors declare no competing interests.
