## [Peer Review File · Nature Communications]

Reviewers' Comments:

Reviewer #1:

Remarks to the Author:

In this paper the authors use genetic, biochemical, and single-cell transcriptomic methods to provide evidence that the demethylase Jmjd3 interacts with the Isl1-Lhx3 complex to promote medial and hypaxial motor column fates in the developing mouse spinal cord. Additionally the authors reveal novel diversity in developing spinal-cord motor neurons. Although taken together, the results suggest that Jmjd3 plays a role in spinal cord development, the authors' analysis could be strengthened by addressing the questions below:

- For the single-cell experiments, how many embryos were used for the control and cKO samples? Which version of the 10X kit was used, and were only two 10X samples processed (one control and one cKO)?
- It would be useful to see QC panels in the supplementary, e.g. a t-SNE with UMI counts, gene counts, mitochondrial counts, etc. It might also be useful to score cells with a doublet-finding algorithm like DoubletFinder, particularly to support that CC8 and CC9 have hybrid characteristics.
- Why are the control and cKO datasets first analyzed separately (Fig 3-4) and then together (Fig 5) if the cell types are mostly redundant across datasets? Would it be possible to avoid using the "matching score" to match cell types (i.e. Fig 5C) by performing only a combined analysis, and calling a single cluster set?
- Fig 4E: My understanding is that trajectory methods like Monocle will "force" a trajectory onto a dataset, which warrants caution when interpreting the results. For example, what would the Monocle results look like if the V2IN, V3IN, and ND clusters were included (if I understand correctly, Monocle would still connect them)? Especially because there are large gaps between the clusters on the UMAP, I think more data and/or analysis would be necessary to support the hypothesis that HMC and PGC cells pass through an MMC state (e.g. by capturing intermediate cell types or examining these "island" cells for a hybrid transcriptional character). Similarly, the claim that PGC-Isl1⁻ cells emerge from PGC-Isl1⁺ cells would be better supported by adding, for example, RNA velocity analysis on PGC cells. However, Figure 4 does not seem essential to the authors' claims about Jmjd3.
- Fig 5: Why have the authors analyzed the cKO and control datasets using CCA (according to supp. methods), which would seem to force integration of the two conditions? What does the analysis look like with PCA (i.e. without integration)? Do control and cKO cells form entirely different clusters for all cell types, or do some cell types still largely overlap transcriptionally? This kind of analysis additionally seems like it might address some questions about Figure 6, below.
- Fig 6A: If the authors have only analyzed two 10X samples, how certain is it that these differences in cell ratio are not caused by sampling / batch effects?
- Fig 6B: Does the authors' method control for cluster size (number of cells) and depth (UMI counts) when comparing the results of DE analyses across clusters? Perhaps larger clusters mean more statistical power, and fewer statistically DE genes are found in immature cell types because they express fewer genes in general.

Minor points:

- Fig 1A in situ results are hard to see. Is there an antisense control for comparison, or maybe arrowheads can be used to annotate better?
- Fig 1C: how many sections were analyzed?
- Fig 6D: Is there a way the authors could mark which genes in this plot are differentially expressed?
- The resolution of some of the figures seems low, especially the t-SNE plots (e.g. Figure 3E-I), and the color scheme used for plotting gene expression makes it a little hard to distinguish low-expressing from non-expressing genes.

Reviewer #2:

Remarks to the Author:

In this study, the authors address the question of cell fate diversification in the developing CNS using specification of MN subtypes in the developing spinal cord as a model system. Specifically,

they examine the role of the histone demethylase Jmjd3 in this process by phenotypic analyzes of Jmjd3 mutant embryos and by characterizing MNs from WT and Jmjd3 mutant embryos by single cell RNA-sequencing (scRNA-seq). Based on this, the authors propose a model whereby Jmjd3 act as a co-activator of the Isl1-Lhx3 complex, thereby promoting MMC and HMC MN fate and suppressing acquisition of LMC and PGC MN identities. The study is well-performed and interesting albeit there are some significant concerns outlined below that the authors should address.

Major points

To examine the role of Jmjd3 in MN diversification the authors generate conditional Jmjd3 mutants in which Jmjd3 is deleted at a progenitor stage (Jmjd3cko-Olig2cre), which results in changes in the numbers of MN columnar subtypes, defects in axonal projection, and decreased expression of genes defining the different MN clusters. However, when deleting Jmjd3 in post-mitotic MN (Jmjd3cko-Isl1cre) the effects on MN cluster production are not fully recapitulated. They observe a reduction of MMC cluster (as observed with Jmjd3cko-Olig2cre) but not a corresponding increase of LMC or PGC MNs. This discrepancy of phenotypes needs to be better addressed. The authors mention that Isl1-Cre expresses the Cre recombinase at a later time in the MN lineage than Olig2-Cre, which formally is correct. However, expression of Isl1 is nevertheless initiated at an early stage when progenitors exit the cell cycle and differentiate into neurons. This indicate that Jmj3 may function already at an immature progenitor stages and via a mechanisms that at least in part should be independent to the Isl1-Lhx3 complex. Thus, data from the Jmjd3cko-Isl1cre analysis is not in agreement with the author's model assigning the fate determining activity to Jmjd3 Isl1-Lhx3 complex. The authors should try to better define the temporal requirement Jmjd3 in the differentiation process, and explore potential additional functions for Jmjd3 in this differentiation lineage. One experiment that could provide insight into this would be to compare scRNA-sequencing data from Jmjd3cko-Olig2cre embryos to corresponding data generated from Jmjd3cko-Isl1cre embryos. It would also be interesting to compare MN axonal projection patterns in WT and Jmjd3cko-Olig2cre embryos with data from Jmjd3cko-Isl1cre mice.

The authors provide nice data set of scRNA-seq data from WT cells which allows the authors to identify the different MN clusters and describe developmental trajectories. However, when analyzing cells from Jmjd3cko-Olig2cre mutant embryos, instead of making a projection into the clustering analysis of WT cells, the authors pooled both wt and mutant cells and performed a new clustering analysis. It is not clear to me why the authors did this. The authors should also provide a projection of mutant cells into the 2D coordinate space defined/trained by WT cells (e.g. using UMAP's transform feature) to allow readers to better understand the transcriptional shifts and distribution of MN cluster identities in Jmjd3 mutants. By pooling mutant and WT cells together, the authors risk adulterating WT cell classifications through dilution with KO transcriptional signatures. Additionally, the authors should provide a developmental trajectory analysis for the mutant cells as performed for WT cells (Fig-4E). In their analysis of WT cells, there is a bifurcation at the nbMN level that results in a MMC and LMC differentiation branch at branchial level or in a MMC and HMC/PGC differentiation branch at thoracic level. It would be interesting to know, and possibly provide insights into the function of Jmjd3, how these developmental trajectories are maintained or changed in mutant cells.

The analysis of scRNA-seq provided a more refined overview of the changes on MN clusters observed in Jmjd3cko-Olig2cre mutants (e.g., increase of LMCI and decrease of LMCm/LMCD; increase of PGC-Isl1- and decrease of PGC-Isl1+). The authors should validate these results in sections of mutant spinal cords.

The authors propose that Jmjd3 is recruited to Isl1-Lhx3-bound target genes and activate their transcription. They show by ChIP-PCR in spinal cord tissue the recruitment of Jmjd3 to targets of Isl1-Lhx3 complex, interaction of Jmjd3 to Lhx3 but not Isl1, and an increase of reporter activity in luciferase assays. While Jmjd3 can interact with Lhx3 in IP assays, the authors propose that Jmjd3 interacts with the Isl1-Lhx3 complex to regulate gene expression. This would be better supported if the authors could provide data establishing the presence of Jmjd3 in the Isl1-Lhx3 complex. Additionally, as Jmjd3 is a histone demethylase as highlighted by the authors in their title. The authors should provide more direct evidence that Jmjd3 regulates transcription of target MN genes via its histone demethylase activity.

Minor points

In Figure 1 the authors should count the total number of MN in control vs Olig2-Cre::Jmjd3fl/fl embryos in order to show that the loss of MN subtypes in mutant embryos is not also due to cell loss. The measurement of Slc18a3 expression area may not be appropriate considering expression of Slc18a3 also in V3 and V2 INs

Authors argue for an increase of PGC-Nos1 MNs in mutant spinal cords, but from the data presented (FigS1B, 1B), there seems to be a reduction. At E13.5 the signal of Nos1 is present but it is difficult to know if there is a change in numbers. In the scRNA-seq analysis the authors identify 2 PGC clusters (Fig3C). The authors argue that both clusters are Nos1+, but their data actually argues the opposite, one cluster seems to be FoxP1+/Zeb2+/Nos1+/Isl1+ (cluster CC6) and the other FoxP1+/Zeb2+/Nos1-/Isl1- (cluster CC7). The authors should quantify the number of both clusters in the mutants.

It would be interesting to see the distribution of Jmjd3 expression in the different clusters identified in wt cells, similarly to what the authors show in Fig.3B.C.

The authors should update the nomenclature used for protein and genes names.

Reviewer #3:

Remarks to the Author:

In this extensive manuscript Wang et al characterize the role of the H3K27 demethylase Jmjd3 in developing motor neurons. The authors provide evidence that Jmjd3 promotes the formation of specific MN subtypes at the expense of others. Wang et al propose that this involves the MN specifying transcription factors Isl1 and Lhx3. The authors also report single cell transcriptome data from MNs.

Understanding the basis of MN subtype identity is an interesting and well explored question. The current study provides a new angle to this question. The study also provides a useful dataset of single cell mRNA sequencing of MNs.

A difficulty with this manuscript is that it has the feel of two studies that have been merged. On the one hand the authors have undertaken a single cell transcriptome analysis of motor neurons from e12.5 mouse embryos. Distinct from this, the authors characterized a role for Jmjd3 in MN subtype identity. It might be easier for readers if these two studies were presented separately.

Specific points

-The authors show (Fig 1A) that Jmjd3 is expressed in all neurons but only comment on MNs. It should be clearer to a reader that Jmjd3 is not specific to MNs. Related to this the authors conclude that the perinatal death of mice with Jmjd3 knocked out using Olig2-cre suggests a role of Jmjd3 in MN development. However, the cause of death could relate to another function of Jmjd3 in Olig2 expressing cells.

- Since the Olig2-cre will also delete Jmjd3 in V3 neurons, ventral to MNs, and some V2 neurons, dorsal to MNs, the authors should also examine whether there is an effect on these neuronal populations.

- Do the increases in LMC and PGC MNs account for the decreases in MMC and HMC? I accept this analysis is not straightforward but some indication of whether the data could be explained solely by a change in identity would be helpful.

-The single cell transcriptome data (Fig 3) was generated by selecting cells expressing the cholinergic marker Slc18a3. This is expressed by post-mitotic neurons. It is surprising, therefore, that the authors identify clusters of cells that appear to have the gene expression profile of undifferentiated progenitors (expressing Sox2 and Hes5). Some explanation for this should be offered. Related to this the cells were isolated from e12.5 embryos, which is relatively late in the differentiation window of MNs. This raises the possibility that the Sox2/Hes5 cells might not be MN progenitors and this would mean the developmental trajectory proposed in Fig 4 is erroneous.

-The interpretation that all of the cells profiled in the analysis are MNs relies on the specificity of expression of Slc18a3. This caveat should be noted and the possibility that Slc18a3 is expressed in some subtypes of interneurons (eg. V2 neurons) should be considered.

-While the trajectory analysis (Fig 4) is consistent with assumptions in the field, the limitations should be noted. As indicated above, using e12.5 progenitors to anchor the origin of MNs that were generated a day or two earlier is potentially problematic. The inferred relationships between cell types indicates similarities in gene expression, but does not necessarily relate directly to the lineage relationships. To determine lineage, fate mapping experiments would be necessary.

-The late inactivation of *Jmjd3* (with *Isl1-cre*) is a nice experiment to determine when *Jmjd3* is necessary for its activity. It would be interesting to know the functional consequences of this experiment in the mutant animals. Does the *Isl1-cre* mediated deletion of *Jmjd3* have an effect on MN axon navigation or muscle innervation? Does the deletion result in functional defects?

We would like to thank all three reviewers for commenting on our manuscript. We have addressed all comments as shown in blue below, and we believe that the revised manuscript became substantially stronger thanks to the reviewers' insightful comments.

REVIEWER COMMENTS

Reviewer #1 (Remarks to the Author):

In this paper the authors use genetic, biochemical, and single-cell transcriptomic methods to provide evidence that the demethylase Jmjd3 interacts with the Isl1-Lhx3 complex to promote medial and hypaxial motor column fates in the developing mouse spinal cord. Additionally the authors reveal novel diversity in developing spinal-cord motor neurons. Although taken together, the results suggest that Jmjd3 plays a role in spinal cord development, the authors' analysis could be strengthened by addressing the questions below:

→ Thank you for many constructive suggestions, which helped us to improve our manuscript.

- For the single-cell experiments, how many embryos were used for the control and cKO samples? Which version of the 10X kit was used, and were only two 10X samples processed (one control and one cKO)?

→ We utilized 10X Genomics droplet-based platform using Chromium Single Cell 3' v2 chemistry. As our mating scheme (Jmjd3 fl/+;Olig2-Cre;mTmG/mTmG male x Jmjd3 fl/fl female) produce one Jmjd3-cKO (Jmjd3 f/f;Olig2-Cre;mTmG/+) in each litter most of the time, we used one Jmjd3-cKO and one littermate control embryo as a set of the scRNAseq libraries. We performed the scRNAseq analyses in two separately collected biological replicates (two sets of one control and one Jmjd3-cKO; total four 10x samples processed). The first set included less of LMC MNs, and we optimized our dissection range of the spinal cord for the second set to improve the representation of all MN subtypes. The results from the two biological replicates were highly similar to each other in cell ratio changes. The scRNAseq results were also consistent with immunohistochemical analyses on embryonic sections, as shown in the validation quantification in a new figure (Fig. 8b) as well as other figures (Fig. 1,2). We reported the second set of scRNAseq results as this set had a better representation of all MN subtypes in both limb and thoracic levels.

- It would be useful to see QC panels in the supplementary, e.g. a t-SNE with UMI counts, gene counts, mitochondrial counts, etc. It might also be useful to score cells with a doublet-finding algorithm like DoubletFinder, particularly to support that CC8 and CC9 have hybrid characteristics.

→ As advised, we included QC panels, including t-SNE with UMI counts, gene counts, mitochondrial counts (Supplementary Figure 4a,5a, 8a,b, also see Rebuttal Fig. 1).

In the manuscript, we reported that CC8 (V2IN) and CC9 (V3IN) have "hybrid neurotransmitter

characteristics” given that CC8 and CC9 cells express both Slc17a6 and Slc18a3 (Fig. 4c, also please check the cyan-colored cells in Fig. 4f).

We applied the DoubletFinder¹ to our scRNAseq dataset using two different criteria (3% and a more stringent condition 7%) to all sequenced cells; total 8,815 cells in the control sample. We then monitored how many cells in CC0~CC10 were identified as doublets. As shown in the table below (Rebuttal Fig. 2), a small fraction of CC8 (V2IN) and CC9 (V3IN) was classified as possible doublets; with a 7% cut-off, 3 cells out of 167 cells in CC8 and 16 cells out of 144 cells in CC9. Notably, a substantial fraction of CC0 (pMN) and CC1 (nbMN) was identified as doublets; with a 7% cut-off, 45 cells out of 91 cells in CC0 (49.5% of CC0 cells) and 152 cells out of 224 cells in CC1 (47.9% of CC0 cells). Considering that the pMN (CC0) and nbMN (CC1) cells are actively transitioning from progenitor state to differentiated MN state, our results are in line with the view that “It is difficult to interpret doublet predictions in data containing cellular trajectories” (i.e. cells in developing tissues). By definition, cells in the middle of a trajectory are always intermediate between other cells and are liable to be incorrectly detected as doublets.”² Further supporting this view, CC3-CC7 containing differentiated MNs with established MN subtype properties had no more than 1 doublet even with a 7% cut-off (0~0.3% of cells in each cluster). Intriguingly, CC2 (MMC) has a relatively high number of possible doublets in this analysis compared to other MN subtype clusters; 5.1% (3% cutoff) or 10.1% (7% cutoff). Two factors are likely to have contributed to this unique property of MMC among MN subtype clusters. First, given that nbMN and MMC are neighboring clusters that share expression of many genes including Isl1/2, Lhx3/4 and Mnx1 (Fig. 4), the distinction between nbMN and MMC may be relatively fluid and cells transitioning from nbMN to MMC state may be counted as doublets. Second, our Monocle analysis suggest that MN subtypes undergo MMC-like state in acquiring fully established MN subtype characteristics (Fig. 5e). This transit nature of some of MMC cells could contribute to a relatively high doublet percentage in MMC.

Cluster #	Cell identity	n.doublets (7%)	n.doublets (3%)	n.cells	% CC (7% cutoff)	% CC (3% cutoff)
CC0	pMN	45	42	91	49.5	46.2
CC1	nbMN	152	37	224	67.9	16.5
CC2	MMC	54	27	533	10.1	5.1
CC3	HMC	0	0	293	0.0	0.0
CC4	LMCm	0	0	397	0.0	0.0
CC5	LMCI	1	1	341	0.3	0.3
CC6	PGC.Isl1+	0	0	325	0.0	0.0
CC7	PGC.Isl1-	0	0	463	0.0	0.0
CC8	V2IN	3	2	167	1.8	1.2
CC9	V3IN	16	13	144	11.1	9.0
CC10	ND	1	0	42	2.4	0.0
		272	122	3020		

Rebuttal Figure 2. DoubletFinder revealed possible doublets or cells in the middle of a trajectory in control samples.

- Why are the control and cKO datasets first analyzed separately (Fig 3-4) and then together (Fig 5) if the cell types are mostly redundant across datasets? Would it be possible to avoid using the “matching score” to match cell types (i.e. Fig 5C) by performing only a combined analysis, and calling a single cluster set?

→ There are two main reasons why it was needed to analyze the two datasets separately and then together as a merged dataset. First, our study is the first detailed report of single cell transcriptome of developing MNs. Thus, it is crucial to test if the clustering analysis of control dataset based on single cell transcriptomes identifies MN subtypes that were defined by the anatomic locations of the cell bodies, the axonal trajectory pattern and target tissues, and a few established markers. If we present only the analysis of the merged dataset (combining the control and Jmjd3-cKO samples) without showing the result of control sample, the question will remain if Jmjd3-deficient MNs affected the overall clustering analysis, leading to the clusters significantly different from control clusters. Second, the analysis of Jmjd3-cKO dataset alone did not identify all MN subtypes. We omitted the analysis of Jmjd3-cKO dataset in the original manuscript to make it more concise. However, this analysis is now included in the revised manuscript (Fig. 6, also see Rebuttal Fig. 3). The unsupervised clustering of Jmjd3-cKO sample identified LMC and PGC cell types but failed to define clusters with clear MMC or HMC transcriptome profiles, consistent with the phenotypes detected by the marker analysis on embryonic spinal cord sections (Fig. 1, Supplementary Fig. 1). These results indicate that Jmjd3-deficient MNs failed to establish clear MMC and HMC subtype identity. However, if those Jmjd3-deficient presumptive MMC and HMC cells

maintain the overall gene expression profiles resembling their control counterparts, they can cluster together with their counterpart cell-type when analyzed as a merged dataset (control plus cKO samples). Thus, we performed the clustering analysis of the merged dataset. To determine if the clusters from the merged datasets are objectively similar to MN subtypes in control dataset, we used the “matching scores” rather than relying only on the several marker genes. Without first defining single cell transcriptome characteristics (i.e. markers) of MN subtypes in control dataset, it is difficult to determine if MN subtypes from the merged dataset have the gene expression profiles similar to the control MN subtypes.

- Fig 4E: My understanding is that trajectory methods like Monocle will “force” a trajectory onto a dataset, which warrants caution when interpreting the results. For example, what would the Monocle results look like if the V2IN, V3IN, and ND clusters were included (if I understand correctly, Monocle would still connect them)? Especially because there are large gaps between the clusters on the UMAP, I think more data and/or analysis would be necessary to support the hypothesis that HMC and PGC cells pass through an MMC state (e.g. by capturing intermediate cell types or examining these “island” cells for a hybrid transcriptional character). Similarly, the claim that PGC-Isl1- cells emerge from PGC-Isl1+ cells would be better supported by adding, for example, RNA velocity analysis on PGC cells. However, Figure 4 does not seem essential to the authors’ claims about Jmjd3.

→ While the trajectory analysis provides useful information to the field of MN development, we agree that this analysis is not essential to understand the role of Jmjd3 in MN development, the main theme of our study. Thus, we revised the text to clarify that the Monocle analysis results suggest the relationships among the MN lineage cells based on the single-cell transcriptomes, but more analyses are needed to determine the developmental trajectory of MN subtypes. We removed the statements “HMC and PGC cells pass through an MMC state” and “PGC-Isl1- cells emerge from PGC-Isl1+ cells”.

It is important to note that CC8 and CC9 cells do NOT represent the fully differentiated V2IN and V3IN. To make this point clear, in the revised manuscript, we included the clustering analysis of all Olig2-Cre lineage cells (Fig. 3). Please note, among all V2IN and V3IN captured by scRNAseq, only a small subset of cells expressed Slc18a3 and included in the analysis of control MN lineage cells. For instance, 2770 cells belong to V3IN clusters in Olig2-Cre lineage cells, whereas only 144 V3IN cells were captured in Slc18a3+ MN lineage cell analysis (compare Fig. 3b and Fig. 4b). Given that CC8 and CC9 express a relatively low level of neurofilament genes and are close to nbMN cluster in hierarchical analysis (Fig. 4c, 5d), they may represent the cells that undergo neuronal differentiation while consolidating their neurotransmitter identity soon after being born from the dorsal and ventral boundaries of the pMN domain.

We performed the Monocle analyses with all clusters including the V2IN (CC8), V3IN (CC9) and ND (CC10) clusters, as the reviewer suggested (Rebuttal Fig. 4). As we do not know if V2IN, V3IN and ND clusters belong to specific spinal cord segments, we included them in both thoracic and brachial level analyses. In brachial analysis, CC8 and CC9

cells connect pMN and nbMN cells (Rebuttal Fig. 4a). In thoracic analysis, CC8 and CC9 cells were similarly located between pMN and nbMN cells but they made a separate branch. Interestingly, under this condition, CC10 (ND) cells were not connected to any of cell clusters in brachial analysis, whereas they were closely linked to HMC cells in thoracic analysis. Overall, our findings are in line with the previous studies that *Olig2* is transiently expressed in pMN-neighboring progenitor domains^{3,4} and the cross-repressive interaction among TFs in differentiating neurons, such as that between MN-specific TF *Mnx1* and V2aIN-specific TF *Vsx2*, is needed to effectively separate MN and V2IN fates^{5,6}.

- Fig 5: Why have the authors analyzed the cKO and control datasets using CCA (according to supp. methods), which would seem to force integration of the two conditions? What does the analysis look like with PCA (i.e. without integration)? Do control and cKO cells form entirely different clusters for all cell types, or do some cell types still largely overlap transcriptionally? This kind of analysis additionally seems like it might address some questions about Figure 6, below.

→ We explained why the merged dataset analysis was needed above and in the revised manuscript. In the revised manuscript, we also present the PCA results (without integration) (Supplementary Fig. 8c,d).

The major sources of batch effects arise from samples with significantly different sequencing depth and saturation, and varying sequencing instruments. A practical way to observe potential batch effects is to visualize the cell groups on a low-dimensional space using PCA plot by labelling cells by their sample group (i.e. control and knockout). When we examined the presence of batch effect by projecting two data sets into several PCA spaces (PC1-4) without integration and without applying batch correction, most of the cells from the two groups were well mixed, suggesting there are little batch effects between the two groups (Rebuttal Fig. 5a,b).

The integration procedure and batch correction in Seurat, which we applied in our analysis, are to conduct canonical correlation analysis (CCA) and then use dynamic time warping (DTW) that locally stretches and squeezes the CCA data projections in order to further align different batches. Although it is possible that CCA may lose the subspaces with the largest possible variance leading to wrong alignment result when the cell types of different batches are extremely imbalanced, it does not seem to be the case for our dataset (Rebuttal Fig. 5b-d). In separate UMAP plots of control vs *Jmjd3*-cKO cells without being affected by integration, control and *Jmjd3*-cKO cells appear to largely overlap transcriptionally especially for pMN, nbMN, V2IN, and V3IN clusters and some variations emerge as cells further differentiate into MN subtypes.

- Fig 6A: If the authors have only analyzed two 10X samples, how certain is it that these differences in cell ratio are not caused by sampling / batch effects?

→ We are confident about the cell ratio changes revealed by scRNAseq datasets. First, our scRNAseq analysis results in cell ratio is highly similar to the immunohistochemical analyses of spinal cord sections from *Jmjd3*-cKO and control embryos. The immunohistochemical analyses enabled us to define MN subtype ratios based on not only the marker gene expression but also their cell body locations and axon projection patterns (Fig. 1,2). The scRNAseq analysis allowed us to define the cell types unbiasedly based on single cell transcriptomes (Fig. 3-8). The highly consistent results between the two strong and complementary datasets are striking. Second, in the revised manuscript, we added substantial amount of new data, confirming the observation from scRNAseq analyses with the immunohistochemical and in situ hybridization analyses of spinal cord sections from *Jmjd3*-cKO and control embryos (Fig. 8b,c, Supplementary Figure 10a,b). Third, the PCA above showed that there were minimal sampling/batch effects between control and cKO samples in our scRNAseq analyses. Last, we analyzed the second set of scRNAseq datasets from independent *Jmjd3*-cKO and control samples, and the two biological replicate experiments led to the same trend in the change of MN subtype composition in the *Jmjd3*-cKO spinal cord. We reported one set of scRNAseq results in the manuscript as this set had a better representation of all MN subtypes in both limb and thoracic levels. The other set – which we collected and analyzed first - included a smaller portion of limb levels and therefore a smaller number of LMC MNs, and we optimized the dissection for the second set, which was used in this report.

- Fig 6B: Does the authors' method control for cluster size (number of cells) and depth (UMI counts) when comparing the results of DE analyses across clusters? Perhaps larger clusters mean more statistical power, and fewer statistically DE genes are found in immature cell types because they express fewer genes in general.

→ We performed the depth (UMI counts) normalization in the total datasets, but did not control for cluster size and depth in comparing the results of DE analyses across clusters. The immature cell types expressed the comparable number of genes to mature cell types, but the number of cells for immature cell types was smaller than that of most mature cell types. Taking the reviewer's suggestion, in the revised manuscript, we removed the following description comparing the number of DEG across clusters "Overall, the number of DEG was greater in MN subtype clusters than less-differentiated *Nefm*-low clusters. MMC and LMCm showed the highest number of DEG (141 genes each), followed by

HMC (139 genes)".

Minor points:

- Fig 1A in situ results are hard to see. Is there an antisense control for comparison, or maybe arrowheads can be used to annotate better?

→ We added the schematics to Fig 1a to help understand in situ results. The new immunohistochemistry data also show the expression of Jmjd3 in developing MNs (Fig. 1b).

- Fig 1C: how many sections were analyzed?

→ 9 sections were analyzed for each bar in the graphs. This information is added to the figure legend.

- Fig 6D: Is there a way the authors could mark which genes in this plot are differentially expressed?

→ In the revised manuscript (Figure 8f), we highlighted the differentially expressed genes in this plot.

- The resolution of some of the figures seems low, especially the t-SNE plots (e.g Figure 3E-I), and the color scheme used for plotting gene expression makes it a little hard to distinguish low-expressing from non-expressing genes.

→ We improved the resolution and adjusted the dot size and color scheme in the figures. For instance, we changed the color scheme of the double positive cells from yellow to cyan to visualize them better in Figure 3f,g, and i.

Reviewer #2 (Remarks to the Author):

In this study, the authors address the question of cell fate diversification in the developing CNS using specification of MN subtypes in the developing spinal cord as a model system. Specifically, they examine the role of the histone demethylase Jmjd3 in this process by phenotypic analyzes of Jmjd3 mutant embryos and by characterizing MNs from WT and Jmjd3 mutant embryos by single cell RNA-sequencing (scRNA-seq). Based on this, the authors propose a model whereby Jmjd3 act as a co-activator of the Isl1-Lhx3 complex, thereby promoting MMC and HMC MN fate and suppressing acquisition of LMC and PGC MN identities. The study is well-performed and interesting albeit there are some significant concerns outlined below that the authors should address.

→ Thank you for the insightful comments on our work and helpful suggestions. We have addressed the concerns in the revised manuscript. Please see below for specific responses.

Major points

To examine the role of Jmjd3 in MN diversification the authors generate conditional Jmjd3 mutants in which Jmjd3 is deleted at a progenitor stage (Jmjd3cko-Olig2cre), which results in changes in the numbers of MN columnar subtypes, defects in axonal projection, and decreased expression of genes defining the different MN clusters. However, when deleting Jmjd3 in post-mitotic MN (Jmjd3cko-Isl1cre) the effects on MN cluster production are not fully recapitulated. They observe a reduction of MMC cluster (as observed with Jmjd3cko-Olig2cre) but not a corresponding increase of LMC or PGC MNs. This discrepancy of phenotypes needs to be better addressed. The authors mention that Isl1-Cre expresses the Cre recombinase at a later time in the MN lineage than Olig2-Cre, which formally is correct. However, expression of Isl1 is nevertheless initiated at an early stage when progenitors exit the cell cycle and differentiate into neurons. This indicate that Jmj3 may function already at an immature progenitor stages and via a mechanism that at least in part should be independent to the Isl1-Lhx3 complex. Thus, data from the Jmjd3cko-Isl1cre analysis is not in agreement with the author's model assigning the fate determining activity to Jmjd3 Isl1-Lhx3 complex. The authors should try to better define the temporal requirement Jmjd3 in the differentiation process, and explore potential additional functions for Jmjd3 in this differentiation lineage. One experiment that could provide insight into this would be to compare scRNA-sequencing data from Jmjd3cko-Olig2cre embryos to corresponding data generated from Jmjd3cko-Isl1cre embryos. It would also be interesting to compare MN axonal projection patterns in WT and Jmjd3cko-Olig2cre embryos with data from Jmjd3cko-Isl1cre mice.

→ It has been widely known that the expression of Cre does not guarantee the complete removal of the target gene

product. For instance, Cre-mediated recombination efficiency differs depending on the chromosomal position of the floxed target gene. It is also possible that the target gene product (i.e. protein) remains in cells after the target gene is deleted by Cre, if the target protein is relatively stable. Thus, it is important to determine the timing of the target protein removal (Jmjd3 for our study) in mutant mice. After having tested many different kinds of Jmjd3 antibodies, we finally identified Jmjd3 antibody that works for immunostaining and performed immunohistochemical analyses with the Jmjd3 antibody on Jmjd3cKO;Olig2-Cre and Jmjd3-cKO;Isl1-Cre mouse sections. These new data are included in the revised manuscript.

Jmjd3 was highly expressed in embryonic MNs, consistent with our in situ hybridization results (Fig. 1a,b). Jmjd3 expression was mostly eliminated by E11.5 in Jmjd3cKO;Olig2-Cre mice, whereas Jmjd3 expression persisted in MNs of E11.5 Jmjd3-cKO;Isl1-Cre mice (compare Fig. 1b and 9b). Jmjd3 expression was markedly reduced in Jmjd3-cKO;Isl1-Cre compared to the littermate control at E12.5 (Fig. 9b). Together, our immunostaining data suggest that Jmjd3 expression was removed in nbMNs of Jmjd3cKO;Olig2-Cre mice, but Jmjd3 protein was present in nbMNs of Jmjd3-cKO;Isl1-Cre mice. This explains the phenotypic discrepancy between Jmjd3cKO;Olig2-Cre and Jmjd3-cKO;Isl1-Cre mice in our study. The removal of Jmjd3 protein in Jmjd3-cKO;Isl1-Cre mice occurred after MNs differentiated and begun to acquire MN subtype identity. A low level of Jmjd3 expression in progenitors and an induction of Jmjd3 expression in differentiating MNs support the possibility that Jmjd3 functions during MN differentiation rather than at a progenitor stage. Together, the new immunostaining results provided the insights into the timing of Jmjd3 inactivation in Jmjd3-cKO;Isl1-Cre mice, and overall support our model of the cooperative action between Jmjd3 and Isl1-Lhx3 in developing MNs. Of note, we do not claim that Jmjd3 function only via Isl1-Lhx3. It is possible that Jmjd3 has Isl1-Lhx3-independent function in MN development, which will be an interesting subject for future investigation, as we stated in the manuscript.

To analyze the motor axon projection pattern in Jmjd3-cKO;Isl1-Cre mice, we created Jmjd3-cKO;Isl1-Cre mice that carried the mTmG reporter allele using the parallel genetic condition for the motor axon analyses for Jmjd3-cKO;Olig2-Cre mice through extensive breeding. However, as Isl1-Cre is active in sensory neurons in the dorsal root ganglion in addition to spinal MNs, the GFP+ axonal trajectory labeled both sensory and motor axon projections (Rebuttal Fig. 6) and therefore these genetic schemes could not be used for analyzing Jmjd3-cKO;Isl1-Cre mice. Neurofilament staining could not be used for motor axon navigation in Jmjd3-cKO;Isl1-Cre mice either, as neurofilament stains both motor and sensory axons. However, we were able to add substantial amount of new data for Jmjd3-cKO;Isl1-Cre mice in the revised manuscript (Fig. 9b-f). These results enabled us to compare the effect of Jmjd3 removal at earlier (Olig2-Cre) vs later (Isl1-Cre) stage of MN development.

The authors provide nice data set of scRNA-seq data from WT cells which allows the authors to identify the different MN clusters and describe developmental trajectories. However, when analyzing cells from Jmjd3cKO-Olig2cre mutant embryos, instead of making a projection into the clustering analysis of WT cells, the authors pooled both wt and mutant cells and performed a new clustering analysis. It is not clear to me why the authors did this. The authors should also provide a projection of mutant cells into the 2D coordinate space defined/trained by WT cells (e.g. using UMAP's transform feature) to allow readers to better understand the transcriptional shifts and distribution of MN cluster identities in Jmjd3 mutants. By pooling mutant and WT cells together, the authors risk adulterating WT cell classifications through dilution with KO transcriptional signatures. Additionally, the authors should provide a developmental trajectory analysis for the mutant cells as performed for WT cells (Fig-4E). In their analysis of WT cells, there is a bifurcation at the nbMN level that results in a MMC and LMC differentiation branch at branchial level or in a MMC and HMC/PGC differentiation branch at thoracic level. It would be interesting to know, and possibly provide insights into the function of Jmjd3, how these developmental trajectories are maintained or changed in mutant cells.

→ Taking the advantage of single cell transcriptome datasets, we wanted to identify the comparable MN subtype cells from control and *Jmjd3*-cKO samples, which would enable us to analyze the gene expression changes in comparable MN subtype cells between control and *Jmjd3*-cKO samples, rather than checking the gene expression changes in the whole MN population. The MNs are not homogenous cell population, and if we analyze the gene expression changes only as the whole MN population, we will miss the critical gene expression shift occurring only in specific MN subtypes. When we performed the clustering analysis with cKO sample only, not all MN subtypes were identified. In the revised manuscript, we provide the clustering analysis of *Jmjd3*-cKO sample alone in Fig 6 and Supplementary Fig. 8. The clustering of *Jmjd3*-cKO only sample showed an increased number of LMCI and PGC-*Isl1*⁻ MNs, but it failed to identify cell clusters with clear MMC or HMC identities, likely because MMC and HMC transcriptome identity was eroded in the absence of *Jmjd3* and did not emerge as separate clusters when only *Jmjd3*-cKO cells were analyzed for clustering. Thus, we could not compare the gene expression profile between *Jmjd3*-deficient presumptive MMC/HMC cells and control MMC/HMC from the separate clustering data. We reasoned that if those *Jmjd3*-deficient MMC/HMC MNs maintain the gene expression profiles resembling their counterparts in controls, they may cluster together with their counterparts when analyzed as a merged dataset (control plus cKO samples). To test this idea, we performed the clustering analysis of the merged dataset. The analysis of the merged dataset enabled us to compare the cell number ratio and gene expression profiles between *Jmjd3*-cKO and control cells who have overall similar transcriptome profiles to be clustered together (Figure 7,8).

We agree with the reviewer that “By pooling mutant and WT cells together, the authors risk adulterating WT cell classifications through dilution with KO transcriptional signatures.” Thus, to determine if the clusters from the merged datasets are objectively similar to MN subtypes in control dataset, we used the “matching scores” (Figure 7c). We do *not* claim that the transcriptome signature of the merged clusters is same with the that of the control clusters. We used the transcriptome signatures of control clusters to suggest the marker sets for MN subtypes (Figure 4). Then we employed these marker sets from control clusters to match the cell types of the merged clusters. In the revised manuscript, we also provided the data demonstrating that there is negligible batch effects between control and *Jmjd3*-cKO samples, making the merged data analysis possible. The major sources of batch effects arise from samples with significantly different sequencing depth and saturation, and varying sequencing instruments. A practical way to observe potential batch effects is to visualize the cell groups on a low-dimensional space using PCA plot by labelling cells by their sample group (i.e., control and knockout). When we examined the presence of batch effect by projecting two data sets into several PCA spaces (PC1-4) without integration and without applying batch correction, most of the cells from the two groups were well mixed, suggesting there are little batch effects between the two groups (Rebuttal Fig. 5a,b).

The integration procedure and batch correction in Seurat, which we applied in our analysis, are to conduct canonical correlation analysis (CCA) and then use dynamic time warping (DTW) that locally stretches and squeezes the CCA data projections in order to further align different batches. Although it is possible that CCA may lose the subspaces with the largest possible variance leading to wrong alignment results when the cell types of different batches are extremely imbalanced, it does not seem to be the case for our dataset (Rebuttal Fig. 5b-d). In separate UMAP plots of control vs *Jmjd3*-cKO cells without being affected by integration, control and *Jmjd3*-cKO cells appear to largely overlap transcriptionally especially for pMN, nbMN, V2IN, and V3IN clusters and some variations emerge as cells further differentiate into MN subtypes (Rebuttal Fig. 5c,d). We also projected both control and *Jmjd3*-cKO cells into the same

UMAP feature space without Seurat (CCA) integration, which presents the locations of mutant cells relative to control cells in the 2D space (Rebuttal Fig. 7). This analysis shows that control and *Jmjd3*-cKO cells in pMN, nbMN, V2IN, and V3IN clusters are relatively well intermingled, while MN subtype cells tend to show more segregated patterns between control and *Jmjd3*-cKO cells, consistent with other analysis results.

We performed a developmental trajectory analysis for the mutant cells as performed for WT cells (Rebuttal Fig. 8). The cell types were defined based on the clustering analysis of *Jmjd3*-cKO cells alone (Fig. 6a), and thus this data should be interpreted with caution because the clustering analysis with *Jmjd3*-cKO cells alone did not find clusters with clear MMC/HMC identities and instead had a cluster of possible presumptive MMC/HMC cells, as described in the revised manuscript. In this analysis, the trajectory of pMN-nbMN-presumptive MMC/HMC in *Jmjd3*-cKO sample resembled the trajectory of pMN-nbMN-MMC in control sample. Interestingly, however, in *Jmjd3*-cKO sample, *Isl1*-negative LMC and PGC population (LMCI, PGC-*Isl1*⁻ in brachial and thoracic levels, respectively) emerged first and then transitioned to *Isl1*-positive LMC and PGC populations (LMCm and PGC-*Isl1*⁺ in brachial and thoracic levels, respectively).

The analysis of scRNA-seq provided a more refined overview of the changes on MN clusters observed in *Jmjd3*cko-Olig2cre mutants (e.g., increase of LMCI and decrease of LMCm/LMCd; increase of PGC-*Isl1*⁻ and decrease of PGC-*Isl1*⁺). The authors should validate these results in sections of mutant spinal cords.

→ As suggested, we did validation experiments in sections of *Jmjd3*-cKO spinal cords. The quantification of MN subtypes demonstrated that, in *Jmjd3*-cKO spinal cords, LMCI and PGC-*Isl1*⁻ increased, whereas LMCm, PGC-*Isl1*⁺, MMC and HMC significantly reduced (Figure 8b). We also performed in situ hybridization assays with a newly defined HMC marker *Klf5* and confirmed a drastic reduction of *Klf5*⁺ HMC in *Jmjd3*-cKO (Fig. 8c). Our new immunostaining data also showed a substantial reduction of *Mecom*⁺ MMC MNs (Fig. 8c), further validating scRNAseq results.

The authors propose that *Jmjd3* is recruited to *Isl1*-Lhx3-bound target genes and activate their transcription. They show by ChIP-PCR in spinal cord tissue the recruitment of *Jmjd3* to targets of *Isl1*-Lhx3 complex, interaction of *Jmjd3* to Lhx3 but not *Isl1*, and an increase of reporter activity in luciferase assays. While *Jmjd3* can interact with Lhx3 in IP assays, the authors propose that *Jmjd3* interacts with the *Isl1*-Lhx3 complex to regulate gene expression. This would be better supported if the authors could provide data establishing the presence of *Jmjd3* in the *Isl1*-Lhx3 complex. Additionally, as *Jmjd3* is a histone demethylase as highlighted by the authors in their title. The authors should provide more direct evidence that *Jmjd3* regulates transcription of target MN genes via its histone demethylase activity.

→ *Jmjd3* did not co-immunopurify *Isl1*, while interacting with Lhx3. To test if Lhx3, which forms a complex with *Isl1*, can mediate the association between *Jmjd3* and *Isl1*, we performed the coIP experiments in cells co-expressing *Isl1*, Lhx3 and *Jmjd3*. Interestingly, when Lhx3 was co-expressed, *Jmjd3* pulled down both *Isl1* and Lhx3 (Figure 10b). This new data supports the model that *Jmjd3* associates with the *Isl1*-Lhx3 complex.

The best-known function of *Jmjd3* is a histone demethylase activity, as evident by its another *KDM6b* (K for histone lysine residue, DM for DeMethylase), and thus we used this name in the title. However, we do not claim that *Jmjd3* regulates transcription of target MN genes via its histone demethylase activity.

Minor points

In Figure 1 the authors should count the total number of MN in control vs Olig2-Cre::*Jmjd3*fl/fl embryos in order to show

that the loss of MN subtypes in mutant embryos is not also due to cell loss. The measurement of Slc18a3 expression area may not be appropriate considering expression of Slc18a3 also in V3 and V2 lns

→ As suggested, we quantified the total number of MNs. The quantification showed no significant difference in the total number of MNs between control and Jmjd3-cKO embryos, suggesting that cell loss is not a main mechanism underlying a loss of MN subtypes in mutant embryos (Fig. 8b). Consistently, our new data also showed that there was no significantly increased apoptosis in Jmjd3-null MNs, as monitored by the TUNEL assays (Supplemental Figure 10c).

Authors argue for an increase of PGC-Nos1 MNs in mutant spinal cords, but from the data presented (FigS1B, 1B), there seems to be a reduction. At E13.5 the signal of Nos1 is present but it is difficult to know if there is a change in numbers. In the scRNA-seq analysis the authors identify 2 PGC clusters (Fig3C). The authors argue that both clusters are Nos1+, but their data actually argues the opposite, one cluster seems to be FoxP1+/Zeb2+/Nos1+/Isl1+ (cluster CC6) and the other FoxP1+/Zeb2+/Nos1-/Isl1- (cluster CC7). The authors should quantify the number of both clusters in the mutants.

→ As Nos1 becomes significantly upregulated as PGC MNs mature between E12.5 and E13.5, Foxp1 (or Foxp1/Zeb2 co-expression) is a better marker of PGC than Nos1 in E12.5 thoracic spinal cord. As noted by the reviewer, the Nos1 expression level is higher in CC6 (PGC-Isl1+) than CC7 (PGC-Isl1-). However, the average Nos1 level in CC7 is higher than in non-PGC MN clusters (Fig. 4j, also see Rebuttal Fig. 9a), raising the possibility that the full-level induction of Nos1 in CC7 (PGC-Isl1-) occurs later than that in CC6 (PGC-Isl1+). Based on the immunohistochemistry data (Fig. 4k), we predict that both PGC-Isl1⁺ and PGC-Isl1⁻ cells would express Nos1 by E13.5. Together, the clear co-expression pattern of Foxp1 and Zeb2 in CC6 and CC7 at E12.5 indicate that both clusters are PGC MNs.

To validate a change in PGC MNs in Jmjd3-cKO spinal cord from scRNAseq analysis, we performed the quantification analysis in spinal cord sections (Figure 8b). Of note, PGC MNs are much more readily identifiable in the spinal cord sections than in scRNAseq datasets because spinal cord sections provide positional information for MNs (the spinal cord segment – brachial or thoracic, and unique intermediate location of PGC MNs, as opposed to the ventro-lateral location of other MN subtypes, in the transverse sections of the spinal cords). Our quantification validated the findings in scRNAseq analyses that, in Jmjd3-cKO spinal cord, PGC-Isl1⁻ MNs increased and PGC-Isl1⁺ MNs decreased. When combined, the total number of PGC increased in Jmjd3-cKO spinal cord compared to the control spinal cord.

As noted by the reviewer, it is difficult to quantify Nos1⁺ cell numbers because Nos1 is expressed in cytosolic compartment of the cells, labeling not only soma but also dendrites and axons (Supplementary Fig. 10a). Thus, the identification and quantification of PGC based on nuclear marker expression and the cell body position produce more reliable data.

It would be interesting to see the distribution of Jmjd3 expression in the different clusters identified in wt cells, similarly to what the authors show in Fig.3B.C.

→ We included the distribution of Jmjd3 expression in the revised manuscript (Supplementary Figure 5b,c, also see Rebuttal Fig. 9b,c).

The authors should update the nomenclature used for protein and genes names.

→ We updated the nomenclature in the revised manuscript.

Jmjd3 → Kdm6b

Utx → Kdm6a

Uty → Kdm6c

NLI → Ldb

Reviewer #3 (Remarks to the Author):

In this extensive manuscript Wang et al characterize the role of the H3K27 demethylase Jmjd3 in developing motor neurons. The authors provide evidence that Jmjd3 promotes the formation of specific MN subtypes at the expense of others. Wang et al propose that this involves the MN specifying transcription factors Isl1 and Lhx3. The authors also report single cell transcriptome data from MNs.

Understanding the basis of MN subtype identity is an interesting and well explored question. The current study provides a new angle to this question. The study also provides a useful dataset of single cell mRNA sequencing of MNs.

A difficulty with this manuscript is that it has the feel of two studies that have been merged. On the one hand the authors have undertaken a single cell transcriptome analysis of motor neurons from e12.5 mouse embryos. Distinct from this, the authors have characterized a role for Jmjd3 in MN subtype identity. It might be easier for readers if these two studies were presented separately.

→ Thank you for acknowledging the significance of our study. We agree with the reviewer that it would be easier for readers if the two studies were presented separately.

The core subject of our study is to understand the role of Jmjd3 in MN development, and scRNAseq was a highly useful tool to this aim. However, there was no prior investigation of single cell transcriptomes of developing MNs to the scale that we presented here. Thus, it would be highly beneficial to the field if we explain the findings from the single cell transcriptome studies of control MNs, and this also sets the foundation, based on which Jmjd3-cKO transcriptome changes can be explained.

Specific points

-The authors show (Fig 1A) that Jmjd3 is expressed in all neurons but only comment on MNs. It should be clearer to a reader that Jmjd3 is not specific to MNs. Related to this the authors conclude that the perinatal death of mice with Jmjd3 knocked out using Olig2-cre suggests a role of jmj3 in MN development. However, the cause of death could relate to another function of Jmjd3 in Olig2 expressing cells.

→ We revised the text to clarify that 1) Jmjd3 expression is not specific to MNs, and 2) the perinatal death of Jmjd3-cKO:Olig2-Cre mice suggest an important role of Jmjd3 in Olig2-Cre lineage cells, which include but are not limited to MNs.

Related to the perinatal phenotype of Jmjd3-cKO, two points are noteworthy. First, our newly provided immunostaining assays with Jmjd3 antibody show that Jmjd3 protein is most highly expressed in MNs in the developing spinal cord (Fig. 1b). Second, among the most noted Olig2 lineage cells in the CNS, which include cortical neurons, myelinating oligodendrocytes, MNs and spinal interneurons, perinatal death had been mostly associated with MN deficits. The defects of myelinating oligodendrocytes do not cause perinatal death as the myelination occurs after birth in mice, for instance. More studies are needed to determine the precise cause of death of Jmjd3-cKO:Olig2-Cre mice.

- Since the Olig2-cre will also delete Jmjd3 in V3 neurons, ventral to MNs, and some V2 neurons, dorsal to MNs, the authors should also examine whether there is an effect on these neuronal populations.

→ As suggested, we analyzed V2 and V3 INs in Jmjd3-cKO mice, and there was no significant change in Jmjd3-cKO mice (Supplementary Figure 1c,d).

- Do the increases in LMC and PGC MNs account for the decreases in MMC and HMC? I accept this analysis is not straightforward but some indication of whether the data could be explained solely by a change in identity would be helpful.

→ As suggested, we performed the quantification of MN subtypes using combinations of markers in serial sections of the spinal cord (Figure 8b). At brachial levels, the increases in LMC account for the decrease in MMC. At thoracic levels, the increases in PGC almost fully account for the reduction of MMC and HMC. These results support the notion of unbalanced MN subtype production in the absence of Jmjd3; in the absence of Jmjd3, LMC and PGC MNs, particularly LMCI and PGC-Isl1- MNs, increase at the expense of MMC and HMC. Of note, there was not significant induction of apoptosis in Jmjd3-cKO (Supplementary Figure 10c), and thus the cell death is unlikely the main contributor to the reduced MMC and HMC.

-The single cell transcriptome data (Fig 3) was generated by selecting cells expressing the cholinergic marker Slc18a3. This is expressed by post-mitotic neurons. It is surprising, therefore, that the authors identify clusters of cells that appear to have the gene expression profile of undifferentiated progenitors (expressing Sox2 and Hes5). Some explanation for this should be offered. Related to this the cells were isolated from e12.5 embryos, which is relatively late in the differentiation window of MNs. This raises the possibility that the Sox2/Hes5 cells might not be MN progenitors and this would mean the developmental trajectory proposed in Fig 4 is erroneous.

→ In the original manuscript, we did not include the analysis of all Olig2-Cre lineage cells, and presented only the analysis of Slc18a3⁺ cells to keep the paper focused on MN development. However, we realized that this omission raised the question that the reviewer asked. To address this point, we included the analysis of all Olig2-Cre lineage control cells in the revised manuscript (Fig. 3).

We selected Slc18a3⁺ cells among single cells sequenced to identify the cells that are committed to the MN lineage. The Slc18a3 gene is induced as progenitors differentiate to MNs, and some of these differentiating “progenitors” (immediate daughters of actively cycling, typical progenitors) could still have remaining transcripts of progenitor markers, such as Sox2 and Hes5, creating a transient population of the cells that have mRNAs for Slc18a3, Sox2 and Hes5, although their levels in the differentiating “progenitors” are not as high as the Sox2 and Hes5 levels in “actively cycling, typical progenitors”. We labeled these cells as pMN cluster (CC0, Fig. 4a) for the sake of clarity when we performed the clustering analyses only with Slc18a3⁺ cells, but CC0 cells are the cells transitioning from pMN progenitor identity to nbMNs, rather than “actively cycling, typical progenitors”. Please note that Slc18a3 expression level in CC0 (pMN) is substantially lower than Slc18a3 levels in MN subtypes (CC1-CC7) (Fig. 4c), suggesting that Slc18a3 level just begun to rise.

Then, where are “actively cycling, typical progenitors”? They were not included in our clustering analyses of Slc18a3⁺ cells, because they do not express Slc18a3, as the reviewer assumed. This would be best illustrated in the context of all sequenced single cells that contain all Olig2-Cre active cell-lineage cells (Fig. 3). 8,815 single cells were sequenced in the control sample containing all GFP⁺ sorted cells. The clustering analysis of 8,815 single cells identified 13 clusters (cluster OC0-OC12). Among those, the cells in five clusters expressed Sox2 and Hes5; OC0, OC7 and OC9 had a high level of Sox2 and Hes5, and OC6 and OC12 had a medium level of Sox2 and Hes5 (Fig. 3a-c, Supplementary Fig. 4b). Consistently, the neurofilament gene Nefm expression was remarkably low in OC0, OC6, OC7, and OC9 (Fig. 3a,b,d, Supplementary Fig. 4b). Nefm levels in C12 cells were higher than OC0, OC6, OC7, and OC9, but lower than MN clusters (OC2, OC4, OC5, and OC11). OC0 and a subset of OC7 cells showed the pMN identity based on the expression profile of progenitor markers (Fig. 3a,b,e, Supplementary Fig. 4b,d). OC0 cluster alone consists of 1059 cells. Among Slc18a3⁺ cells, the cell number of CC0 (pMN, Sox2/Hes5⁺) is only 98. Thus, in the context of all sequenced Olig2-Cre active cell-lineage cells, only 98 cells out of >1059 pMN progenitors expressed Slc18a3. Together, these data suggest that CC0 (pMN) cells in Figure 4 are the cells transitioning from pMN progenitor identity to nbMNs, rather than “actively cycling, typical progenitors”, and they began to induce Slc18a3 expression and thus Slc18a3 level in CC0 was substantially lower than Slc18a3 levels in MN clusters.

Given these observations that CC0 (pMN) cells show a possible commitment to the MN fate (i.e., Slc18a3 induction), we think it is reasonable to include Sox2/Hes5/Slc18a3⁺ cells (CC0) for the developmental trajectory analysis, although the limitation needs to be noted, as described below.

Of note, we performed the Monocle analysis without CC0 (pMN) cluster, and the overall trajectory pattern remains similar (Rebuttal Fig. 10).

-The interpretation that all of the cells profiled in the analysis are MNs relies on the specificity of expression of Slc18a3. This caveat should be noted and the possibility that Slc18a3 is expressed in some subtypes of interneurons (eg. V2 neurons) should be considered.

→ This was included in the revised manuscript. The following sentence in the Discussion is also related to this point “CC8 and CC9 cells may also contribute to the heterogeneity of interneurons in the mature spinal cord. In this regard, it is noteworthy that a subpopulation of V2aIN in the adult spinal cord co-express glutamatergic and cholinergic neurotransmitter genes^{7,8}.”

-While the trajectory analysis (Fig 4) is consistent with assumptions in the field, the limitations should be noted. As indicated above, using e12.5 progenitors to anchor the origin of MNs that were generated a day or two earlier is potentially problematic. The inferred relationships between cell types indicates similarities in gene expression, but does not necessarily relate directly to the lineage relationships. To determine lineage, fate mapping experiments would be necessary.

→ The text was revised to incorporate this input from the reviewer.

As the MNs are still being born and MN subtype identity is being shaped in E12.5 spinal cords that also contain fully differentiated MN subtypes, E12.5 MN lineage cells have both MN cells in transition steps and MN cells with mature characteristics, which made the trajectory analysis possible. As the reviewer noted, however, in our analysis, the relationships among cell types were inferred based on the gene expression profiles from E12.5 MN lineage cells and do not necessarily relate directly to the lineage relationships. We noted the limitation of our trajectory analysis in the revised manuscript. Future studies will be needed to further refine the trajectory of MN lineage cells.

-The late inactivation of Jmjd3 (with Isl1-cre) is a nice experiment to determine when Jmjd3 is necessary for its activity. It would be interesting to know the functional consequences of this experiment in the mutant animals. Does the Isl1-cre mediated deletion of Jmjd3 have an effect on MN axon navigation or muscle innervation? Does the deletion result in functional defects?

→ Jmjd3-cKO;Isl1-Cre mice show perinatal lethality. We found that Jmjd3-cKO;Isl1-Cre mice die by P1, indicating that Jmjd3 plays an essential role in Isl1-Cre lineage cells.

To analyze MN axon navigation and muscle innervation in Jmjd3-cKO;Isl1-Cre mice, we created Jmjd3-cKO;Isl1-Cre mice that carry the mTmG reporter allele using the parallel genetic paradigm for the motor axon analyses for Jmjd3-cKO;Olig2-Cre mice (Fig. 2). However, as Isl1-Cre is active in sensory neurons in the dorsal root ganglion in addition to spinal MNs, the GFP⁺ axonal trajectory labeled both sensory and motor axon projections (Rebuttal Figure 6) and therefore these genetic schemes could not be used for analyzing Jmjd3-cKO;Isl1-Cre mice. Neurofilament staining could not be used for motor axon navigation in Jmjd3-cKO;Isl1-Cre mice either, as Neurofilament stains both motor and sensory axons.

To better understand the consequences of the late inactivation of Jmjd3 with Isl1-Cre, we added new data for Jmjd3-cKO;Isl1-Cre mice in the revised manuscript. We show that, using immunostaining analysis with Jmjd3 antibody, the removal of Jmjd3 protein in MNs occurs relatively late in MN development, after E11.5 (Figure 9b). We show that the expression of Mnx1, a target of Isl1-Lhx3, reduced in Jmjd3-cKO;Isl1-Cre mice (Fig. 9e,g). No remarkable change was observed with pan-MN marker Slc18a3 and non-MMC MN markers, Foxp1 (LMC, PGC), Aldh1a2 (LMC), Kcnip4 (PGC) and Klf5 (HMC), in Jmjd3-cKO;Isl1-Cre mice (Figure 9e-f).

References

- 1 McGinnis, C. S., Murrow, L. M. & Gartner, Z. J. DoubletFinder: Doublet Detection in Single-Cell RNA Sequencing Data Using Artificial Nearest Neighbors. *Cell Syst* **8**, 329-337.e324, doi:10.1016/j.cels.2019.03.003 (2019).
- 2 Lun, A. T. L. *Detecting doublets in single-cell RNA-seq data*, (2019).
- 3 Dessaud, E. *et al.* Interpretation of the sonic hedgehog morphogen gradient by a temporal adaptation mechanism. *Nature* **450**, 717-720, doi:10.1038/nature06347 (2007).
- 4 Kaltezioti, V. *et al.* Prox1 regulates Olig2 expression to modulate binary fate decisions in spinal cord neurons. *J Neurosci* **34**, 15816-15831, doi:10.1523/JNEUROSCI.1865-14.2014 (2014).
- 5 Clovis, Y. M. *et al.* Chx10 Consolidates V2a Interneuron Identity through Two Distinct Gene Repression Modes.

- Cell Rep* **16**, 1642-1652, doi:10.1016/j.celrep.2016.06.100 (2016).
- 6 Lee, S. *et al.* A regulatory network to segregate the identity of neuronal subtypes. *Dev Cell* **14**, 877-889, doi:10.1016/j.devcel.2008.03.021 (2008).
- 7 Pedroni, A. & Ampatzis, K. Large-Scale Analysis of the Diversity and Complexity of the Adult Spinal Cord Neurotransmitter Typology. *iScience* **19**, 1189-1201, doi:10.1016/j.isci.2019.09.010 (2019).
- 8 Zagoraïou, L. *et al.* A cluster of cholinergic premotor interneurons modulates mouse locomotor activity. *Neuron* **64**, 645-662, doi:10.1016/j.neuron.2009.10.017 (2009).

Reviewers' Comments:

Reviewer #1:

Remarks to the Author:

Thank you to the authors for their detailed responses. The revised manuscript presents a nice single-cell dataset and shows that Kdm6b plays a role in motor neuron diversification.

Based on Rebuttal Figure 7, I would argue that Seurat integration should not be used for the merged analysis. "Squeezing" the control and cKO data together hides interesting and important similarities and differences between cell types in the knockout and the control. For example:

1. The authors state that the cKO contains no clear MMC clusters, but in Rebuttal Figure 7 some of the cKO cells labeled nbMN clearly mix with control MMC cells on the embedding. Are these cKO cells actually MMC? Or are some of the control MMC cells mislabeled? This discrepancy seems relevant to the cell ratio analysis in Figure 8a.

The MMC cluster also seems more irregularly shaped on the UMAP than I would have expected, suggesting heterogeneity.

2. Some types clearly split into multiple subtypes on the UMAP. For example, PGC-Is11+ (maybe three groups, one of which mixes with "ND" cells) and LMCI cells. Some cKO LMCI cells are similar to control LMCI cells (left on the UMAP), while others clearly separate (right on the UMAP). Can anything be said about these different populations? How might they affect the cell ratio / differential expression analysis? These subtleties are lost in the Seurat integration.

In my opinion, Rebuttal figure 7 more intuitively represents the differences between cKO and control than the integrated analysis + differential-expression testing used in the manuscript. The embedding nicely shows (as the authors point out) that V2IN, V3IN, and pMN cells are intermingled across cKO and control, whereas other cell types diverge, particularly the MMC, HMC, LMC cells.

A few other minor points:

- The information the authors provide in the rebuttal about the experimental design should probably be added to the methods (e.g. how many biological replicates contribute to the data)
- The QC panels are a little difficult to read; the text is small, and it's hard to appreciate subtle differences in the purple. Perhaps a different colormap can be used.
- I found it a little difficult to assess the DoubletFinder results without seeing the scores of the individual cells. Based on the table, I agree with the authors' assessment of their results, but point out that the best cutoff score to use for identifying putative doublets probably should be assessed by looking at the full distribution (histogram) of the cells' doublet scores. And if the distribution reveals clear outliers, you can remove these cells before further analysis.

Reviewer #2:

Remarks to the Author:

I appreciate the effort the authors made to address my comments and those of the other reviewers. While the authors address most of my comments there are still some minor remaining issues the authors could address.

An important conceptual point of this manuscript, which is extensively discussed by the authors, is the definition of Kdm6b activity in MN subtype specification through Kdm6b acting as a co-activator of Is11-Lhx3 complex. However, the changes in MN subtype generation when Kdm6b is deleted only in post-mitotic MN (Kdm6b-cKO;Is11-cre) does not recapitulate mutants with deletion at progenitor stages (Kdm6b-cKO;Olig2-cre) weaken the model proposed by the authors. In their revised manuscript the authors provide immunocytochemistry data for Kdm6b expression suggesting for a delayed loss of Kdm6b expression in MN of Kdm6b-cKO;Is11-cre mutants (Fig 1b and Fig9b), which could account for the difference of phenotype between the two mutants. Regarding this there are certain remaining concerns:

1. From the data presented by the authors it is difficult to access the level of Kdm6b expression remaining in MNs. Data on the Kdm6b-cko;Isl1-cre (Figure 9b) is not very clear as the signal both for controls and mutants is very weak. Can the authors provide better images? There seems to be some ventral signal in the mutants at E11.5 and E12.5. Is this expression in MNs? Are the remaining Kdm6b cells in the mutants Mnx1+ or Isl1+?

2. Due to the late loss of Kdm6b in the MN pool as revealed by the new data presented by the authors (between E11.5-E12.5), the authors should analyse switching of MN subtypes at stages later than E12.5 (maybe E13.5/E14.5). This would help to understand if the difference of phenotype relative to Kdm6b-cko;Olig2-cre mutants reflects just a temporal delay of protein loss or a change of plasticity of differentiating MN. This is an important point that is also raised by the authors in the discussion section that state (line 616): "Our data also suggest that MNs lose the plasticity in switching among different MN subtypes as they acquire terminally differentiated state."

3. In Kdm6b-cko;Olig2-cre mutants how is the expression of Kdm6b in MN at E9.5 and E10.5 embryos? Is expression of Lhx3 reduced/lost in MNs at E10.5? Is there a premature induction of Foxp1 in MNs at E10.5?

In Fig.8b the authors should precisely define the combination of markers used to quantify different MN subtypes and how the total number of MNs were calculated.

In Figures 4a and 7a the authors maintain consistency of cluster numbering and colour coding, but not in Fig.6a. The authors should reformat Fig. 6a to maintain consistency.

In the discussion section where the authors discuss MN differentiation and acquisition of MN subtypes it is not clear why Lhx3 would become downregulated with time, considering that Kdm6b is expressed in the entire MN population, which would reinforce/maintain Lhx3/Isl1 expression via Mnx1.

Fig. 1b the authors should update the mutant nomenclature.

Line 338: typo. The authors may mean "Kdm6b-null MNs" instead of "Kdm6a-null MNs".

Reviewer #4:

Remarks to the Author:

Point 1:

\ The authors have addressed this concern and the co-presentation of single cell sequencing and histological analyses are interesting and strengthen each other.

Point 2:

\ The authors have addressed this concern.

Point 3:

\ This analysis is a bit superficial, as the authors could also perform comparative analysis in the sequencing data. While it would extend the relevance of their findings by including interneuron analysis, it is not central to the main claims of this manuscript.

Point 4:

\ The authors have addressed this concern.

Point 5:

\ The authors have not fully addressed this concern. As stated, e12.5 is quite a late timepoint to see such early MN progenitors and while the author's explanation is plausible – a hybrid state of late progenitors/early MN that co-express markers for both, this could easily be tested directly with in situ hybridization in tissue sections. As the authors also observe some glial progenitor markers, it is particularly important to determine the identity of cells used as the anchor for Monocle analysis. In addition, the authors overstate conclusions from the lineage analysis, which can be used to find "trajectories" through any set of cells/clusters. Given that this is from single replicate data from a single timepoint, the authors should scale back their claims about developmental lineage.

Point 6:

\ The authors have addressed this concern, however they should also note that this could reflect

doublets. While they have performed some doublet analysis, they correctly state above that this is difficult to interpret during developmental time-points. "Barnyard" analysis of MN markers vs V2/V3 markers on a cell by cell basis could help to resolve this, though this is not central to the main claims of the manuscript and is not necessary.

Point 7:

\ Please see the comments above regarding Slc18a3. This work is not conclusive and the authors should scale back their conclusions about lineage or remove this analysis.

Point 8:

\ It is appreciated that the authors attempted this and they have addressed this concern.

Misc. Small Comments

\ The authors should edit the text in line 328 to specify that a single sample was used for each condition. In addition, they should add a sentence to specify that the single cell sequencing analysis was considered exploratory (because only one replicate was used) but key findings were validated with independent means (eg. antibody staining).

\ The authors should present a merged table of Fig. 4b, 6b, and possibly Fig. 7 to facilitate comparison of cell numbers in each phase of the analysis/condition.

REVIEWER COMMENTS

Reviewer #1 (Remarks to the Author):

Thank you to the authors for their detailed responses. The revised manuscript presents a nice single-cell dataset and shows that *Kdm6b* plays a role in motor neuron diversification.

→ Thank you for your kind comments.

Based on Rebuttal Figure 7, I would argue that Seurat integration should not be used for the merged analysis. “Squeezing” the control and cKO data together hides interesting and important similarities and differences between cell types in the knockout and the control. For example:

1. The authors state that the cKO contains no clear MMC clusters, but in Rebuttal Figure 7, some of the cKO cells labeled nbMN clearly mix with control MMC cells on the embedding. Are these cKO cells actually MMC? Or are some of the control MMC cells mislabeled? This discrepancy seems relevant to the cell ratio analysis in Figure 8a.

The MMC cluster also seems more irregularly shaped on the UMAP than I would have expected, suggesting heterogeneity.

→ Thank you for the question.

Our data suggest that the identity of specific MN lineage cell types is eroded in *Kdm6b*-cKO, rather than *Kdm6b*-cKO cells completely switching from one cell type to another cell type. Thus, many *Kdm6b*-cKO cells show the transcriptomes similar but not completely matched to the transcriptomes of control cell types. As many cKO cells do not acquire a well-defined transcriptome of MN lineage cell types, the exact number for each cell type in cKO sample can show variations depending on the analysis method, although overall trends may hold. Thus, it is vital to validate scRNAseq results with independent methods, as we have done with immunostaining and motor axon projection analyses.

Regarding less defined cell identity of *Kdm6b*-cKO cells, it is notable that the transcriptomes of nbMN and MMC cells share many marker genes, such as *Lhx3/4* (Fig. 4d), *Mecom*, *Pou3f*, *Casz1*, *Rasgef1c*, and *RXRg* (Fig. 5b), while they differ by expression levels of neurogenic TFs, such as *Neurod2* (Fig. 4c). Gene expression heatmap of individual cells (columns) for marker genes of each cluster also revealed the similarity of transcriptomes between nbMN and MMC (Fig. 5a).

When we performed a separate clustering analysis with only cKO cells, no cluster with either obvious MMC or HMC identity emerged (Fig. 6a, b). Instead, we found KC2, which clearly is neither LMC nor PGC, given a lack of LMC or PGC markers. KC2 cells have a low-level *Mnx1* but do not have marker expression for MMC (e.g., *Lhx3*) or HMC (e.g., *Klf5*). Our data suggest that many genes conferring MMC and HMC identities were downregulated in cKO, and therefore MMC and HMC were not readily segregated in cKO only analysis. In addition, in the cKO alone analysis, nbMN cluster (KC3) showed less defined nbMN characteristics, such as much lower level of neurogenic TFs, like *Neurod2*, than control nbMN (compare *Neurod2* level in control nbMN CC1 Fig. 4c and cKO nbMN KC3 Fig. 6j). This suggests that some of the presumptive MMC cells in cKO sample might have been clustered together with nbMN cells, which possess similar marker genes, contributing to the lower level of *Neurod2* and other neurogenic TFs in *Kdm6b*-cKO nbMN cluster (KC3) relative to the control nbMN cluster (CC1). This would also contribute to the reviewer’s observation that some of the cKO cells labeled nbMN clearly mix with control MMC cells on the embedding and would generate less irregularly shaped MMC cluster. Also, in UMAP of Rebuttal Figure 7 (Supplementary Fig. 8b in the revised manuscript), a similar color code was used given to control MMC (CC2, Figure 4) and cKO MMC/HMC (KC2, Figure 6). Thus, the “MMC” color-coded cells in Rebuttal Figure 7 include both control MMC and cKO MMC/HMC cells. This would also

have contributed to the reviewer's observation that the MMC cluster appears to be irregularly shaped on the UMAP in Rebuttal Figure 7.

It is not straightforward to answer what are MMC in Kdm6b-cKO sample, as Kdm6b-cKO cells do NOT completely transform from one cell type to another cell type, as described above. We asked what are most likely presumptive MMC cells, which may not have acquired full-blown MMC features but have the highest similarity to control MMC at single-cell transcriptome levels. Importantly, the merged analysis of control and cKO samples was used to identify those MMC cells in cKO dataset, as cKO MMC cells would have the highest similarity to control MMC, and thus they would form an MMC cluster along with control MMC cells. Our merged analysis identified an MMC cluster that contains both control and cKO cells (MC2 in Figure 7a, b).

In the revised manuscript, we added cell ratio analysis for MN subtypes from separate data analysis (Supplementary Figure 8d) and merged data analysis (Figure 8a). Both analyses revealed a similar trend that LMCI and PGC-Is1⁻ cell types were overrepresented whereas MMC and HMC cell types (MMC+HMC for separate analyses as cKO alone analysis did not separate MMC and HMC) were underrepresented in Kdm6b-cKO sample. Most importantly, these results were validated using immunohistochemistry and in situ hybridizations with established sets of markers and motor axon projection pattern analyses.

2. Some types clearly split into multiple subtypes on the UMAP. For example, PGC-Is1⁺ (maybe three groups, one of which mixes with "ND" cells) and LMCI cells. Some cKO LMCI cells are similar to control LMCI cells (left on the UMAP), while others clearly separate (right on the UMAP). Can anything be said about these different populations? How might they affect the cell ratio / differential expression analysis? These subtleties are lost in the Seurat integration.

In my opinion, Rebuttal figure 7 more intuitively represents the differences between cKO and control than the integrated analysis + differential-expression testing used in the manuscript. The embedding nicely shows (as the authors point out) that V2IN, V3IN, and pMN cells are intermingled across cKO and control, whereas other cell types diverge, particularly the MMC, HMC, LMC cells.

→ We agree that Rebuttal Figure 7 provides an advantage in revealing subtle differences and similarities between cKO and control. The merged analysis offered different strengths, enabling us to investigate gene expression differences between cKO and control samples in the matching cell types. Differential gene expression analysis was useful to understand the mechanisms by which Kdm6b controls MN development. Therefore, to take advantage of both types of analyses, we included Rebuttal Figure 7 as Supplemental Figure 8.

Thank you for the observations regarding sub-groups of PGC-Is1⁺ and LMCI cell types. As described above, when we analyzed cKO sample only without the "anchoring effect" of control cells, the clusters emerge with less obvious MN subtype identities because cKO cells have eroded identity as MN subtypes and they do not have transcriptome distinctions among subtypes as clearly as control cells. Thus, although we attempted to match cKO clusters with control clusters as best as we could, the cell type matching between control clusters and cKO clusters, after separate unsupervised clustering, is disadvantageous as explained. A majority PGC-Is1⁺ cells are clustered, next to PGC-Is1⁻ population, as expected. Interestingly, a smaller number of PGC-Is1⁺ cells were found in two other areas, and most of the "ectopically localized" PGC-Is1⁺ cells appear to be Kdm6b-cKO cells (compare Supplementary Figure 8a and b). One group of "ectopically localized" PGC-Is1⁺ cells are intermingled with HMC, and the other group of "ectopically localized" PGC-Is1⁺ cells are mixed with ND cells. Given that cKO clusters lack HMC and ND clusters, these results raised a possibility that a subset of presumptive HMC and ND cells in cKO sample were clustered together with PGC-Is1⁺ cells in cKO alone analysis. It is also interesting to note

that cKO LMCI appears to segregate into two groups in UMAP, one group that is similar to control LMCI and the other group that are more distantly located. Given that LMCI neurons markedly increased in cKO, relative to control, it is possible that cKO LMCI consists of a typical LMCI group and atypical LMCI group transitioning from other underrepresented MN subtypes.

The cell ratio analysis in Figure 8a was performed using the cell numbers from the merged data analysis result in Figure 7a, not based on the cell number from the separate control and cKO analyses (Figure 4 and 6). In the revised manuscript, we added the cell ratio analysis for MN subtypes from separate data analysis (Supplementary Figure 8d). Both cell ratio analyses revealed a similar trend that LMCI and PGC-Is1- cell types were overrepresented whereas MMC and HMC cell types (MMC+HMC for separate analyses as cKO alone analysis did not separate MMC and HMC) were underrepresented in Kdm6b-cKO sample. While single-cell transcriptome analyses were instrumental to survey overall transcriptome profiles among > 6,000 MN lineage cells and find cell type clusters, they lack the positional information (brachial vs thoracic locations and medio-lateral and dorso-ventral positions in the ventral spinal cord). Thus, the key findings from scRNAseq analysis, such as changes of MN subtype ratios and several marker gene expressions, were independently confirmed by immunohistochemistry, in situ hybridizations, and motor axon projection pattern analyses.

--

A few other minor points:

- The information the authors provide in the rebuttal about the experimental design should probably be added to the methods (e.g., how many biological replicates contribute to the data)

→ We included the information in the methods section, including biological replicate information.

- The QC panels are a little difficult to read; the text is small, and it's hard to appreciate subtle differences in the purple. Perhaps a different colormap can be used.

→ To address this point, we increased the size and font of QC panels and also used images with higher resolution (Supplementary Figure 5a, 7b).

- I found it a little difficult to assess the DoubletFinder results without seeing the scores of the individual cells. Based on the table, I agree with the authors' assessment of their results but point out that the best cutoff score to use for identifying putative doublets probably should be assessed by looking at the full distribution (histogram) of the cells' doublet scores. And if the distribution reveals clear outliers, you can remove these cells before further analysis.

→ Thank you for the helpful comment.

Reviewer #2 (Remarks to the Author):

I appreciate the effort the authors made to address my comments and those of the other reviewers. While the authors address most of my comments there are still some minor remaining issues the authors could address.

→ Thank you for the kind comment. We did our best to address the remaining minor issues below.

An important conceptual point of this manuscript, which is extensively discussed by the authors, is the definition of Kdm6b activity in MN subtype specification through Kdm6b acting as a co-activator of Is1-Lhx3 complex. However, the changes in MN subtype generation when Kdm6b is deleted only in post-

mitotic MN (*Kdm6b*-cKO;*Isl1*-cre) does not recapitulate mutants with deletion at progenitor stages (*Kdm6b*-cKO;*Olig2*-cre) weaken the model proposed by the authors. In their revised manuscript, the authors provide immunocytochemistry data for *Kdm6b* expression suggesting for a delayed loss of *Kdm6b* expression in MN of *Kdm6b*-cKO;*Isl1*-cre mutants (Fig 1b and Fig9b), which could account for the difference of phenotype between the two mutants. Regarding this there are certain remaining concerns:

1. From the data presented by the authors it is difficult to access the level of *Kdm6b* expression remaining in MNs. Data on the *Kdm6b*-cKO;*Isl1*-cre (Figure 9b) is not very clear as the signal both for controls and mutants is very weak. Can the authors provide better images? There seems to be some ventral signal in the mutants at E11.5 and E12.5. Is this expression in MNs? Are the remaining *Kdm6b* cells in the mutants *Mnx1*+ or *Isl1*+?

→ We provided better images in Figure 9b in the revised manuscript. Our immunostaining data suggest that, in *Kdm6b*-cKO;*Isl1*-Cre spinal cord, *Kdm6b* expression is detected at E11.5, but is markedly reduced at E12.5, suggesting a delayed removal of *Kdm6b* protein in *Kdm6b*-cKO;*Isl1*Cre mice. In E11.5 *Kdm6b*-cKO;*Isl1*Cre mice, *Kdm6b*+ ventral cells are *Isl1*/*Mnx1* positive. In E12.5 *Kdm6b*-cKO;*Isl1*Cre mice, *Kdm6b* protein is much lower and more cytoplasmic than in control mice. Regarding the slightly remaining *Kdm6b* protein in *Kdm6b*-cKO mice, it should be noted that the mouse *Kdm6b* gene consists of 23 exons. Upon Cre-mediated recombination, *Kdm6b* flox allele loses only exons 14-20, encoding the enzymatic JmjC domain (1339-1501 amino acids for human *Kdm6b*, 1337-1500 amino acids for mouse *Kdm6b*)¹, but a significant fraction of *Kdm6b*-coding exons (exons 1-13, and exons 21-23) remains intact in Cre-mediated recombined *Kdm6b* flox allele (Rebuttal Figure 1). Thus, it is possible that the recombined *Kdm6b* flox allele by Cre still expresses *Kdm6b* mRNA and may produce *Kdm6b* protein lacking JmjC domain at low levels. If this is the case, our *Kdm6b* antibody (antigenic area, 798-1095 amino acids for human *Kdm6b*) would still detect JmjC domain-lacking *Kdm6b* protein expressed from the recombined *Kdm6b* flox allele. This may have contributed a weak *Kdm6b* immunostaining signal in *Kdm6b*-cKO.

2. Due to the late loss of Kdm6b in the MN pool as revealed by the new data presented by the authors (between E11.5-E12.5), the authors should analyse switching of MN subtypes at stages later than E12.5 (maybe E13.5/E14.5). This would help to understand if the difference of phenotype relative to Kdm6b-cKO;Olig2-cre mutants reflects just a temporal delay of protein loss or a change of plasticity of differentiating MN. This is an important point that is also raised by the authors in the discussion section that state (line 616): “Our data also suggest that MNs lose the plasticity in switching among different MN subtypes as they acquire terminally differentiated state.”

→ We agree with the reviewer that the MN subtype analysis in Kdm6b-cKO;Isl1Cre needs to be done later than E12.5. Thus, we presented the data from E13.5 Kdm6b-cKO;Isl1Cre mice (Figure 9, the embryonic stage is noted in Figure legend).

3. In Kdm6b-cKO;Olig2-cre mutants how is the expression of Kdm6b in MN at E9.5 and E10.5 embryos? Is expression of Lhx3 reduced/lost in MNs at E10.5? Is there a premature induction of Foxp1 in MNs at E10.5?

→ Kdm6b protein signal was very faint at E10.5 with our Kdm6b antibody and we did not observe a significant difference between Kdm6b-cKO;Olig2Cre and control mice at E10.5 (Rebuttal Fig. 2a). This raises the possibility that there is a delay between Kdm6b mRNA expression and Kdm6b protein expression, which has been seen with other genes in the developing spinal cord. Consistently, the expression of Lhx3 or Foxp1 did not exhibit a remarkable difference between Kdm6b-cKO;Olig2Cre and control mice at E10.5 (Rebuttal Fig. 2b). Likewise, neither Isl1 nor Mnx1 changed in E10.5 Kdm6b-cKO;Olig2Cre (Supplementary Fig. 1a).

In Fig.8b the authors should precisely define the combination of markers used to quantify different MN subtypes and how the total number of MNs were calculated.

→ We defined MN subtypes based on the following criteria, as now noted in the figure legend for Figure 8b.

Brachial spinal cord

MMC: Lhx3+ cells in the ventral spinal cord

LMCm: Foxp1+ Isl1+ cells in the ventro-lateral area

LMCl: Foxp+ Isl1- cells in the ventro-lateral area

Total LMC: Foxp1+ cells in the ventro-lateral area

Total MNs: MMC + LMC

Thoracic spinal cord

MMC: Lhx3+ cells in the ventral spinal cord

HMC: Isl1+ Lhx3- cells in the ventral spinal cord

PGC-Isl1+: Foxp1+ Isl1+ cells in the intermediate spinal cord

PGC-Isl1-: Foxp+ Isl1- cells in the intermediate spinal cord

Total PGC: Foxp1+ cells in the intermediate spinal cord

Total MNs: MMC+HMC+PGC

In Figures 4a and 7a the authors maintain consistency of cluster numbering and colour coding, but not in Fig.6a. The authors should reformat Fig. 6a to maintain consistency.

→ We kept the original colors from the Seurat² program for Fig. 6a, because a new type of MN cluster (presumptive MMC/HMC) emerged from cKO only analysis and also two clusters were present for each of LMCm and LMCl-like cell types. Although we attempted to match the cell types between control only analysis and cKO only analysis as best as we could, as requested in the previous review cycle, control and cKO clusters were not completely matched cell types and there are some fluidities in determining the cell identity of cKO cells depending on the analysis tools. The analysis of merged datasets increased our confidence in this matter, as in this analysis, control cells provide “anchoring points” for each of MN subtypes and cKO cells would cluster with the most similar control cell types based on single-cell transcriptomes. Thus, we thought the original colors and clustering from the Seurat would show unbiased outcomes.

REDACTED

In the discussion section where the authors discuss MN differentiation and acquisition of MN subtypes, it is not clear why Lhx3 would become downregulated with time, considering that Kdm6b is expressed in the entire MN population, which would reinforce/maintain Lhx3/Isl1 expression via Mnx1.

→ Our and others' data suggest that a complex interplay of transcription factors and coregulators is involved in MN subtype diversification. One mechanism driving Lhx3 downregulation is the induction of

Foxp1, which suppresses Lhx3 expression³ in LMC and PGC. Foxp1 induction may interfere with the action of Mnx1 in maintaining Lhx3 expression, which would be an exciting topic for future study.

Fig. 1b the authors should update the mutant nomenclature.

Line 338: typo. The authors may mean “Kdm6b-null MNs” instead of “Kdm6a-null MNs”.

→ Thank you for catching our mistake. We corrected the label in Figure 1b and Line 338.

Reviewer #4 (Remarks to the Author):

Point 1:

\ The authors have addressed this concern and the co-presentation of single-cell sequencing and histological analyses are interesting and strengthen each other.

→ Thank you.

Point 2:

\ The authors have addressed this concern.

→ Thank you.

Point 3:

\ This analysis is a bit superficial, as the authors could also perform comparative analysis in the sequencing data. While it would extend the relevance of their findings by including interneuron analysis, it is not central to the main claims of this manuscript.

→ We agree that our scRNAseq results provide rich information, including the transcriptome profiles of interneurons. However, as the reviewer acknowledged, such analysis is not directly related to the main theme of the paper and thus would rather make the manuscript too unfocused. We uploaded raw data of our single-cell RNAseq and any scientists interested in interneurons will be able to analyze the data.

Point 4:

\ The authors have addressed this concern.

→ Thank you.

Point 5:

\ The authors have not fully addressed this concern. As stated, e12.5 is quite a late timepoint to see such early MN progenitors and while the author’s explanation is plausible – a hybrid state of late progenitors/early MN that co-express markers for both, this could easily be tested directly with in situ hybridization in tissue sections. As the authors also observe some glial progenitor markers, it is particularly important to determine the identity of cells used as the anchor for Monocle analysis. In addition, the authors overstate conclusions from the lineage analysis, which can be used to find “trajectories” through any set of cells/clusters. Given that this is from single replicate data from a single timepoint, the authors should scale back their claims about developmental lineage.

→ Thank you for the feedback. Taking the reviewer’s points 5 and 7 (below) and considering that the lineage analysis is not critical for the overall conclusion of the paper as the reviewer noted in the previous review, we removed a paragraph describing the developmental trajectory analysis from the revised manuscript.

Of note, we performed immunostaining with the pMN progenitor marker Olig2 and the MN marker Mnx1 to test a hybrid state of late progenitors/early MN that co-express markers for both in tissue sections, as the reviewer asked. We found cells that co-express Olig2 and Mnx1 (Rebuttal Fig. 3), consistent with scRNAseq data. These results are also consistent with the previous reports that Mnx1 and Lhx3/4 are expressed in Mpm2+ proliferating progenitors^{4,5}.

Point

6:

\ The authors have addressed this concern, however, they should also note that this could reflect doublets. While they have performed some doublet analysis, they correctly state above that this is difficult to interpret during developmental time-points. “Barnyard” analysis of MN markers vs V2/V3 markers on a cell by cell basis could help to resolve this, though this is not central to the main claims of the manuscript and is not necessary.

→ Thank you for the insights. In the revised manuscript, we noted the limitation of single-cell analyses during developmental time-points; it is challenging to distinguish doublets and cells transitioning from one state to the next state (i.e., differentiation) in single-cell transcriptome data from developing tissues because, by definition, cells in the middle of a trajectory are always intermediate between other cells and are liable to be incorrectly detected as doublets⁶.

Point 7:

\ Please see the comments above regarding Slc18a3. This work is not conclusive and the authors should scale back their conclusions about lineage or remove this analysis.

→ As the reviewer advised, we removed the developmental lineage analysis in the revised manuscript.

Point 8:

\ It is appreciated that the authors attempted this and they have addressed this concern.

→ Thank you.

Misc. Small Comments

\ The authors should edit the text in line 328 to specify that a single sample was used for each condition. In addition, they should add a sentence to specify that the single-cell sequencing analysis was considered exploratory (because only one replicate was used) but key findings were validated with independent means (eg. antibody staining).

→ We revised the result, discussion and method sections, as advised.

⌋ The authors should present a merged table of Fig. 4b, 6b, and possibly Fig. 7 to facilitate comparison of cell numbers in each phase of the analysis/condition.

→ We added a merged table in Supplementary Figure 8c.

References

- 1 Iwamori, N., Iwamori, T. & Matzuk, M. M. H3K27 demethylase, JMJD3, regulates fragmentation of spermatogonial cysts. *PLoS one* **8**, e72689, doi:10.1371/journal.pone.0072689 (2013).
- 2 Satija, R., Farrell, J. A., Gennert, D., Schier, A. F. & Regev, A. Spatial reconstruction of single-cell gene expression data. *Nat Biotechnol* **33**, 495-502, doi:10.1038/nbt.3192 (2015).
- 3 Rousso, D. L., Gaber, Z. B., Wellik, D., Morrisey, E. E. & Novitsch, B. G. Coordinated actions of the forkhead protein Foxp1 and Hox proteins in the columnar organization of spinal motor neurons. *Neuron* **59**, 226-240, doi:10.1016/j.neuron.2008.06.025 (2008).
- 4 Sharma, K. *et al.* LIM homeodomain factors Lhx3 and Lhx4 assign subtype identities for motor neurons. *Cell* **95**, 817-828, doi:10.1016/s0092-8674(00)81704-3 (1998).
- 5 Thaler, J. *et al.* Active suppression of interneuron programs within developing motor neurons revealed by analysis of homeodomain factor HB9. *Neuron* **23**, 675-687, doi:10.1016/s0896-6273(01)80027-1 (1999).
- 6 Lun, A. T. L. *Detecting doublets in single-cell RNA-seq data*, (2019).

Reviewers' Comments:

Reviewer #1:

Remarks to the Author:

Overall, my questions have been addressed. I would suggest clarifying in the text (main and/or Methods) how the analysis was performed for Supplementary Figure 8. As I understand it (based on the rebuttal), CCA integration was used for Figure 7 and not Supp Fig 8, which could be noted at line 375, for example.

Reviewer #2:

Remarks to the Author:

The authors have satisfactorily addressed most issues and concerns raised, but they have not fully addressed point 3 regarding the early dynamics of Foxp1-MN generation in Kdm6bcko-Olig2-Cre mutants. The authors provide in their rebuttal letter data indicating that the early specification of Foxp1-MN at E10.5 is unaffected in these mutants. The authors should include this data in the revised manuscript (as supplementary data) and provide the corresponding quantifications in Fig.1d. This is important for the reader to fully understand the changes of MN specification in this mutant and the role of Kdm6b in the dynamics of MN subtype specification. With this data included, I think the manuscript is suitable for publication in Nature Communications

Reviewer #4:

Remarks to the Author:

The authors have addressed each of my concerns.

REVIEWER COMMENTS

Reviewer #1 (Remarks to the Author):

Overall, my questions have been addressed. I would suggest clarifying in the text (main and/or Methods) how the analysis was performed for Supplementary Figure 8. As I understand it (based on the rebuttal), CCA integration was used for Figure 7 and not Supp Fig 8, which could be noted at line 375, for example. → CCA integration was noted at line 375 in the main text. Supplementary Fig. 8 legend was also revised to clarify that the CCA integration was not used for the data in Supplementary Fig. 8a and b. The detailed analysis method was provided in the Methods section.

Reviewer #2 (Remarks to the Author):

The authors have satisfactorily addressed most issues and concerns raised, but they have not fully addressed point 3 regarding the early dynamics of Foxp1-MN generation in Kdm6bcko-Olig2-Cre mutants. The authors provide in their rebuttal letter data indicating that the early specification of Foxp1-MN at E10.5 is unaffected in these mutants. The authors should include this data in the revised manuscript (as supplementary data) and provide the corresponding quantifications in Fig.1d. This is important for the reader to fully understand the changes of MN specification in this mutant and the role of Kdm6b in the dynamics of MN subtype specification. With this data included, i think the manuscript is suitable for publication in Nature Communications
→ We added the requested data, including quantification graphs, in Supplementary Figures 1a and b. We also updated the text to reflect these changes.

Reviewer #4 (Remarks to the Author):

The authors have addressed each of my concerns.